# Navigating Text-to-Image Customization: From LyCORIS Fine-Tuning to Model Evaluation

**Shih-Ying Yeh**[*]
NTHU
ay@kblueleaf.net

**Yu-Guan Hsieh**[*†]
Apple
cyberhsieh212@gmail.com

**Zhidong Gao**[*]
UTSA
zhidong.gao@utsa.edu

**Bernard B W Yang**
University of Toronto
by3976@gmail.com

**Giyeong Oh**
Yonsei University
hard2251@yonsei.ac.kr

**Yanmin Gong**
UTSA
gongyanmin@gmail.com

## Abstract

Text-to-image generative models have garnered immense attention for their ability to produce high-fidelity images from text prompts. Among these, Stable Diffusion distinguishes itself as a leading open-source model in this fast-growing field. However, the intricacies of fine-tuning these models pose multiple challenges from new methodology integration to systematic evaluation. Addressing these issues, this paper introduces LyCORIS (Lora beYond Conventional methods, Other Rank adaptation Implementations for Stable diffusion), an open-source library that offers a wide selection of fine-tuning methodologies for Stable Diffusion. Furthermore, we present a thorough framework for the systematic assessment of varied fine-tuning techniques. This framework employs a diverse suite of metrics and delves into multiple facets of fine-tuning, including hyperparameter adjustments and the evaluation with different prompt types across various concept categories. Through this comprehensive approach, our work provides essential insights into the nuanced effects of fine-tuning parameters, bridging the gap between state-of-the-art research and practical application.

## 1 Introduction

The recent advancements in deep generative models along with the availability of vast data on the internet have ushered in a new era of text-to-image synthesis (Balaji et al., 2022; Ramesh et al., 2022; Saharia et al., 2022). These models allow users to transform text prompts into high-quality, visually appealing images, revolutionizing the way we conceive of and interact with digital media (Ko et al., 2023; Zhang et al., 2023). Moreover, the models' wide accessibility and user-friendly interfaces extend their influence beyond the research community to laypeople who aspire to create their own artworks. Among these, Stable Diffusion (Rombach et al., 2022) emerges as one of the pioneering open-source models offering such capabilities. Its open-source nature has served as a catalyst for a multitude of advances, attracting both researchers and casual users alike. Extensions such as cross-attention control (Liu et al., 2022) and ControlNet (Zhang et al., 2023) have further enriched the landscape, broadening the model's appeal and utility.

While these models offer an extensive repertoire of image generation, they often fall short in capturing highly personalized or novel concepts, leading to a burgeoning interest in model customization techniques. Initiatives like DreamBooth (Ruiz et al., 2023) and Textual Inversion (Gal et al., 2023) have spearheaded efforts in this domain, allowing users to imbue pretrained models like Stable Diffusion with new concepts through a small set of representative images (see Appendix A for detailed related work). Coupled with user-friendly trainers designed to customize Stable Diffusion, the ecosystem now boasts a plethora of specialized models and dedicated platforms that host them—often witnessing the upload of thousands of new models to a single website in just one week.

---

[*]Equal contribution.
[†]Corresponding author. Work done during the author's Ph.D. at Université Grenoble Alpes.

In spite of this burgeoning landscape, our understanding of the intricacies involved in fine-tuning these models remains limited. The complexity of the task—from variations in datasets, image types, and captioning strategies, to the abundance of available methods each with their own sets of hyperparameters—renders it a challenging terrain to navigate. While new methods proposed by researchers offer much potential, they are not always seamlessly integrated into the existing ecosystem, which can hinder comprehensive testing and wider adoption. Moreover, current evaluation paradigms lack a systematic approach that covers the full depth and breadth of what fine-tuning entails. To address these gaps and bridge the divide between research innovations and casual usage, we present our contributions as follows.

1. We develop LyCORIS, an open source library dedicated to fine-tuning of Stable Diffusion. This library encapsulates a spectrum of methodologies ranging from the most standard LoRA to a number of emerging strategies such as LoHa, LoKr, GLoRA, and (IA)$^3$ that are newer and lesser-explored in the context of text-to-image models.

2. To enable rigorous comparisons between methods, we propose a comprehensive evaluation framework that incorporates a wide range of metrics, capturing key aspects such as concept fidelity, text-image alignment, image diversity, and preservation of the base model's style.

3. Through extensive experiments, we compare the performances of different fine-tuning algorithms implemented in LyCORIS and assess the impacts of various hyperparameters, offering insights into how these factors influence the results.

## 2 Preliminary

In this section, we briefly review the two core components of our study: Stable Diffusion and LoRA for model customization.

### 2.1 STABLE DIFFUSION

Diffusion models (Ho et al., 2020; Sohl-Dickstein et al., 2015) are a family of probabilistic generative models that are trained to capture a data distribution through a sequence of denoising operations. Given an initial noise map $\mathbf{x}_T \sim \mathcal{N}(0, \mathbf{I})$, the models iteratively refine it by reversing the diffusion process until it is synthesized into a desired image $\mathbf{x}_0$. These models can be conditioned on elements such as text prompts, class labels, or low-resolution images, allowing conditioned generation.

Specifically, our work is based on Stable Diffusion, a text-to-image latent diffusion model (Rombach et al., 2022) pretrained on the LAION 5-billion image dataset (Schuhmann et al., 2022). Latent diffusion models reduce the cost of diffusion models by shifting the denoising operation into the latent space of a pre-trained variational autoencoder, composed of an encoder $\mathcal{E}$ and a decoder $\mathcal{D}$. During training, the noise is added to the encoder's latent output $\mathbf{z} = \mathcal{E}(\mathbf{x}_0)$ for each time step $t \in \{0, \dots, T\}$, resulting in a noisy latent $\mathbf{z}_t$. Then, the model is trained to predict the noise applied to $\mathbf{z}_t$, given text conditioning $\mathbf{c} = \mathcal{T}(l)$ obtained from an image description $l$ (also known as the image's caption) using a text encoder $\mathcal{T}$. Formally, with $\theta$ denoting the parameter of the denoising U-Net and $\boldsymbol{\epsilon}_\theta(\cdot)$ representing the predicted noise from this model, we aim to minimize

$$\mathcal{L}(\theta) = \mathbb{E}_{\mathbf{x}_0, \mathbf{c}, \boldsymbol{\epsilon}, t}[||\boldsymbol{\epsilon} - \boldsymbol{\epsilon}_\theta(\mathbf{z}_t, t, \mathbf{c})||_2^2], \tag{1}$$

where $\mathbf{x}_0$, $\mathbf{c}$ are drawn from the dataset, $\boldsymbol{\epsilon} \sim \mathcal{N}(0, \mathbf{I})$, and $t$ is uniformly drawn from $\{1, \dots, T\}$.

### 2.2 MODEL CUSTOMIZATION WITH LoRA

To enable more personalized experiences, model customization has been proposed as a means to adapt foundational models to specific domains or concepts. In the case of Stable Diffusion, this frequently involves fine-tuning a pretrained model by minimizing the original loss function (1) on a new dataset, containing as few as a single image for each target concept. In this process, we introduce a *concept descriptor* $[V]$ for each target concept, comprising a neutral *trigger word* $[V_{\text{trigger}}]$ and an optional *class word* $[V_{\text{class}}]$ to denote the category to which the concept belongs. This concept descriptor is intended for use in both the image captions and text prompts. While it is possible to include a prior-preservation loss by utilizing a set of regularization images (Kumari et al., 2023; Ruiz et al., 2023), we have chosen not to employ this strategy in the current study.

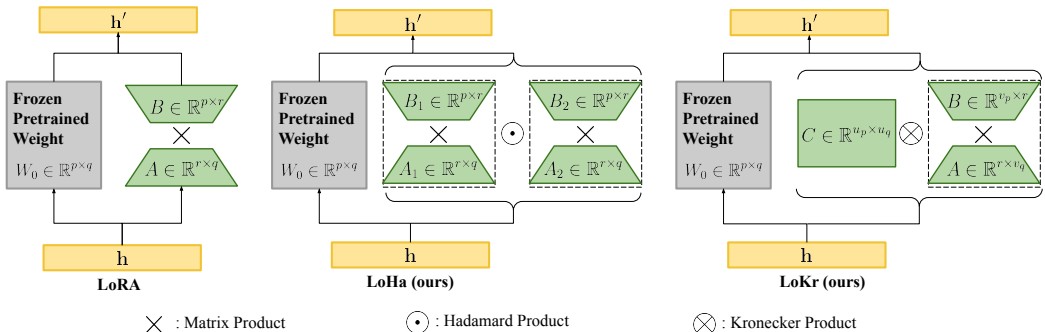

**Figure 1:** This figure shows the structure of the proposed LoHa and LoKr modules implemented in LyCORIS.

**Low-Rank Adaptation (LoRA).**    When integrated into the model customization process, Low-Rank Adaptation (LoRA) could substantially reduce the number of parameters that need to be updated. It was originally developed for large language models (Hu et al., 2021), and later adapted for Stable Diffusion by Simo Ryu (2022). LoRA operates by constraining fine-tuning to a low-rank subspace of the original parameter space. More specifically, the *weight update* $\Delta W \in \mathbb{R}^{p \times q}$ is pre-factorized into two low-rank matrixes $B \in \mathbb{R}^{p \times r}, A \in \mathbb{R}^{r \times q}$, where $p, q$ are the dimensions of the original model parameter, $r$ is the dimension of the low-rank matrix, and $r \ll \min(p, q)$. During fine-tuning, the foundational model parameter $W_0$ remains frozen, and only the low-rank matrices are updated. Formally, the forward pass of $\mathbf{h}' = W_0\mathbf{h} + \mathbf{b}$ is modified to:

$$\mathbf{h}' = W_0\mathbf{h} + \mathbf{b} + \gamma \Delta W \mathbf{h} = W_0\mathbf{h} + \mathbf{b} + \gamma BA\mathbf{h}, \tag{2}$$

where $\gamma$ is a *merge ratio* that balances the retention of pretrained model information and its adaptation to the target concepts.[1] Following Hu et al. (2021), we further define $\alpha = \gamma r$ so that $\gamma = \alpha/r$.

## 3   The LyCORIS Library

Building upon the initiative of LoRA, this section introduces LyCORIS, our open-source library that provides an array of different methods for fine-tuning Stable Diffusion.

### 3.1   DESIGN AND OBJECTIVES

LyCORIS stands for ***L**ora be**y**ond **C**onventional methods, **O**ther **R**ank adaptation **I**mplementations for **S**table diffusion*. Broadly speaking, the library's main objective is to serve as a test bed for users to experiment with a variety of fine-tuning strategies for Stable Diffusion models. Seamlessly integrating into the existing ecosystem, LyCORIS is compatible with easy-to-use command-line tools and graphic interfaces, allowing users to leverage the algorithms implemented in the library effortlessly. Additionally, native support exists in popular user interfaces designed for image generation, facilitating the use of models fine-tuned through LyCORIS methods. For most of the algorithms implemented in LyCORIS, stored parameters naturally allow for the reconstruction of the weight update $\Delta W$. This design brings inherent flexibility: it enables the weight updates to be scaled and applied to a base model $W_0'$ different from those originally used for training, expressed as $W' = W_0' + \lambda \Delta W$. Furthermore, a weight update can be combined with those from other fine-tuned models, further compressed, or integrated with advanced tools like ControlNet. This opens up a diverse range of possibilities for the application of these fine-tuned models.

### 3.2   IMPLEMENTED ALGORITHMS

We now discuss the core of the library—the algorithms implemented in LyCORIS. For conciseness, we will primarily focus on three main algorithms: LoRA (LoCon), LoHa, and LoKr. The merge ratio $\gamma = \alpha/r$ introduced in (2) is implemented for all these methods.

---

[1]Setting $\gamma$ is mathematically equivalent to scaling the initialization of $B$ and $A$ by $\sqrt{\gamma}$ and scaling the learning rate by $\sqrt{\gamma}$ or $\gamma$, depending on the used optimizer. See Appendix B.1 for a generalization of this result.

**LoRA (LoCon).** In the work of Hu et al. (2021), the focus was centered on applying the low-rank adapter to the attention layer within the large language model. In contrast, the convolutional layers play a key role in Stable Diffusion. Therefore, we extend the method to the convolutional layers of diffusion models (details are provided in Appendix B.2). The intuition is with more layers getting involved during fine-tuning, the performance (generated image quality and fidelity) should be better.

**LoHa.** Inspired by the basic idea underlying LoRA, we explore the potential enhancements in fine-tuning methods. In particular, it is well recognized that methods based on matrix factorization suffer from the *low-rank constraint*. Within the LoRA framework, weight updates are confined within the low-rank space, inevitably impacting the performance of the fine-tuned model. To achieve better fine-tuning performance, we conjecture that a relatively large rank might be necessary, particularly when working with larger fine-tuning datasets or when the data distribution of downstream tasks greatly deviates from the pretraining data. However, this cloud leads to increased memory usage and more storage demands.

FedPara (Hyeon-Woo et al., 2022) is a technique originally developed for federated learning that aims to mitigate the low-rank constraint when applying low-rank decomposition methods to federated learning. One of the advantages of FedPara is that the maximum rank of the resulting matrix is larger than those derived from conventional low-rank decomposition (such as LoRA). More precisely, for $\Delta W = (B_1 A_1) \odot (B_2 A_2)$, where $\odot$ denotes the Hadamard product (element-wise product), $B_1, B_2 \in \mathbb{R}^{p \times r}$, $A_1, A_2 \in \mathbb{R}^{r \times q}$, and $r \leq \min(p, q)$, the rank of $\Delta W$ can be as large as $r^2$. To make a fair comparison, we assume the low-rank dimension in equation (2) is $2r$, such that they have the same number of trainable parameters. Then, the reconstructed matrix $\Delta W = BA$ has a maximum rank of $2r$. Clearly, $2r < r^2$, if $r > 2$. This implies decomposing the weight update with the Hadamard product could improve the fine-tuning capability given the same number of trainable parameters. We term this method as LoHa (**Lo**w-rank adaptation with **Ha**damard product). The forward pass of $\mathbf{h}' = W_0 \mathbf{h} + \mathbf{b}$ is then modified to:

$$\mathbf{h}' = W_0 \mathbf{h} + \mathbf{b} + \gamma \Delta W \mathbf{h} = W_0 \mathbf{h} + \mathbf{b} + \gamma \left[ (B_1 A_1) \odot (B_2 A_2) \right] \mathbf{h}. \tag{3}$$

**LoKr.** In the same spirit of maximizing matrix rank while minimizing parameter count, our library offers LoKr (**Lo**w-rank adaptation with **Kr**onecker product) as another viable option. This method is an extension of the KronA technique, initially proposed by Edalati et al. (2022) for fine-tuning of language models, and employs Kronecker products for matrix decomposition. Importantly, we have adapted this technique to work with convolutional layers, similar to what we achieved with LoCon. A unique advantage of using Kronecker products lies in the multiplicative nature of their ranks, allowing us to move beyond the limitations of low-rank assumptions.

Going further, to provide finer granularity for model fine-tuning, we additionally incorporate an optional low-rank decomposition (which users can choose to apply or not) that focuses exclusively on the right block resulting from the Kronecker decomposition.[2] In summary, writing $\otimes$ for the Kronecker product, the forward pass $\mathbf{h}' = W_0 \mathbf{h} + \mathbf{b}$ is modified to:

$$\mathbf{h}' = W_0 \mathbf{h} + \mathbf{b} + \gamma \Delta W \mathbf{h} = W_0 \mathbf{h} + \mathbf{b} + \gamma \left[ C \otimes (BA) \right] \mathbf{h}, \tag{4}$$

The size of these matrices are determined by two user-specified hyperparameters: the factor $f$ and the dimension $r$. With these, we have $C \in \mathbb{R}^{u_p \times u_q}$, $B \in \mathbb{R}^{v_p \times r}$, and $A \in \mathbb{R}^{r \times v_q}$, where

$$u_p = \max \left( u \leq \min(f, \sqrt{p}) \mid p \bmod u = 0 \right), \quad v_p = \frac{p}{u_p}. \tag{5}$$

The two scalars $u_q$ and $v_q$ are defined in the same way. Interestingly, LoKr has the widest range of potential parameter counts among the three methods and can yield the smallest file sizes when appropriately configured. Additionally, it can be interpreted an adapter that is composed of a number of linear layers, as detailed in Appendix B.3.

**Others.** In addition to LoRA, LoHa, and LoKr described earlier, our library features other algorithms including DyLoRA (Valipour et al., 2022), GLoRA (Chavan et al., 2023), and (IA)$^3$ (Liu et al., 2022). Moreover, between the date of submission and the preparation of the camera-ready version for the main conference, we have further expanded LyCORIS by incorporating more recent advancements, notably OFT (Qiu et al., 2023), BOFT (Liu et al., 2024), and DoRA (Liu et al., 2024). However, the discussion of these supplementary algorithms is beyond the scope of this paper.

---

[2]As shown in Eq. (5), in our implementation, the right block is always the larger of the two.

# 4 Evaluating Fine-Tuned Text-To-Image Models

With the wide range of algorithmic choices and hyperparameter settings made possible by LyCORIS, one naturally wonders: Is there an optimal algorithm or set of hyperparameters for fine-tuning Stable Diffusion? To tackle this question in a comprehensive manner, it is essential to first establish a clear framework for model evaluation.

With this in mind, in this section, we turn our focus to two independent but intertwined components that are crucial for a systematic evaluation of fine-tuned text-to-image models: *i*) the types of prompts used for image generation and *ii*) the evaluation of the generated images. While these two components are commonly considered as a single entity in existing literature, explicitly distinguishing between them allows for a more nuanced evaluation of model performance (see Appendix A for a comprehensive overview of related works on text-to-image model evaluation). Below, we explore each of these components in detail.

## 4.1 CLASSIFICATION OF PROMPTS FOR IMAGE GENERATION

To fully understand the model's behavior, it is important to distinguish between different types of prompts that guide image generation. We categorize these into three main types as follows:

- **Training Prompts**: These are the prompts originally used for training the model. The images generated from these prompts are expected to closely align with the training dataset, providing insight into how well the model has captured the target concepts.

- **Generalization Prompts**: These prompts seek to generate images that generalize learned concepts to broader contexts, going beyond the specific types of images encountered in the training set. This includes, for example, combining the innate knowledge of the base model with the learned concepts, combining concepts trained within the same model, and combining concepts trained across different models which are later merged together. Such prompts are particularly useful to evaluate the disentanglement of the learned representations.

- **Concept-Agnostic Prompts**: These are prompts that deliberately avoid using trigger words from the training set and are often employed to assess concept leak, see e.g., Kumari et al. (2023). When training also involves class words, this category can be further refined to distinguish between prompts that do and do not use these class words.

## 4.2 EVALUATION CRITERIA

After detailing the different types of prompts that guide the image generation process, the next important step is to identify the aspects that we would like to look at when evaluating the generated images, as we outline below.

- **Fidelity** measures the extent to which generated images adhere to the target concept.

- **Controllability** evaluates the model's ability to generate images that align well with text prompts.

- **Diversity** assesses the variety of images that are produced from a single or a set of prompts.

- **Base Model Preservation** measures how much fine-tuning affects the base model's inherent capabilities, particularly in ways that may be undesirable. For example, if the target concept is an object, retaining the background and style as generated by the base model might be desired.

- **Image Quality** concerns the visual appeal of the generated images, focusing primarily on aspects like naturalness, absence of artifacts, and lack of weird deformations. Aesthetics, though related, are considered to be more dependent on the dataset than on the training method, and are therefore not relevant for our purpose.

Taken together, the prompt classification of Section 4.1 and the evaluation criteria listed above offer a nuanced and comprehensive framework for assessing fine-tuned text-to-image models. Notably, these tools also enable us to evaluate other facets of model performance, such as the ability to learn multiple distinct concepts without mutual interference and the capability for parallel training of multiple models that can later be successfully merged.

# 5 Experiments

In this section, we perform extensive experiments to compare different LyCORIS algorithms and to assess the impact of the hyperparameters. Our experiments employ the non-EMA version of Stable Diffusion 1.5 as the base model. All the experimental details not included in the main text along with presentations of additional experiments can be found in the appendix.

## 5.1 Dataset

Contrary to prior studies that primarily focus on single-concept fine-tuning with very few images, we consider a dataset that spans across a wide variety of concepts with an imbalance in the number of images for each. Our dataset is hierarchically structured, featuring 1,706 images across five categories: anime characters, movie characters, scenes, stuffed toys, and styles. These categories further break down into various classes and sub-classes. Importantly, classes under "scenes" and "stuffed toys" contain only 4 to 12 images, whereas other categories have 45 to 200 images per class.

The influence of training captions on the fine-tuned model is also widely acknowledged in the community. It is particularly observed that training with uninformative captions such as "`A photo of [V]`", which are commonly employed in the literature, can lead to subpar results. In light of this, we use a publicly available tagger to tag the training images. We then remove tags that are inherently tied to each target concept. The resulting tags are combined with the concept descriptor to create more informative captions as "`[V], {tag1}, ..., {tagk}`". To justify this choice, comparative analyses for models trained using different captions are presented in Appendix H.3.

## 5.2 Algorithm Configuration and Evaluation

Our experiments focus on methods that are implemented in the LyCORIS library, and notably LoRA, LoHa, LoKr, and native fine-tuning (note that DreamBooth Ruiz et al., 2023 can be simply regarded as native fine-tuning with regularization images). For each of these four algorithms, we define a set of default hyperparameters and then individually vary one of the following hyperparameters: learning rate, trained layers, dimension and alpha for LoRA and LoHa, and factor for LoKr. This leads to 26 distinct configurations. For each configuration, three models are trained using different random seeds, and three checkpoints are saved along each fine-tuning, giving in this way 234 checkpoints in the end. While other parameter-efficient fine-tuning methods exist in the literature, most of the proposed modifications are complementary to our approach. We thus do not include them for simplicity.

**Data Balancing.** To address dataset imbalance, we repeat each image a number of times within each epoch to ensure images from different classes are equally exposed during training.

**Evaluation Procedure.** To evaluate the trained models, we consider the following four types of prompts *i*) <train> training captions, *ii*) <trigger> concept descriptor alone, *iii*) <alter> generalization prompts with content alteration, and *iv*) <style> generalization prompts with style alteration. Using only the concept descriptor tests if the model can accurately reproduce the concept without using the exact training captions. As for generalization prompts, only a single target concept is involved. This thus evaluates the model's ability to combine innate and fine-tuned knowledge. For each considered prompt type, we generate 100 images for each class or sub-class, resulting in a total of 14,900 images per checkpoint. Note that we do not include concept-agnostic prompts in our experiments.

**Evaluation Metrics.** Our evaluation metrics are designed to capture the criteria delineated in Section 4.2 and are computed on a per (sub)-class basis. We briefly describe below the metrics that are used for each criterion.

- **Fidelity**: We assess the similarity between the generated and dataset images using *average cosine similarity* and *squared centroid distance* between their DINOv2 embeddings (Oquab et al., 2023).
- **Controllability**: The alignment between generated images and corresponding prompts is measured via *average cosine similarity* in the CLIP feature space (Radford et al., 2021).
- **Diversity**: Diversity of images generated with a single prompt is measured by the *Vendi score* (Friedman & Dieng, 2023), calculated using the DINOv2 embeddings.
- **Base Model Preservation**: This generally needs to be evaluated on a case-by-case basis, depending on which aspect of the base model we would like to retain. Specifically, we examine

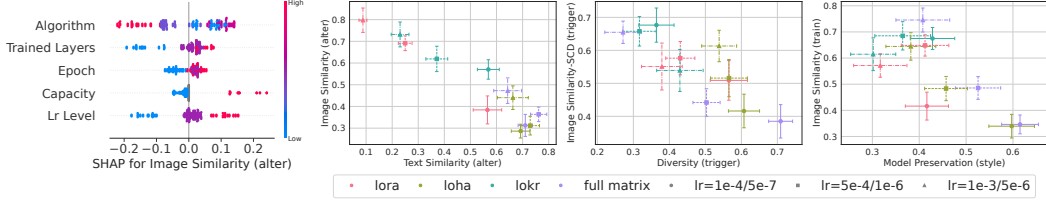

**(a)** Plots for category "movie characters" (scatter plots are for 30 epoch checkpoints)

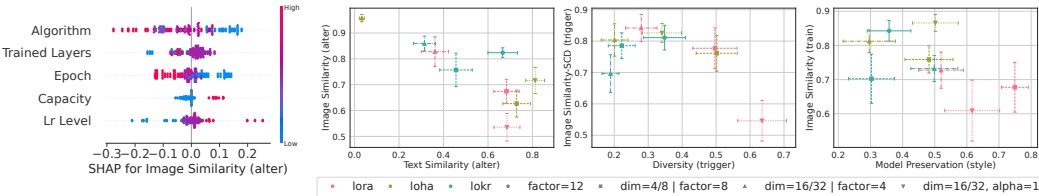

**(b)** Plots for category "scene" (scatter plots are for 10 epoch checkpoints)

**Figure 2:** SHAP beeswarm charts and scatter plots for analyzing the impact of change in different algorithm components. In the beeswarm plots, LoRA is in blue, LoHa is in purple, LoKr is in purple red, and native fine-tuning is in red. Model capacity is adjusted by either increasing dimension (for LoRA or LoHa) or decreasing factor (for LoKr). In the scatter plots, SCD indicates that we use squared centroid distance to measure image similarity. This removes the implicit penalization towards more diverse image sets in the computation of average cosine similarity (see Appendix D.4 for details). We believe it is thus more suitable when we are interested in the trade-off between fidelity and diversity. The error bars in the scatter plots represent standard errors of the metric values across random seeds and classes.

> potential style leakage by measuring the standard *style loss* (Johnson et al., 2016) between base and fine-tuned model outputs for <style> prompts.

- **Image Quality**: Although numerous methods have been developed for image quality assessment, most of them target natural images. As far as we are aware, currently, there still lacks a systematic approach for assessing the quality of AI-generated images. We attempted experiments with three leading pretrained quality assessment models, as detailed in Appendix H.2, but found them unsuitable for our context. We thus do not include any quality metrics in our primary experiments.

Further justification for our metric choices is provided in Appendices E and H.1, where we compute correlation coefficients for a wider range of metrics and conduct experiments across three classification datasets to assess the sensitivity of different image features to change in a certain image attribute.

## 5.3 EXPERIMENTAL RESULTS

To carry out the analysis, we first transform each computed metric value into a normalized score, ranging from 0 to 1, based on its relative ranking among all the examined checkpoints (234 in total). A score closer to 1 signifies superior performance. Subsequently, these scores are averaged across sub-classes and classes to generate a set of metrics for each category and individual checkpoint. Alongside the scatter plots, which directly indicate the values of these metrics, we also employ SHAP (SHapley Additive exPlanations) analysis (Lundberg & Lee, 2017) in conjunction with CatBoost regressor (Prokhorenkova et al., 2018) to get a clear visualization of the impact of different algorithm components on the considered metrics, as shown in Figure 2. In essence, a SHAP value quantifies the impact of a given feature on the model's output. For a more exhaustive presentation of the results, readers are referred to Appendix F.

### 5.3.1 CHALLENGES IN ALGORITHM EVALUATION

Before delving into our analysis, it is crucial to acknowledge the complexities inherent in evaluating the performance of fine-tuning algorithms. Specifically, we identify several key challenges, including *i*) sensitivity to hyperparameters, *ii*) performance discrepancy across concepts, *iii*) influence of dataset, *iv*) conflicting criteria, and *v*) unreliability of evaluation metrics, among others. These challenges are discussed in detail in Appendix C. To mitigate some of these issues, our evaluation encompasses a

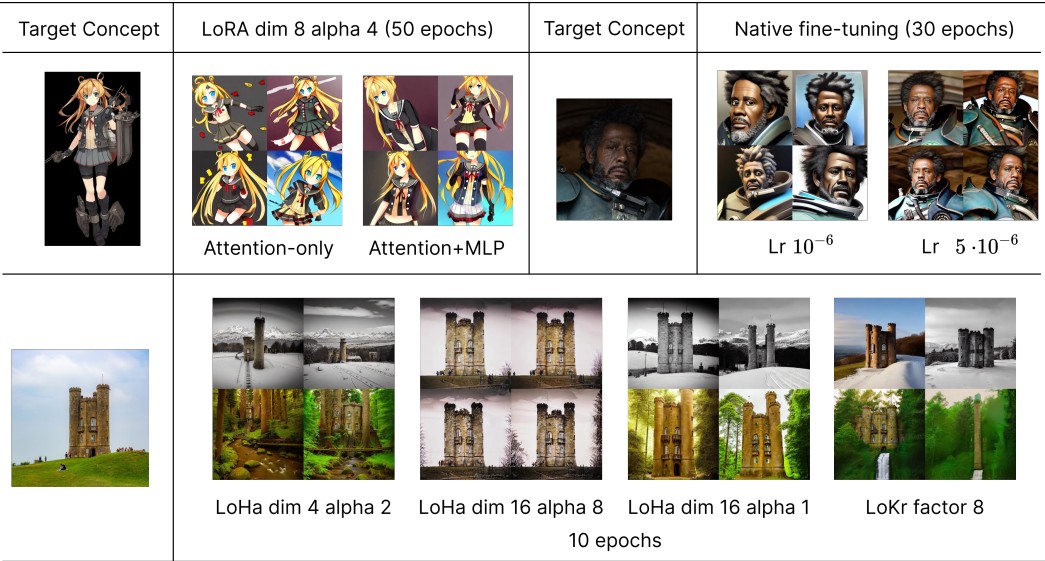

| Target Concept | LoRA dim 8 alpha 4 (50 epochs) | | Target Concept | Native fine-tuning (30 epochs) | |
|---|---|---|---|---|---|
| | Attention-only | Attention+MLP | | Lr $10^{-6}$ | Lr $5 \cdot 10^{-6}$ |

| | LoHa dim 4 alpha 2 | LoHa dim 16 alpha 8 | LoHa dim 16 alpha 1 | LoKr factor 8 |
|---|---|---|---|---|
| | | 10 epochs | | |

**Figure 3:** Qualitative comparison of checkpoints trained with different configurations. Samples of the top row are generated using only concept descriptors while samples of the bottom row are generated with the two prompts "[$V_{castle}$] scene stands against a backdrop of snow-capped mountains" and "[$V_{castle}$] scene surrounded by a lush, vibrant forest". The number of training epochs is chosen according to the concept category.

large number of configurations and includes separate analyses for each category. Additionally, we complement our quantitative metrics with visual inspections conducted throughout our experiments (see Appendix G for an extensive set of qualitative results). Finally, we exercise caution in making any definitive claims, emphasizing that there is no one-size-fits-all solution, and acknowledging that exceptions do exist to our guiding principles.

### 5.3.2 ANALYSIS OF RESULTS AND INSIGHTS FOR FINE-TUNING WITH LYCORIS

In this part, we delve into a detailed examination of our experimental results, aiming to glean actionable insights for fine-tuning with LyCORIS. These insights should not be considered as rigid guidelines, but rather as empirical observations designed to serve as foundational reference points.

**Number of Training Epochs.**  To analyze the evolution of models over training, we generate images from checkpoints obtained after 10, 30, and 50 epochs of training. Due to the small number of images available for "scenes" and "stuffed toys" categories and the use of data balancing, the 30 and 50 epoch checkpoints are almost universally overtrained for these concepts, explaining why increasing training epochs decreases image similarity as shown in the SHAP plot of Figure 2b.[3] Otherwise, increasing the number of epochs generally improves concept fidelity while compromising text-image alignment, diversity, and base model preservation. Exceptions to this trend exist. Specifically, we observe the *overfit-then-generalize* phenomenon, as often illustrated through the double descent curve (Nakkiran et al., 2021), in certain situations. We explore this further in Appendix G.4.2.

**Learning Rate.**  We consider three levels of learning rate, $5 \cdot 10^{-7}$, $10^{-6}$, and $5 \cdot 10^{-6}$ for native fine-tuning, and $10^{-4}$, $5 \cdot 10^{-4}$, and $10^{-3}$ for the other three algorithms. Within a reasonable range, increasing the learning rate seems to have the same effect as increasing the number of training epochs. A qualitative example is provided in Figure 3. It is worth noting, however, that an excessively low learning rate cannot be remedied by simply extending the training duration (see Appendix G.3.2).

**Algorithm.**  A central question driving the development of LyCORIS is to assess how different methods of decomposing the model update affect the final model's performance. We summarize our observations in Table 1. We distinguish here between two learning rates for native fine-tuning as they result in models that perform very differently. In particular, Figure 2a reveals that native fine-tuning at a learning rate of $5 \cdot 10^{-6}$ achieves high image similarity for training prompts and high text similarity

---

[3]We also experimented without data balancing and observed undertraining even after 50 epochs.

|  | LoRA | LoHa | LoKr | Native (lr $5 \times 10^{-6}$) | Native (lr $10^{-6}$) |
|---|---|---|---|---|---|
| Fidelity | 3 | 2 | 4 | 5 | 1 |
| Controllability | 2 | 4 | 1 | 3 | 5 |
| Diversity | 3 | 4 | 2 | 1 | 5 |

**Table 1:** A tentative ranking based on different evaluation criteria for the methods we explore in our experiments (the *higher* the number, the *better* the method's performance). Although this ranking reflects the general trend observed across different concept categories, deviations of varying degrees are common.

for generalization prompts. However, this strong performance in text similarity for generalization prompts also leads to a lower image similarity score for these prompts, highlighting in this way both the challenges in comprehensive method comparison and the necessity of evaluating on different types of prompts independently. As for LoRA, LoHa, and LoKr, comparisons are based on configurations with similar parameter counts and otherwise the same hyperparameters. Style preservation is not included in this comparison as no consistent trend across concept categories is observed.

**Trained Layers.** To investigate the effects of fine-tuning different layers, we examine three distinct presets: *i*) attn-only: where we only fine-tune attention layers; *ii*) attn-mlp: where we fine-tune both attention and feedforward layers; and *iii*) full network: where we fine-tune all the layers, including the convolutional ones. As can be seen from the SHAP plots in Figure 2, when all the other parameters are fixed, restricting fine-tuning to only the attention layers leads to a substantial decrease in image similarity while improving other metrics. This could be unfavorable in certain cases; for instance, the top-left example in Figure 3 shows that neglecting to fine-tune the feedforward layers prevents the model from correctly learning the character's uniform. The impact of fine-tuning the convolutional layers is less discernible, possibly because the metrics we use are not sensitive enough to capture subtle differences. Overall, our observations align with those made by Han et al. (2023). Moreover, it is worth noting that with the "attn-only" preset, we also fine-tune the self-attention layers in addition to the cross-attention layers, but this may still be insufficient, as demonstrated above.

**Dimension, Alpha, and Factor.** We finally inspect a number of parameters that are specific to our algorithms—dimension $r$, alpha $\alpha$, and factor $f$. By default, we set the dimension and alpha of LoRA to $8$ and $4$, and of LoHa to $4$ and $2$. As for LoKr, we set the factor to $8$ and do not perform further decomposition of the second block. These configurations result in roughly the same parameter counts across the three methods. To increase model capacity, we either increase the dimension or decrease the factor. This is what we refer to as "capacity" in the SHAP plots. We note that when the ratio between dimension and alpha is fixed, increasing model capacity has roughly the same effect as increasing learning rate or training epochs, though we expect the model's performance could now vary more greatly when varying other hyperparameters. We especially observe in Figure 2b that the effects could be reversed when alpha is set to $1$ (these checkpoints are not included in our SHAP analysis). Some qualitative comparisons are further provided in the bottom row of Figure 3.

## 6  Concluding Remarks

In conclusion, this paper serves three main purposes. First, we introduce LyCORIS, an open-source library implementing a diverse range of methods for Stable Diffusion fine-tuning. Second, we advocate for a more comprehensive evaluation framework that better captures the nuances of different fine-tuning methods. Lastly, our extensive experiments shed light on the impact of the choice of the algorithm and their configuration on model performance, revealing their relative strengths and limitations. For instance, based on our experiments, LoHa seems to be better suited for simple, multi-concept fine-tuning, whereas LoKr with full dimension is better for complex, single-concept tasks. This distinction in their applicability also indicates that the rank of the matrix update, often considered a key factor, may not always be the definitive predictor of a method's efficacy in various fine-tuning scenarios.

Despite our extensive efforts, the scope of our study remains limited. For example, we have not explored the task of generating images with multiple learned concepts, as this aspect is highly sensitive to input prompts and more challenging to evaluate. However, recent works such as Huang et al. (2023) aim to address these issues, and we believe that incorporating these emerging evaluation frameworks will further enrich the comparison of fine-tuning methods in future studies.

## Acknowledgement

The work of Z. Gao and Y. Gong was supported in part by the US National Science Foundation under grant CNS-2047761. We extend our heartfelt appreciation to the community for their invaluable support, which has been instrumental in bringing LyCORIS to fruition. Our particular gratitude goes to those who have directly contributed to the library and those who have further integrated it into the existing ecosystem, helping to improve and expand its functionality and reach.

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
