# Appendix

## Table of Contents

# A    Related Works

In this section, we review related works for two interconnected themes that are vital to our study: text-to-image model customization and text-to-image model evaluation.

**Model Customization.**    There has been a long-standing effort in adapting pretrained deep generative models to learn new concepts with limited data (Bau et al., 2019; Robb et al., 2020; Roich et al., 2022; Wang et al., 2018), but the surge of Stable Diffusion and similar large-scale text-to-image generation models have accelerated this progress. Specifically, DreamBooth (Ruiz et al., 2023) proposes fine-tuning the U-Net with a prior-preservation loss, which serves as the regularizer in combating overfitting and improving the generation performance. Another pioneering work in this direction is Textual Inversion (TI) (Gal et al., 2023), which instead optimizes the input text embedding vector with the subject images and uses that optimized text embedding for generation.

Expanding on the initial concept of pivotal tuning (Roich et al., 2022) for StyleGAN (Karras et al., 2019), other works have explored concurrent fine-tuning of both text embeddings and network architectures for diffusion models (Gu et al., 2023; Kawar et al., 2023; Kumari et al., 2023; Smith et al., 2023; Tewel et al., 2023). For instance, Imagic by Kawar et al. (2023) targets single-image editing by initially optimizing the text embedding for a given input before further network fine-tuning. On the other hand, Custom Diffusion (Kumari et al., 2023) and Perfusion (Tewel et al., 2023) both focus on fine-tuning only the K-V cross-attention layers to reduce overfitting but adopt different strategies. Custom Diffusion considers native fine-tuning of these layers and offers a way to merge fine-tuned models without additional training, while Perfusion employs a more complex gated mechanism and "locks" the K pathway using class-specific words, thereby enhancing the model's capacity to generate learned concepts across diverse contexts.

Efforts have also been made to address more specific challenges. For example, C-LoRA (Smith et al., 2023) addresses the issue of catastrophic forgetting through a self-regularization mechanism while fine-tuning the K-V cross-attention layers with LoRAs. In parallel, ED-LoRA (Gu et al., 2023) employs LoRA dropout to counterbalance the otherwise dominant influence of LoRA in the learning process, ensuring that the embeddings remain a significant component of concept learning.

Alongside these focused endeavors, there has been complementary progress in expanding and improving upon Textual Inversion. Notably, Voynov et al. (2023) introduced Extended Textual Inversion (XTI), a layer-wise embedding method that was also used in ED-LoRA. Another noteworthy innovation is DreamArtist (Dong et al., 2022), which leverages positive and negative embeddings for efficient one-shot model customization. Further improvements to the TI framework include the works by Alaluf et al. (2023); Han et al. (2023).

While TI-based methods often struggle to generate concepts outside of the pretrained models' domain (Gu et al., 2023; Smith et al., 2023), they do offer an advantage in terms of parameter efficiency. Specifically, the number of stored parameters required by these methods is orders of magnitude smaller than those necessitated by most network fine-tuning methods. This gap is partially bridged by SVDiff (Han et al., 2023) and Cones (Liu et al., 2023). The former optimizes singular values of the weight matrices of the network, which leads to a compact and efficient parameter space that reduces the risk of overfitting and language-drifting. The latter proposes to focus on concept neurons, a small set of parameters of the K-V attention layers, which are posited to be sufficient to encode the target concept in an ideal scenario. Remarkably, several methods implemented in our LyCORIS library, such as LoKr and $(IA)^3$ can also result in weight updates with parameter counts that are comparable to these methods.

In addition to the aforementioned methods, Qiu et al. (2023) proposed orthogonal fine-tuning (OFT) that offers another approach to retain the knowledge of pretrained models. OFT consists in fine-tuning a block diagonal orthogonal matrix that is multiplied with the original weight matrix. This has the unique advantage of preserving the hyperspherical energy of the weight matrices, which could be helpful in preserving the pretrained model's generative capabilities. We have also incorporated this method in the LyCORIS library as mentioned in Section 3.2.

A partial summary of the related works in text-to-image diffusion model customization is provided in Table 2. This table shows that even though these methods may focus on optimizing different components involved in the image generation process, they often introduce new techniques that can be integrated in a complementary fashion. For example, elements like the positive and negative

| Method | K-V | Linear | Conv | TE | Emb | PP Loss | Further Innovations |
|---|---|---|---|---|---|---|---|
| DreamBooth [67] | ✓ | ✓ | ✓ | | | ✓ | |
| Textual Inversion [18] | | | | | ✓ | | |
| DreamArtist [13] | | | | | ✓ | | Positive and Negative Embeddings |
| XTI [89] | | | | | ✓ | | Layer-Wise Embeddings |
| Custom Diffusion [41] | ✓ | | | | ✓ | ✓ | Model Fusion Technique |
| C-LoRA [77] | ✓ | | | | ✓ | ✓ | Self-Regularization for Continual Learning |
| Perfusion [82] | ✓ | | | | ✓ | | Key-Locking, Gated Rank-1 Update |
| Cones [47] | ✓ | | | | | ✓ | Concept Neurons |
| SVDiff [25] | ✓ | ✓ | ✓ | | | ✓ | Fine-Tune Singular Values, Cut-Mix-Unmix |
| ED-LoRA [21] | ✓ | - | | ✓ | ✓ | | XTI + LoRA Dropout, Gradient Fusion |
| OFT [60] | ✓ | - | | | | ✓ | Orthogonal Transformation |
| LyCORIS (ours) | ✓ | ✓ | ✓ | ✓ | | | LoHa, LoKr, etc. |

**Table 2:** A brief recapitulation for the functioning of different existing diffusion model fine-tuning strategies. K-V, Linear, Conv, TE, Emb, and PP Loss respectively stand for K-V cross-attention layers, linear layers in attention and feed-forward blocks in U-Net, convolutional layers, text encoder, embedding, and prior-preservation loss. We put a checkmark when the corresponding block is optimized or when the corresponding mechanism is employed. Both ED-LoRA and OFT fine-tune linear layers within attention blocks, and this is why we put - instead of a checkmark. For LyCORIS, we focus on the setup of this paper, but in reality, we have the liberty to decide which part to fine-tune and whether to use prior-preservation loss or not. Note for the "further innovations" column, we only include the main contributions from each work and further details concerning the implementation of each method are omitted (for example, Textual Inversion also considers an additional regularization term and progressive extensions, while Perfusion weights the diffusion reconstruction loss by a soft segmentation mask).

guiding introduced in DreamArtist can be effectively coupled with algorithms implemented in our LyCORIS library. Furthermore, other techniques, such as the layer-wise embeddings in XTI and key-locking from Perfusion, can also be incorporated concurrently. Collectively, these innovations thus form a robust toolkit conducive to the fine-tuning of text-to-image diffusion models.

Finally, there exists a distinct line of research that aims to facilitate test-time adaptation without the need for further fine-tuning (see e.g., Gal et al., 2023; Ma et al., 2023; Ruiz et al., 2023; Shi et al., 2023; Wei et al., 2023). In these cases, we need to train separate networks or additional modules, using larger and more diverse datasets that encompass data from a single domain or multiple domains. As a result, the adaptability of these models is generally confined to the domains encountered during training. While such methods are useful for specialized, ad-hoc applications where users might want to generate variations of a target concept with a limited set of input images, they don't offer the broad applicability inherent to direct fine-tuning strategies.

**Model Evaluation.** The evaluation of generative models has been a central concern since the inception of these models. Early efforts focused on measuring the distance between the data distribution and the distribution learned by the models, utilizing metrics like FID (Heusel et al., 2017), KID (Bińkowski et al., 2018), and Precision-Recall (Sajjadi et al., 2018) which have been widely

adopted in the community. Meanwhile, the limitations of these metrics are increasingly being recognized, as detailed by Stein et al. (2023), who critically examined the flaws of various metrics used for evaluating generative models. In our context, these metrics are primarily useful for evaluating the concept fidelity of the generated images on the condition that we have a sufficiently large training set.

Another dimension comes into the scene when considering text-to-image models. In such scenarios, assessing the text-image alignment of generated images becomes crucial. Commonly used metrics for this purpose include CLIPScore (Hessel et al., 2021), R-precision (Xu et al., 2018), and BLEU (Papineni et al., 2002) or CIDEr (Vedantam et al., 2015) for evaluating the similarity between the captions generated for the synthesized images and the original text prompts. To enhance interpretability and enable finer-grained evaluation, recent works like TIFA (Hu et al., 2023) and X-IQE (Chen et al., 2023) have also explored the use of pretrained large vision-language models such as mPLUG (Li et al., 2022) and MiniGPT-4 (Zhu et al., 2023). Together with image fidelity/quality, these represent the two main aspects on which text-to-image models are typically evaluated (Dinh et al., 2022; Saharia et al., 2022). Notably, nearly all the text-to-image model customization studies we have discussed have limited their quantitative evaluations to these two sets of metrics.

Yet, the need for going beyond these two aspects has also been acknowledged. Specifically, other criteria such as visual reasoning, social bias, and creativity have been considered in DALL-EVAL (Cho et al., 2022) and HRS-Bench (Bakr et al., 2023). While some of these can be regarded as the design of more dedicated metrics to measure text-image alignment in particular situations, others necessitate a completely different attack angle.

Importantly, all the previous works focus exclusively on the evaluation of general text-to-image models, while we zoom in on the evaluation of fine-tuned models. Although we can borrow metrics and criteria from these works, certain nuances exist. For example, as discussed in Section 4, we may want to ensure that new concepts do not adversely affect the innate knowledge of the original model, and image fidelity, image quality, and aesthetics should be evaluated independently (Silverstein & Farrell, 1996). Moreover, we separate how the model is prompted (or how it is used more generally) from how the generated images are evaluated, which together define the so-called "skills" in the aforementioned papers. On top of this, we identify a few key aspects that could be the most affected by model fine-tuning. That said, the metrics that we currently employ remain relatively rudimentary, and integrating more advanced metrics from the literature would undoubtedly be beneficial.

Complementary to automatic evaluation, human evaluation is generally considered to yield the most accurate assessments of model performance. Recent initiatives in this area include the works of Otani et al. (2023); Petsiuk et al. (2022). In particular, Otani et al. (2023) introduced a well-defined protocol for human evaluation to ensure verifiable and reproducible results. Nonetheless, most studies employing human evaluations compare a limited set of models using relatively few generated samples and evaluation criteria. Given the scale of our work, which involves hundreds of trained checkpoints and millions of generated images, conducting a comprehensive human evaluation would incur prohibitive costs, both in terms of time and resources. Therefore, we have opted to omit human evaluations from this study. In the meantime, we believe that methods from the fields of multi-armed bandits and Bayesian optimization may offer promising avenues for tackling the scalability challenges of human evaluation (Moss et al., 2019; Turner et al., 2021).

As another avenue for exploration, several datasets have been introduced to capture human preferences for AI-generated images (Kirstain et al., 2023; Wu et al., 2023; Xu et al., 2023). These datasets pave the way for training human preference models, which could further be used for evaluation and for fine-tuning text-to-image models for closer alignment with human preferences (Xu et al., 2023). However, one should note that human preference represents just one facet in a multi-dimensional evaluation landscape and is often more tied to the dataset than the training algorithm itself. Moreover, as Casper et al. (2023) aptly points out, "A single reward function cannot represent a diverse society of humans". Over-reliance on such a reward function could inadvertently marginalize or overlook the preferences and needs of under-represented groups, thus reinforcing existing biases and inequalities.

# B  Further Discussion on Implemented Algorithms

This appendix discusses finer details of the algorithms implemented in our study. We especially focus on the the effect of the merge ratio on training dynamics, Tucker decomposition of convolutional layers, and the interpretation of LoKr as a number of consecutive linear layers.

## B.1  Effect of Merge Ratio

In this section, we demonstrate an equivalence between scaling the merge ratio and scaling the initialization parameters and the learning rates. For simplicity, we will write $\mathbf{h}' = W_0\mathbf{h} + \mathbf{b}$ to represent the transformation performed by all types of layers in the neural network, whatever we deal with convolutional or linear layers. Our result holds in a general setup in which each layer, identified by the index $\ell$, can use its own distinct decomposition function $T_\ell$ and merge ratio $\gamma_\ell$.

Formally, we assume that the forward pass of layer $\ell$ is modified to

$$\mathbf{h}'_\ell = W_{\ell,0}\mathbf{h}_\ell + \mathbf{b}_\ell + \gamma_\ell \Delta W_\ell \mathbf{h}_\ell = W_{\ell,0}\mathbf{h}_\ell + \mathbf{b}_\ell + \gamma_\ell T_\ell(A_{\ell,1}, \ldots, A_{\ell,m_\ell})\mathbf{h}_\ell, \tag{6}$$

where $A_{\ell,1}, \ldots, A_{\ell,m_\ell}$ are a set of tensors that together define the weight update $\Delta W_\ell$. In LyCORIS, the decomposition function $T_\ell$ is mainly composed of low rank decomposition, Hadarmard decomposition, and Kronecker decomposition. However, other decomposition functions such as Tucker decomposition (Tucker, 1966) can also be used (see Appendix B.2.2). The effect of the merge ratios $\boldsymbol{\gamma} = (\gamma_\ell)_\ell$ is stated in the following theorem.

**Theorem 1.** *Assume that we train a neural network with forward pass modified as in* (6) *and that every $T_\ell$ is homogeneous, i.e., for all $a \in \mathbb{R}$ and all possible input $A_{\ell,1}, \ldots, A_{\ell,m_\ell}$, we have*

$$T_\ell(aA_{\ell,1}, \ldots, aA_{\ell,m_\ell}) = a^{m_\ell}T_\ell(A_{\ell,1}, \ldots, A_{\ell,m_\ell}). \tag{7}$$

*Then, replacing $\gamma_\ell$ by $1$ in* (6)*, scaling the initialization parameters and learning rate of each layer $\ell$ respectively by $(\gamma_\ell)^{\frac{1}{m_\ell}}$ and $(\gamma_\ell)^{\frac{c}{m_\ell}}$ is mathematically equivalent to training with the original initialization parameters, learning rates, and merge ratios. Here, $c = 2$ if we train with stochastic gradient descent (SGD), and $c = 1$ if we train with Adam, RMSProp, or AdaGrad with the $\varepsilon$ parameter set to $0$.*

*Proof.* Let $\mathcal{L}$ be the loss function for a specific step when the network is parameterized with $(\Delta W_\ell)_\ell$. Note that this loss function is different from the one defined in Section 2.1, and in particular, it depends on the data sampled at each step, and also the sampled noise and sampled diffusion step in the case of diffusion model. Similarly, for any $\boldsymbol{\gamma} = (\gamma_\ell)_\ell$, we define $\tilde{\mathcal{L}}^\gamma$ as the loss function of the same step (i.e., resulting from the same sampled data, noise, diffusion step etc.) but with respect to the tensors $\mathbf{A} = (A_{\ell,i})_{\ell,i}$ when the forward pass is modified following (6). By defining

$$\mathbf{T}^\gamma: \mathbf{A} \to (\Delta W_\ell)_\ell = (\gamma_\ell T_\ell(A_{\ell,1}, \ldots, A_{\ell,m_\ell}))_\ell, \tag{8}$$

as the function that maps the decomposed tensors to weight updates, we have clearly $\tilde{\mathcal{L}}^\gamma = \mathcal{L} \circ \mathbf{T}^\gamma$. To simplify the notation, we further write $\tilde{\mathcal{L}} = \tilde{\mathcal{L}}^1$ and $\mathbf{T} = \mathbf{T}^1$. We claim that

$$\nabla_{\mathbf{A}_\ell} \tilde{\mathcal{L}}^\gamma(\mathbf{A}) = (\gamma_\ell)^{\frac{1}{m_\ell}} \nabla_{\mathbf{A}_\ell} \tilde{\mathcal{L}}(\boldsymbol{\gamma}^{\frac{1}{m}} \cdot \mathbf{A}). \tag{9}$$

In the above, $\mathbf{A}_\ell = (A_{\ell,i})_i$ collects the decomposed tensors of layer $\ell$, $\nabla_{\mathbf{A}_\ell}$ represents the gradient with respect to these tensors, and $\boldsymbol{\gamma}^{\frac{1}{m}} \cdot \mathbf{A} = ((\gamma_\ell)^{\frac{1}{m_\ell}} \mathbf{A}_\ell)_\ell$ is obtained from scaling all the tensors by a layer-dependent scalar $(\gamma_\ell)^{\frac{1}{m_\ell}}$. For ease of mathematical treatment, it is convenient to consider both the inputs and outputs of functions $\mathcal{L}, \tilde{\mathcal{L}}^\gamma$, and $\mathbf{T}^\gamma$ as one-dimensional vectors, which are formed by flattening the tensors and then concatenating them. Lastly, by slight abuse of notation, we use $\mathbf{A}_\ell$ for both the variable and the value at which we evaluate the gradient.

To prove (9), we first apply chain rule to get

$$\begin{aligned}
\nabla \tilde{\mathcal{L}}^\gamma(\mathbf{A})^\top &= \nabla \mathcal{L}(\mathbf{T}^\gamma(\mathbf{A}))^\top \operatorname{Jac}_{\mathbf{T}^\gamma}(\mathbf{A}), \\
\nabla \tilde{\mathcal{L}}(\boldsymbol{\gamma}^{\frac{1}{m}} \cdot \mathbf{A})^\top &= \nabla \mathcal{L}(\mathbf{T}(\boldsymbol{\gamma}^{\frac{1}{m}} \cdot \mathbf{A}))^\top \operatorname{Jac}_{\mathbf{T}}(\boldsymbol{\gamma}^{\frac{1}{m}} \cdot \mathbf{A}),
\end{aligned} \tag{10}$$

where $\operatorname{Jac}_T$ represents the Jacobian matrix of operator $T$. Note that by the definition of $\mathbf{T}^\gamma$, input variable $A_{\ell,i}$ only affects output variable $\Delta W_\ell$, and thus $\operatorname{Jac}_{\mathbf{T}^\gamma}(\mathbf{A})$ is blockwise diagonal. This

indicates that (10) can be written in a layer-wise way as following.

$$\nabla_{\mathbf{A}_\ell} \tilde{\mathcal{L}}^{\gamma}(\mathbf{A})^{\top} = \nabla_{\Delta W_\ell} \mathcal{L}(\mathbf{T}^{\gamma}(\mathbf{A}))^{\top} \operatorname{Jac}_{\gamma_\ell T_\ell}(\mathbf{A}_\ell),$$
$$\nabla_{\mathbf{A}_\ell} \tilde{\mathcal{L}}(\boldsymbol{\gamma}^{\frac{1}{\mathbf{m}}} \cdot \mathbf{A})^{\top} = \nabla_{\Delta W_\ell} \mathcal{L}(\mathbf{T}(\boldsymbol{\gamma}^{\frac{1}{\mathbf{m}}} \cdot \mathbf{A}))^{\top} \operatorname{Jac}_{T_\ell}((\gamma_\ell)^{\frac{1}{m_\ell}} \mathbf{A}_\ell). \tag{11}$$

Since each $T_\ell$ is homogeneous, it holds that

$$\gamma_\ell T_\ell(\mathbf{A}_\ell) = T_\ell((\gamma_\ell)^{\frac{1}{m_\ell}} \mathbf{A}_\ell). \tag{12}$$

Differentiating both sides with respect to $\mathbf{A}_\ell$ gives immediately

$$\operatorname{Jac}_{\gamma_\ell T_\ell}(\mathbf{A}_\ell) = (\gamma_\ell)^{\frac{1}{m_\ell}} \operatorname{Jac}_{T_\ell}((\gamma_\ell)^{\frac{1}{m_\ell}} \mathbf{A}_\ell). \tag{13}$$

On the other hand, it also follows from (12) that

$$\mathbf{T}^{\gamma}(\mathbf{A}) = \mathbf{T}(\boldsymbol{\gamma}^{\frac{1}{\mathbf{m}}} \cdot \mathbf{A}). \tag{14}$$

Plugging (13) and (14) into (11) gives immediately (9).

To complete the proof, we just need to note that an important difference between AdaGrad-type methods and vanilla SGD is in whether the learning rates are scaled by some scalar computed based on gradient magnitude or not. This is not the case for SGD, and we can simply write

$$(\gamma_\ell)^{\frac{1}{m_\ell}} (\mathbf{A}_\ell - \eta_\ell \nabla_{\mathbf{A}_\ell} \tilde{\mathcal{L}}^{\gamma}(\mathbf{A})) = (\gamma_\ell)^{\frac{1}{m_\ell}} \mathbf{A}_\ell - \eta_\ell (\gamma_\ell)^{\frac{2}{m_\ell}} \nabla_{\mathbf{A}_\ell} \tilde{\mathcal{L}}(\boldsymbol{\gamma}^{\frac{1}{\mathbf{m}}} \cdot \mathbf{A}), \tag{15}$$

where $\eta_\ell$ is the learning rate of layer $\ell$. This shows that if for each layer $\ell$, we scale the initialized parameters by $(\gamma_\ell)^{\frac{1}{m_\ell}}$ and learning rate by $(\gamma_\ell)^{\frac{2}{m_\ell}}$, while setting the merge ratio to 1, then after each stochastic gradient step we still have $\mathbf{A}^2 = \boldsymbol{\gamma}^{\frac{1}{\mathbf{m}}} \cdot \mathbf{A}^1$. Here, $\mathbf{A}^1$ and $\mathbf{A}^2$ are respectively the tensors obtained from the updates with merge ratio $\boldsymbol{\gamma}$ and merge ratio $\mathbf{1}$ but with scaled initialization and learning rates. Together with (14) we then see that the two approaches lead to the same weight update $(\Delta W_\ell)_\ell$ at the end.

As for Adam, RMSProp, and AdaGrad, when $\varepsilon$ is set to 0, the scaling of the learning rate causes the two versions to have the same scaled update vector. In other words, instead of having a relation like (9), the update is performed with the same vector $\mathbf{d}$. One has clearly

$$(\gamma_\ell)^{\frac{1}{m_\ell}} (\mathbf{A}_\ell - \eta_\ell \mathbf{d}) = (\gamma_\ell)^{\frac{1}{m_\ell}} \mathbf{A}_\ell - \eta_\ell (\gamma_\ell)^{\frac{1}{m_\ell}} \mathbf{d}. \tag{16}$$

This shows that one should rather scale the learning rate by $(\gamma_\ell)^{\frac{1}{m_\ell}}$ in this case to maintain the relation $\mathbf{A}^2 = \boldsymbol{\gamma}^{\frac{1}{\mathbf{m}}} \cdot \mathbf{A}^1$, concluding the proof. $\qquad\square$

While Theorem 1 provides an intuitive way to understand how the merge ratio $\gamma$, or the related $\alpha$ affects training, it is crucial to keep in mind that these quantities were specifically introduced by Hu et al. (2021) to address numerical precision issues. In fact, if we were to simply scale the initialization parameters and learning rates instead of using the merge ratios, the stored parameters in the decomposed tensors would be much smaller. This could lead to numerical instability or reduced precision during the optimization process, potentially affecting the model's training and final performance adversely.

## B.2 DECOMPOSITION OF CONVOLUTIONAL LAYERS

In this section, we delve into the application of matrix decomposition techniques for convolutional layers, with a focus on two distinct approaches implemented in the LyCORIS library.

### B.2.1 THE STANDARD APPROACH

Consider a convolutional layer with a weight update denoted as $\Delta W \in \mathbb{R}^{c_{\text{out}} \times c_{\text{in}} \times k \times k}$, where $k$ represents kernel size, $c_{\text{in}}$ and $c_{\text{out}}$ indicate the number of input channels and output channels. To facilitate the application of our method, this weight update can be unrolled into a 2-D matrix represented as $\Delta W \in \mathbb{R}^{c_{\text{out}} \times c_{\text{in}} k^2}$. Factorizing this 2-D matrix with LoRA gives us two matrices of reduced rank: $B \in \mathbb{R}^{c_{\text{out}} \times r}$, $A \in \mathbb{R}^{r \times c_{\text{in}} k^2}$. The matrix $A$ can be reshaped back to a 4-D tensor: $A \in \mathbb{R}^{r \times c_{\text{in}} \times k \times k}$. Such a transformation implies that the given convolutional layer can be effectively

approximated by two consecutive convolutional layers with kernel sizes $k$ and 1. Notably, the low-rank dimension $r$ is the number of output and input channels of the first and second layers. This decomposition method is well-established and has been extensively adopted in previous research (see e.g., Wang et al., 2021). Adapting this approach to other factorization methods like LoHa and LoKr is straightforward.

### B.2.2 TUCKER DECOMPOSITION

Besides the standard approach that we just dedscribed Tucker decomposition (Tucker, 1966) can also be applied to the convolutional layers to achieve higher computational and memory efficiency.

To explain this, let us denote by $\times_n$ the $n$-mode product which computes the matrix product on dimension $n$ of a tensor while casting over the remaining dimensions. Formally, for $G = (g_{i_1 \dots i_d}) \in \mathbb{R}^{I_1 \times \dots \times I_d}$ a tensor of order $d$ and $A = (a_{i_n j_n}) \in \mathbb{R}^{I_n \times J_n}$ a matrix of size $I_n \times J_n$, their $n$-mode product $G \times_n A \in \mathbb{R}^{I_1 \times \dots \times I_{n-1} \times J_n \times I_{n+1} \times \dots I_d}$ has its elements given by

$$(G \times_n A)_{i_1 \dots i_{n-1} j_n i_{n+1} \dots i_d} = \sum_{i_n=1}^{I_n} g_{i_1 \dots i_{n-1} i_n i_{n+1} \dots i_d} a_{i_n j_n}. \tag{17}$$

With this in mind, Tucker decomposition is simply the decomposition of a tensor into a set of matrices and a small core tensor using $n$-mode product.

In the context of fine-tuning convolutional layers, we can apply Tucker decomposition to decompose the weight update $\Delta W \in \mathbb{R}^{c_{\text{out}} \times c_{\text{in}} \times k \times k}$ into one core tensor $G \in \mathbb{R}^{r \times r \times k \times k}$ and two matrices $B \in \mathbb{R}^{r \times c_{\text{out}}}$, $A \in \mathbb{R}^{r \times c_{\text{in}}}$, leading to

$$\mathbf{h}' = W_0 * \mathbf{h} + \mathbf{b} + \gamma(G \times_1 B \times_2 A) * \mathbf{h}. \tag{18}$$

Compared to the standard approach elaborated in Appendix B.2.1, the number of parameters changes from $r(c_{\text{in}}k^2 + c_{\text{out}})$ to $r(rk^2 + c_{\text{in}} + c_{\text{out}})$. The latter is substantially smaller when $r \ll c_{\text{in}}(1 - 1/k^2)$.

Importantly, the tensor $G$ can also be interpreted as a convolutional kernel with in-channels and out-channels both set to $r$. To recast this decomposition as three convolutional layers (two of them have $k = 1$), we first reshape $B^\top$ and $A$ respectively into $\tilde{B} \in \mathbb{R}^{c_{\text{out}} \times r \times 1 \times 1}$ and $\tilde{A} \in \mathbb{R}^{r \times c_{\text{in}} \times 1 \times 1}$. The forward pass then becomes

$$\mathbf{h}' = W_0 * \mathbf{h} + \mathbf{b} + \tilde{B} * (G * (\tilde{A} * \mathbf{h})). \tag{19}$$

Tucker decomposition was also adopted in FedPara (Hyeon-Woo et al., 2022) for decomposing convolutional layers' kernels. We borrow their methods and implement it for LoHa. In this case, the forward pass is modified to

$$\mathbf{h}' = W_0 * \mathbf{h} + \mathbf{b} + \gamma(G_1 \times_1 B_1 \times_2 A_1) \odot (G_2 \times_1 B_2 \times_2 A_2) * \mathbf{h}, \tag{20}$$

where $G_1, G_2 \in \mathbb{R}^{r \times r \times k \times k}$, $B_1, B_2 \in \mathbb{R}^{r \times c_{\text{out}}}$, and $A_1, A_2 \in \mathbb{R}^{r \times c_{\text{in}}}$. The decomposition for convolutional layers in LoKr follows the same methodology.

### B.3 LOKR AS CONSECUTIVE LINEAR LAYERS

For a more intuitive understanding of LoKr, we introduce here a unique representation for $\Delta \mathbf{h} = (C \otimes BA)\mathbf{h}$ that effectively models it as a sequence three linear layers. The core mechanism for this representation is the use of the mixed Kronecker matrix-vector product property. Consider the linear transformation $\Delta \mathbf{h} = \Delta W \mathbf{h}$, where $\mathbf{h} \in \mathbb{R}^q$, $\Delta \mathbf{h} \in \mathbb{R}^p$ and $\Delta W \in \mathbb{R}^{p \times q}$. The weight update $\Delta W$ is further decomposed as $\Delta W = C \otimes BA$ with $C \in \mathbb{R}^{u_p \times u_q}$, $B \in \mathbb{R}^{v_p \times r}$, and $A \in \mathbb{R}^{r \times v_q}$. Utilizing the mixed Kronecker matrix-vector product property, we can express this as

$$\begin{aligned}
\Delta \mathbf{h} &= \Delta W \mathbf{h} \\
&= (C \otimes BA)\mathbf{h} \\
&= \text{vec}(C \, \text{unvec}(\mathbf{h})(BA)^\top) \\
&= \text{vec}(C((BA \, \text{unvec}^\top(\mathbf{h}))^\top)^\top).
\end{aligned} \tag{21}$$

Here, vec represents the row-major vectorization operator that stacks the rows of a matrix into a vector, and unvec reshapes the vector $\mathbf{h}$ from $\mathbf{h} \in \mathbb{R}^q$ to $\text{unvec}(\mathbf{h}) \in \mathbb{R}^{u_q \times v_q}$ by filling rows of

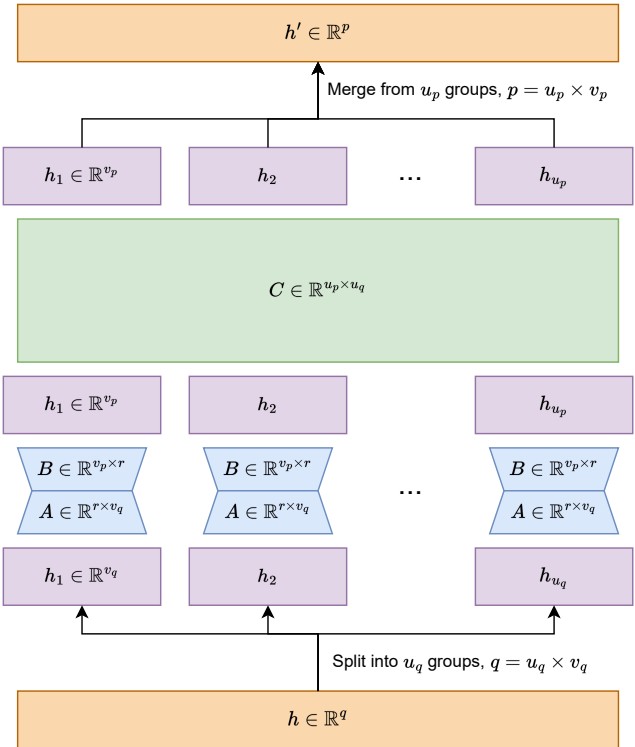

**Figure 4:** This figure shows how to represent LoKr as two or three linear layers (depending on whether we perform the additional low-rank decomposition or not).

---

**Algorithm 1:** PyTorch-style pseudocode for LoKr as linear layers

---

```
 1:  def lokr_linear(h, a, b, c):
 2:      vq = a.size(1)
 3:      uq = c.size(1)
 4:      h_in_group = rearrange(
 5:          h,
 6:          "b ... (uq vq) -> b ... uq vq",
 7:          uq=uq, vq=vq
 8:      )
 9:      ha = F.linear(h_in_group, a)
10:      hb = F.linear(ha, b)
11:      h_cross_group = hb.transpose(-1, -2)
12:      hc = F.linear(h_cross_group, c)
13:      h = rearrange(hc, "b ... vp up -> b ... (up vp)")
14:      return h
```

---

$\mathrm{unvec}(\mathbf{h})$ with consecutive elements from $\mathbf{h}$. The $\mathrm{unvec}^\top$ notation indicates an additional transpose operation following $\mathrm{unvec}(\mathbf{h})$. As illustrated in Figure 4 , this representation enables us to interpret LoKr as a composition of three consecutive linear layers. In Algorithm 1, we further provide a PyTorch-style pseudo-code that implements the decomposition of (21).

## C  Challenges in Algorithm Evaluation

In this appendix, we highlight a number of difficulties in evaluating the performance of fine-tuning algorithms. These complexities caution against simplistic comparisons and underscore the importance of a more comprehensive evaluation framework, as what we proposed in Section 4.

**Sensitivity to Hyperparameters.** As we can see from Figure 2, the algorithms' performance are sensitive to various hyperparameters, such as learning rate, dimension, and factor. Without a single objective metric, pinpointing the optimal hyperparameters for a specific method becomes elusive. As a result, comparing two methods based on a single set of hyperparameters oversimplifies the evaluation process.

**Performance Discrepancy Across Concepts.** The models' performances can differ substantially across different concepts, whether these are trained in isolation or in tandem. This variance is especially pronounced when comparing concepts that are fundamentally different or with differing numbers of training images. Nonetheless, even when these factors are mitigated, discrepancies in performance across concepts can still arise, as illustrated in Appendix G.1.

**Influence of Dataset.** In a similar vein, the composition of the dataset, including the images and accompanying captions, exerts a significant influence on the performance of the fine-tuning model (see Appendix H.3 for an illustration on the importance of having good captions). Additionally, the nuances of each dataset may necessitate tailored approaches or specific configurations to achieve optimal results. For instance, a larger training set or a dataset featuring concepts that diverge substantially from the model's pretrained knowledge may benefit from employing a model with greater capacity. Thus, it is essential to adapt fine-tuning strategies to the particularities of the dataset at hand.

**Conflicting Criteria.** There is an intrinsic trade-off between the various criteria under consideration. We have particularly seen in Section 5.3 that models with higher concept fidelity often have lower controllability, diversity, and base model preservation. Determining the optimal balance among these criteria requires a nuanced, case-by-case analysis rather than a simple aggregation of metrics.

**Unreliability of Evaluation Metrics.** Despite the significant advancements in computer vision and deep learning in recent years, the metrics we employ are still far from perfect. Often, these metrics do not fully align with human judgement and may overlook nuanced details that define a concept, as shown by Thrush et al. (2022); Yuksekgonul et al. (2023) and we further elaborate in Appendix G.2. Additionally, a single numerical value can be insufficiently informative; for instance, a low image similarity score could arise from either underfitting or overfitting.

**Practical Considerations.** In practice, a trained model often undergoes further adjustments. As explained in Section 3.1, we can adjust the scaling factor of the fine-tuned weight differences $\Delta W$, combine multiple trained networks, and apply the fine-tuned parameters to a different base model. This is not to mention the influence of prompts and the potential for prompt engineering. All these variables add layers of complexity to the evaluations.

In light of the aforementioned complexities, it becomes evident that assessing the effectiveness of fine-tuning algorithms is a multifaceted challenge. While existing research often leans on a limited scope of evaluation metrics, hyperparameters, and concept categories, such an approach risks not capturing the full breadth and depth of these algorithms' capabilities and inadvertently results in evaluations that unfairly penalize certain methods. Therefore, we call upon the research community to build upon our initial efforts by investing in the development of more extensive evaluation frameworks and ecosystems for fine-tuned text-to-image models. These should aim to delve deeper into the subtleties of various fine-tuning strategies, thereby promoting more robust and meaningful comparisons.

# D   Experimental Details

This part of the appendix offers detailed information about the experiments we conducted, ranging from dataset specifics, algorithm settings, to details on the evaluation of the models.

## D.1   DATASET

In this section, we provide more details on the composition of our dataset, the sources of the data, and the captioning strategy.

**Dataset Structure.**   Our dataset follows a hierarchical structure, as detailed in Tables 3 and 4. We also include the count of images for each category, class, and subclass, as well as the number of repeats for images in each class (used for dataset balancing) in these tables.

| Category | Class | Sub-class | # of Images | Repeat |
|---|---|---|---|---|
| Anime Characters (571 images) | Yuuki Makoto | N/A | 90 | 2 |
| | Kotomine Kirei | | 51 | 4 |
| | Tushima Yoshiko | | 128 | 2 |
| | Ika Musume | Default outfit | 118 | 1 |
| | | Alternative outfit | 20 | |
| | | Others | 34 | |
| | Abukuma (KanColle) | Dark color uniform | 54 | 2 |
| | | Light color uniform | 48 | |
| | | Others | 28 | |
| Movie Characters (276 images) | K-2SO | N/A | 63 | 3 |
| | Admiral Piett | Figurine | 13 | 4 |
| | | Realistic | 34 | |
| | | Others | 2 | |
| | Bodhi Rook | Figurine | 11 | 3 |
| | | Illustration | 12 | |
| | | Realistic | 34 | |
| | | Others | 2 | |
| | Saw Gerrera | Afro, illustration | 4 | 4 |
| | | Afro, realistic | 23 | |
| | | Bald, 3d | 10 | |
| | | Bald, realistic | 7 | |
| | | Others | 1 | |
| | Rose Tico | Illustration | 10 | 3 |
| | | Realistic | 45 | |
| | | Others | 5 | |
| Styles (776 images) | Ghibli | N/A | 200 | 1 |
| | Ghibli 2 | | 100 | 2 |
| | Old Book | | 100 | 2 |
| | Ukiyo E | | 87 | 2 |
| | Impressionism | | 101 | 2 |
| | Felix Vallotton | | 100 | 2 |
| | Vladimir Borovikovsky | | 88 | 2 |

**Table 3:** Summary of dataset composition—anime characters, movie characters, and styles.

| Category | Class | Subclass | # of Images | Repeat |
|---|---|---|---|---|
| Stuffed Toys (53 images) | Tortoise | N/A | 12 | 17 |
| | Pink | | 10 | 20 |
| | Panda | | 10 | 20 |
| | Bunny | | 7 | 29 |
| | Lobster | | 7 | 29 |
| | Teddy Bear | | 7 | 29 |
| Scenes (30 images) | Waterfall | N/A | 9 | 22 |
| | Garden | | 7 | 29 |
| | Canal | | 5 | 40 |
| | Castle | | 5 | 40 |
| | Sculpture | | 4 | 50 |

**Table 4:** Summary of dataset composition—stuffed toys and scenes.

**Data Sources.** The images in our dataset originate from various sources to encompass a broad range of styles and subjects.

- Images for anime characters are sourced from the DAF:re dataset (Rios et al., 2021).
- Movie character images are extracted from a public dataset available on Kaggle (Young, 2019).
- Scene and stuffed toy images are part of the CustomConcept101 dataset (Kumari et al., 2023).
- Style-related images are compiled from multiple sources, including the "new" WikiArt dataset (Tan et al., 2019), the Old Book Illustrations dataset (gigant, 2007), and Studio Ghibli's official website[4]. For the images obtained from Studio Ghibli's website, it is important to note that Studio Ghibli specifies they should be used "with common sense" and are not intended for commercial use.[5]

**Captioning.** For captioning of our dataset, we employ a tagger with a ConvNeXt V2 architecture (Woo et al., 2023), hosted on Hugging Face.[6] This tagger is trained on the Danbooru dataset (Anonymous et al., 2022). We set the threshold to $0.35$ for our use. After the initial tagging phase, we manually adjust the tags for all the images within the categories "scenes" and "stuffed toys". As for images of other categories, we filter out tags that are naturally bound to the target concept, such as tags that represent hair color or gender. We also remove tags that are related to the corresponding outfit for character outfit sub-classes.

Regarding class words and trigger words, we use unique tokens for each class and outfit sub-class, leading to a total of 32 different tokens. The class word is set to one of the following: anime girl, anime boy, robot, man, woman, scene, stuffed toy, or style. Sub-class keywords are used as is for movie characters and otherwise formed by concatenating the relevant token with the word "outfit" for the two anime characters in question. We do not use sub-class keywords for the sub-class "others". The final prompt consists of a string where the concept descriptor, sub-class keyword, and tags are separated by commas. For illustration, an example is provided below:

```
"[V_abukuma] anime girl, [V_dark uniform] outfit, looking at viewer,
smile, simple background, full body, boots, black background"
```

**A Note on Prior-Preservation Loss.** The prior-preservation loss is frequently used in the literature to enhance the model's performance for generalization and concept-agnostic prompts, see e.g., Han et al. (2023); Kumari et al. (2023); Ruiz et al. (2023). Specifically, this approach has proven effective in mitigating unwanted concept drift and enriching the diversity of the generated images. Despite

---

[4]https://www.ghibli.jp/works/ (Accessed: 2023-07-16)
[5]Studio Ghibli's original note in Japanese: ※画像は常識の範囲でご自由にお使いください。
[6]https://huggingface.co/SmilingWolf/wd-v1-4-convnext-tagger-v2 (Accessed: April 22, 2023)

these merits, we have opted not to incorporate the prior-preservation loss in our experiments for two important reasons.

First, implementing this loss function necessitates the creation of a separate regularization dataset. This can be a labor-intensive process, particularly in our case where the training set encompasses a broad range of concept categories. While one could always build a regularization set using the pretrained model, there remains the challenge of designing the prompts for generation and ensuring the diversity and quality of the generated samples. Furthermore, some users would prefer the fine-tuned model to produce samples distinct from those produced by the pretrained model.

Secondly, it is our hope that by focusing on this more challenging setup, we can better identify the impacts of various algorithmic configurations and settings. In fact, with the prior-preservation loss the model learns from both the training set and the regularization set, making the analysis of the methods even more complicated.

### D.2 ALGORITHM CONFIGURATION AND HARDWARE

This section details the configurations employed for training and image generation, as well as the hardware specifications utilized in our experiments. The configuration files for fine-tuning can also be found at `https://github.com/cyber-meow/LyCORIS-evaluation/tree/main/exp_configs/training_configs`.

**Base Model.** As explained in Section 5, we perform fine-tuning on top of the non-EMA version of Stable Diffusion 1.5, which can be accessed at `https://huggingface.co/runwayml/stable-diffusion-v1-5`.

**Shared Hyperparameters.** The following hyperparameters are used throughout our experiments.

- **Optimizer:** We use 8-bit AdamW (Dettmers et al., 2022; Loshchilov & Hutter, 2019) with weight decay 0.1, $\beta_1 = 0.9$, $\beta_2 = 0.99$, and a constant scheduler with 5 epochs of warm-up. Gradient clipping is also applied, with a maximum gradient norm set to 1.

- **Data:** We load the data using aspect ratio bucketing with resolution set to 512. This ensures that each image is resized to maintain its original aspect ratio as much as possible, while the new height and width must be multiples of 64, and the new area must not exceed $512 \times 512$. Batch size is set at 8, and a caption-dropping rate of 5% is applied, meaning that empty captions are used in the loss calculation with a 5% probability.

- **Image generation:** We use 25 steps of DDIM sampler (Song et al., 2021) with a CFG scale of 7. All the generated images are of size $512 \times 512$. No negative prompt is used.

**Configuration-Dependent Hyperparameters.** By default, we train all the linear layers of the text encoder and U-Net, with a learning rate of $10^{-6}$ for native fine-tuning, and a learning rate of $5 \cdot 10^{-4}$ for LoRA, LoHa and LoKr. The default dimension and alpha for LoRA and LoHa are respectively $(8, 4)$ and $(4, 2)$. The default factor for LoKr is 8. Note that as explained in Section 3.2, it is also possible to specify dimension for LoKr in LyCORIS, and this leads to a low rank decomposition of the second block obtained from the Kronecker product decomposition. To avoid this behavior, we need to set the dimension to a sufficiently large number. The hyperparameters for the remaining experiments where we individually vary 1 to 2 hyperparameters are as specified in Section 5.3.2. The same set of hyperparameters is applied to all the trained layers of the network. The resulting file sizes, average training time, and approximate VRAM usage for each network configuration that we consider are provided in Table 5.

**Remark 1.** The VRAM usage and training efficiency of the methods are influenced by various factors, including their implementation and the underlying hardware specifics. Therefore, what we provide here should just be treated as a reference and may not reflect the efficiency of these methods in the latest version of the library.

Interestingly, LoRA training takes less time compared to LoHa and LoKr when convolutional layers are involved, which is not the case when only linear layers are trained. To understand this, we distinguish between two different ways to implement LoRA.

| Algorithm | Trained Layers | Dimension | Factor | File Size | Time (hr) | VRAM (G) |
|-----------|----------------|-----------|--------|-----------|-----------|----------|
| LoRA | Attention | 8 | N/A | 4.4 M | 4.2 | 13.6 |
| | Linear | 8 | | 9.2 M | 4.9 | 16.2 |
| | Full | 8 | | 20 M | 5.6 | 15.5 |
| | Linear | 32 | | 37 M | 5 | 16.5 |
| LoHa | Attention | 4 | N/A | 4.4 M | 4.3 | 14.5 |
| | Linear | 4 | | 9.3 M | 4.9 | 15.2 |
| | Full | 4 | | 20 M | 6.4 | 18.1 |
| | Linear | 16 | | 37 M | 5 | 15.7 |
| LoKr | Linear | N/A | 12 | 6.4 M | 4.4 | 15.8 |
| | Attention | | 8 | 3.8 M | 3.9 | 14.5 |
| | Linear | | 8 | 11 M | 4.5 | 15.5 |
| | Full | | 8 | 29 M | 6 | 16.6 |
| | Linear | | 4 | 43 M | 4.5 | 15.2 |
| Native Fine-Tuning | Attention | N/A | N/A | 233 M | 4.4 | 14.9 |
| | Linear | | | 672 M | 3.9 | 17.6 |
| | Full | | | 1.8 G | 4.7 | 18.2 |

**Table 5:** Resulting file size (saved in fp16 format), average training time, and approximate VRAM usage from different algorithm configuration that we consider. Note that among the hyperparameters that we consider, only trained layers, dimension, and factor have the largest impact on these metrics.

1. We can first construct the matrix $W_0 + BA$ and then perform the forward pass $(W_0 + BA)\mathbf{h} + \mathbf{b}$ with the entire matrix. This approach is generally more efficient for layers near the input and output of the UNet due to larger batch sizes or sequence lengths.[7]

2. Alternatively, we can compute $W_0\mathbf{h}$ and $B(A\mathbf{h})$ separately and sum them together, with the latter implemented via two consecutive linear layers. This method is particularly beneficial for layers in the middle of the UNet, where the latent resolution is lower, but the dimensions of the unfolded convolutional layers are significantly larger.

We have implemented LoRA using the second approach, while for LoHa and LoKr, we construct the weight matrix, so it is closer to the first approach described above. However, the second approach's advantage becomes more pronounced for convolutional layers in the middle layers of the UNet due to their significantly larger dimensions. This explains why LoRA, implemented using the second method, exhibits a relative advantage in training time when convolutional layers are involved.

**Hardware and Library.** We conduct all experiments on a Ubuntu Server equipped with four A6000 GPUs. For the training script, we re-use the public sourced code from *kohaya-ss/sd-scripts*[8], version 0.6.5. After fine-tuning, we generate the images through the API provided by *stable-diffusion-webui*[9]. Note that support for LyCORIS has been integrated as the default feature of *stable-diffusion-webui* during paper writing. The versions of stable-diffusion-webu and LyCORIS are respectively 1.6.0 and 1.9.0.dev9. We adopt the Python 3.10 interpreter with Pytorch 2.0 (Paszke et al., 2019). Moreover, we employ transformers 4.26 (Wolf et al., 2020), diffusers 0.10.2 (von Platen et al., 2022), and accelerate 0.15 (Gugger et al., 2022) for our experiment.

---

[7]Here, the term *batch size* may also refer to the redefined batch size after unfolding convolutional layers.
[8]https://github.com/kohya-ss/sd-scripts
[9]https://github.com/AUTOMATIC1111/stable-diffusion-webui

| For Anime and Movie Characters | For Stuffed Toys |
|---|---|
| {} selfie standing under the pink blossoms of a cherry tree | {} in grand canyon |
| {} in a chef's outfit, cooking in a kitchen | {} swimming in a pool |
| {} paddling a canoe on a tranquil lake | {} sitting at the beach with a view of the sea |
| {} playing with their pet dog | {} in times square |
| {} in an astronaut suit, floating in a spaceship | {} in front of a medieval castle |
| {} dressed in a a firefighter's outfit, a raging forest fire in the background | {} wearing sunglasses |
| {} wearing Victorian-era clothing, reading a book in a classic British library | {} working on the laptop |
| {} dressed as a knight, riding a horse in a medieval castle | {} on a boat in the sea |
| {} kneeling under trees with aurora in the background | {} wearing headphones |
| {} wearing red dress jumping in the sky in a rainy day | {} lying in the middle of the road |
| **For Styles** | **For Waterfall** |
| {} of a city skyline during sunset | {} at dusk with the first rays of sunlight creeping in |
| {} of a bustling marketplace in the 1800s | {} at night full of stars |
| {} of a lone tree standing in a vast desert | A frozen {} in the winter season and snow all around |
| {} of children flying kites on a breezy day | {} in a neon-lit cyberpunk cityscape |
| {} of a roaring lion in the heart of the jungle | A golden retriever in front of the {} |
| {} of a mountain climber scaling a snowy peak | A cat sitting in front of the {} |
| {} of a dancer lost in the rhythm of music | {} with a vibrant rainbow arching across its mist |
| {} of a quaint countryside cottage surrounded by wildflowers | {} in a fantasy world, with dragons flying around |
| {} of a serene monk meditating atop a hill | A painter painting the scene of the {} on canvas |
| {} of a vintage car speeding along a coastal road | {} of molten lava flowing down |
| **For Canal** | **For Garden** |
| {} surrounded by towering skyscrapers | {} with an active volcano in the background |
| {} against a backdrop of snow-capped mountains | {} with night sky |
| {} under a star-filled night sky | {} with cloudy sky |
| {} in the autumn season with colorful foliage | {} with stone pillars and intricate carvings on it |
| {} with a hot air balloon drifting all over the sky | A British shorthair cat sitting in front of {} |
| {} with a cobblestone bridge arching over the water | A koala eating leaves in {} |
| {} with a rustic wooden boat gently floating in the water | A red cardinal flying in {} |
| {} with a swan gliding gracefully in the water | A rustic wooden swing in {} |
| {} with the water turned into liquid gold, reflecting the setting sun | A laughing Buddha statue in {} |
| {} with the water replaced by a smooth pathway of glowing emeralds | {} with yellow marigold flowers |
| **For Castle** | **For Sculpture** |
| {} stands against a backdrop of snow-capped mountains | {} at a beach with a view of the seashore |
| {} surrounded by a lush, vibrant forest | {} in the middle of a highway road |
| {} in the autumn season with colorful foliage | {} in Times Square |
| {} on a rocky cliff, with crashing waves below | {} on the surface of the moon |
| {} surrounded by a field of grazing sheep | A puppy in front of {} with a close-up view |
| {} overlooks a serene lake | A cat sitting in front of {} in the snow |
| {} overlooks a serene lake, where a family of geese swims | A squirrel in front of {} |
| {} guarded by mythical elves | {} in snowy ice |
| A peacock in front of the {} | {} made of metal |
| {}, made of crystal, shimmers in the sunlight | {} digital painting 3D render in geometric style |

**Table 6:** Evaluation prompts of type <alter> for image generation. {}'s are to be filled with concept descriptors.

### D.3 EVALUATION PROMPTS

We provide below the complete list of the generalization prompts used in the generation of our evaluation images. These prompts are formulated using natural language syntax, in contrast to the tag-based structure used for training captions.

**<Alter> Generalization Prompts with Content Alteration.** The prompts of this type are organized into 8 templates as shown in Table 6. Each template contains 10 prompts, and multiple (sub-)classes use the same template. Many of these prompts are directly taken from the CustomConcept101 dataset.

**<Style> Generalization Prompts with Style Alteration.** For generalization prompts with style alteration, we use a set of 5 prompts that are shared across all the classes, as listed below.

- {} in the style of pencil drawing
- {} in the style of watercolor painting
- {} in the style of Vincent Van Gogh
- {} in the style of Claude Monet
- {} in the style of pixel art

Note that we do not generate images using <style> prompts for classes that fall under the "styles" category and we do not generate images for sub-classes "others" that fall under the "movie characters" category. This results in a total of 14,900 generated images per checkpoint, as mentioned in Section 5.2.

### D.4 EVALUATION METRICS

The metrics we examine are built upon two fundamental elements: a lower-dimensional representation space and simple, analytically defined functions operating within that space. We use an encoder to project images into this representation space, which is intended to capture perceptual relevance that is broadly applicable to a wide array of images. Following this transformation, we compute the analytically defined functions within this embedding space. In this way, the functions and the encoders can be discussed separately, and in this section, our primary focus is on these functions, assuming that the features for both images and text are readily given.

**Image Similarity.** Given two sets of images with their corresponding features $\mathcal{Z}^1 = \{z_i^1\}_{i=1}^n$ and $\mathcal{Z}^2 = \{z_i^2\}_{i=1}^m$, the average cosine similarity between the two sets is computed as

$$S_C(\mathcal{Z}^1, \mathcal{Z}^2) = \frac{1}{nm} \sum_{i=1}^n \sum_{j=1}^m \frac{\langle z_i^1, z_j^2 \rangle}{\|z_i^1\|\|z_j^2\|} = \frac{1}{nm} \sum_{i=1}^n \sum_{j=1}^m \langle \hat{z}_i^1, \hat{z}_j^2 \rangle, \tag{22}$$

where for a vector $z$ we define $\hat{z} = z/\|z\|$ as its normalized vector. We also define

$$\bar{z}^1 = \frac{1}{n} \sum_{i=1}^n \hat{z}_i^1 \quad \text{and} \quad \bar{z}^2 = \frac{1}{m} \sum_{i=1}^m \hat{z}_i^2.$$

These are the centroids of the normalized vectors. The centroid distance that we consider in our experiments is given by

$$\text{dist}_{\text{cent}}(\hat{\mathcal{Z}}^1, \hat{\mathcal{Z}}^2) = \|\bar{z}^1 - \bar{z}^2\|.$$

In the above, we write $\hat{\mathcal{Z}}^1$ and $\hat{\mathcal{Z}}^2$ for the sets of normalized vectors. It is worth noticing that with $\text{Var}(\mathcal{Z})$ denoting the variance of set $\mathcal{Z}$, it holds that

$$1 - S_C(\mathcal{Z}^1, \mathcal{Z}^2) = \frac{1}{2} \left( \|\bar{z}^1 - \bar{z}^2\|^2 + \text{Var}(\hat{\mathcal{Z}}^1) + \text{Var}(\hat{\mathcal{Z}}^2) \right). \tag{23}$$

The above formula reveals that average cosine similarity inherently rewards image sets with lower diversity, as mentioned in Section 5.3.2. This insight compels us to consider squared centroid distance as a complementary measure. Specifically, we utilize squared centroid distance when contrasting results against the Vendi scores of the images to remove the aforementioned bias.

**Text-Image Alignment.** Let $\mathcal{Z}^{\text{image}} = \{z_i^{\text{image}}\}_{i=1}^n$ be the features of a set of images and $\mathcal{Z}^{\text{text}} = \{z_i^{\text{text}}\}_{i=1}^n$ be the features of their corresponding prompts. To evaluate text-image alignment, we compute the cosine similarity between an image and its corresponding prompt and average the results. This gives

$$S_C'(\mathcal{Z}^{\text{image}}, \mathcal{Z}^{\text{text}}) = \frac{1}{n} \sum_{i=1}^n \frac{\langle z_i^{\text{image}}, z_i^{\text{text}} \rangle}{\|z_i^{\text{image}}\|\|z_i^{\text{text}}\|}.$$

Moreover, in our experiments, we process the prompts as following before feature encoding:

- For anime and movie characters, we remove trigger words and retain class words.
- For scenes and stuffed toys, we replace trigger words with the corresponding class names, as given in Table 4.
- For styles and outfit sub-classes, we remove both trigger words and class words. Any extra commas or prepositions that remain are also removed.

**Diversity.** We employ the Vendi score, as proposed by Friedman & Dieng (2023), to assess the diversity within a set of images. Given the images' features $\mathcal{Z} = \{z_i\}_{i=1}^n$, the definition of the Vendi score relies on the eigenvalues $\lambda_1, \ldots, \lambda_n$ of a matrix $K/n$, where $K$ is a kernel matrix constructed using a positive semi-definite kernel function $k$. More precisely, the entries of $K$ are calculated as

$K_{ij} = k(z_i, z_j)$. In this work, we specifically employ a linear kernel between normalized vectors to compute the entries of the kernel matrix, i.e.,

$$k(z, z') = \frac{\langle z, z' \rangle}{\|z\|\|z'\|}.$$

With these in mind and using the convention $0 \log 0 = 0$, the Vendi score of the feature set is given by

$$\mathrm{VS}(\mathcal{Z}) = e^{-\sum_{i=1}^{n} \lambda_i \log \lambda_i}.$$

The Vendi score can be interpreted as the "effective number of modes" within the feature set. In particular, it is more sensitive to changes in the number of modes compared to the intra-dissimilarity. See also Stein et al. (2023) and our experiments in Appendix H.1 for further support for the use of the Vendi score.

**Style Loss.** Unlike the metrics described above, the style loss introduced by Johnson et al. (2016), is intrinsically associated with a specific encoder: the VGG network (Simonyan & Zisserman, 2015). For its computation, we use feature maps from several convolutional layers to extract information. In particular, we select the layers conv1_1, conv2_1, conv3_1, conv4_1, and conv5_1 from a pretrained VGG-19 network in our experiments.[10] Each of these layers provides an output with a shape of $c_i \times h_i \times w_i$, where $i$ identifies the particular layer (ranging from 1 to 5 in our setup), and $c_i$, $h_i$, $w_i$ represent respectively the number of channels, the height, and the width of the feature maps. These outputs are further reshaped to matrices $\psi_i(x)$ of shape $c_i \times h_i w_i$, where $x$ represents the image in input. Following the reshaping, we calculate the corresponding normalized Gram matrices

$$G_i(x) = \frac{1}{c_i h_i w_i} \psi(x) \psi(x)^\top.$$

Finally, to obtain the style loss, we compute the squared Frobenius norm of the differences between the Gram matrices of the two images. This is done for each selected layer, and the results are summed, leading to

$$\mathcal{L}_{\mathrm{style}}(x, x') = \sum_{i=1}^{5} \|G_i(x) - G_i(x')\|^2.$$

Importantly, as the VGG network can take images of any resolution in input (as long as the short edge has at least 224 pixels) and the size of the gram matrices does not depend on the size of the input images, we can compute the style loss between two images of different resolutions. In our evaluation of base model preservation, we compute the style losses between pairs of images generated from identical prompts and seeds but differing in whether the model is fine-tuned or not. We then average these individual style losses to arrive at a single metric for each (sub-)class.

### D.5 ENCODERS

For the computation of image similarity and Vendi score, we consider three different encoders: DINOv2 (Oquab et al., 2023), CLIP (Radford et al., 2021), and ConvNeXt V2 (Woo et al., 2023). This choice is motivated by Stein et al. (2023), where the authors compared a number of encoders and concluded that DINOv2, CLIP VIT-L/14, and MAE (He et al., 2022) are much better at extracting high-level representations that align with human perception. In particular, they showed that the Fréchet distance (Dowson & Landau, 1982; Heusel et al., 2017) measured in the representation space of these networks are more correlated with human error rate in distinguishing real from fake images, compared to other networks they examined. Although ConvNeXt V2 was not considered by Stein et al. (2023), we include it in the correlation analysis of Appendix E for it being itself an improvement over MAE. We use models of size "L" throughout our work.

As for the main experiments, we opt for DINOv2 as our encoder for evaluating image similarity and Vendi scores following the recommendation of Stein et al. (2023). To accommodate DINOv2's input resolution, all the images are padded (in cases where the image is non-square) and resized to dimensions of $224 \times 224$.[11] On the other hand, among the aforementioned encoders, CLIP is the only

---

[10]https://pytorch.org/vision/main/models/generated/torchvision.models.vgg19.html (Accessed: 2023-08-16)

[11]The impact of the resizing method is also investigated in Appendices E and H.1

one that can be used to evaluate text-image alignment. More details on the used pretrained networks are provided below.

- DINOv2: We use the original pretrained DINOv2 ViT-L/14 model available via PyTorch Hub.[12]
- CLIP: Following Stein et al. (2023), we use the OpenCLIP ViT-L/14 implementation (Ilharco et al., 2021) trained on DataComp-1B (Gadre et al., 2023), which is reported to be the best CLIP ViT-L/14 model in the OpenClip library. We use the image features that are obtained after the final projection layer.
- ConvNeXt V2: We load the pretrained convnextv2_large.fcmae_ft_in22k_in1k_384 checkpoint using the timm toolkit (Wightman, 2019). The pre_logits features are used.

### D.6 METRIC VALUE PROCESSING

In this section, we discuss in detail how the resulting metric values are processed and analyzed.

**Normalization and Aggregation.** To account for the varying ranges of metric values across different classes and sub-classes, we employ a rank-based normalization technique. Specifically, for each metric value (e.g., average cosine similarity between dataset and generated images, Vendi score) computed for a given class or subclass and a specific type of prompt, we rank it against other metric values computed with the same metric, for the same class or sub-class and prompt type, but for different checkpoints. This allows us to assign a normalized score to each metric value, with the highest-ranking score set to 1 and the lowest set to 0. These normalized scores are equally distributed between 0 and 1. This rank-based normalization is motivated by our focus on relative performance and enables more equitable comparisons across varying classes.

Building upon the above, we then first average the normalized scores across sub-classes of the same class and then average across different classes within the same category. For the scatter plots, we also average these scores across random seeds, and the error bars indicate the standard error of these scores across both random seeds and classes.

**SHAP Analysis.** The SHAP analysis is performed on the average normalized scores for each category. As shown in Figure 2, we consider 5 dependent variables: algorithm, trained layers, epoch, capacity, and learning rate level. Among these, algorithm and trained layers are treated as categorical features, while the remaining ones are treated as numerical features. The learning rate level is set to either 1, 2, or 3, corresponding to the smallest, intermediate, and largest learning rates considered for each algorithm. As for the capacity variable, we assign a value of 1 for default hyperparameters and 2 for configurations with altered dimensions and factors in LoRA, LoHa, and LoKr. However, LoRA and LoHa configurations with higher dimensions and an alpha of 1, along with LoKr configurations with a factor of 12, are excluded from the SHAP analysis. It is also important to note that the capacity variable is not meaningful for native fine-tuning. To accommodate this, we designate a capacity value of 3 for all native fine-tuning configurations, and subsequently adjust the SHAP value for the algorithm variable by adding up the SHAP value from the capacity variable for these configurations.

Due to the presence of categorical feature, we used CatBoostRegressor for the analysis.[13] We set iterations to 300 and the learning rate to 0.1 while leaving the remaining hyperparameters untouched.

## E  Correlation Analysis

In our study, we are faced with a plethora of metrics that could be considered for evaluating the models. Moreover, the precise definitions and computation of these metrics can vary depending on factors like the encoder used or the resizing methods applied. Given this complexity, we are interested in the following two main questions: how do these variables affect the results, and what relationships exist between different metrics? To answer these, we go beyond the metrics studied in Section 5.3 and present a comprehensive correlation analysis in this appendix.

---

[12]https://github.com/facebookresearch/dinov2 (Accessed: 2023-08-14)
[13]https://catboost.ai/en/docs/concepts/python-reference_ catboostregressor (Accessed: 2023-08-25)

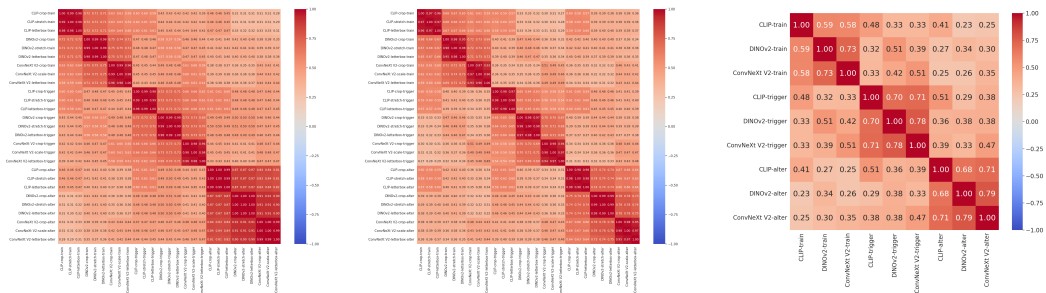

**(a)** Correlation matrix heatmap for image similarity.

**(b)** Correlation matrix heatmap for image similarity based on squared centroid distance.

**(c)** Correlation matrix heatmap for Vendi score.

**Figure 5:** Pearson correlation for metrics computed using different encoders and resizing methods, evaluated on images generated from different types of prompts (best viewed when zoomed in).

**Resizing Methods.** Given a target resolution $s$, we consider four different methods for resizing a rectangular image.

- **Scale:** The image is resized such that the smaller edge matches the target resolution while preserving the original aspect ratio.
- **Letterbox:** This method adds black-colored padding around the original image to fit it within the target resolution, preserving the aspect ratio of the original image.
- **Center Crop:** After scaling the image, the central part fitting within the target resolution is retained, and the outer portions are cropped away.
- **Stretch:** The original image is stretched or compressed to fit the target resolution, potentially causing distortion as the aspect ratio is not preserved.

It is important to note that while the last three methods produce images with dimensions of $s \times s$, the "scale" method maintains the original aspect ratio. Therefore, "scale" is only applicable when the encoder can accept rectangular images. For this reason, in our experiments, we use the "scale" method exclusively for ConvNeXt V2 and Vgg19 (the latter is used in the computation of style loss). The "stretch" method is applied for DINOv2 and CLIP. Both "letterbox" and "center crop" (abbreviated as "crop" in the figures) are tested across DINOv2, CLIP, and ConvNeXt V2.

**Implementation Details.** We compute the Pearson correlation coefficients of the normalized scores. We do not perform aggregation across classes or sub-classes before the computation.

### E.1 INFLUENCE OF ENCODERS, RESIZING METHODS, AND PROMPT TYPES

We first investigate the influence of the choice of encoder and resizing method. To this end, we compute two types of image similarity (based on either average cosine similarity or squared centroid distance as discussed in Appendix D.4) and the Vendi score using different encoders and resizing methods. Note however that as our generated images are all squares, the difference in resizing methods only matters when dataset images are also included in the computation in the metrics (which is not the case for Vendi score). Moreover, for the computation of Vendi score on images generated from the prompts of type <alter>, we first compute the Vendi score for each unique prompt before averaging them to form a single score for each class or sub-class.

The results of the correlation analysis are presented in Figure 5. Generally speaking, we can see that that the choice of resizing method has little influence on the outcomes, with correlation coefficients ranging between 0.95 and 1. While the choice of encoder has a greater influence, the resulting metrics still exhibit strong positive correlations (typically between 0.65 and 0.9). In particular, we observe a higher correlation between image similarity metrics calculated with different encoders when images are generated using generalization prompts, as opposed to using training prompts or prompts containing only the concept descriptor. We believe that this occurs because the image similarity score in the former cases is primarily influenced by how closely the generated images adhere to either the prompts or the dataset images. This overarching trend can be readily discerned

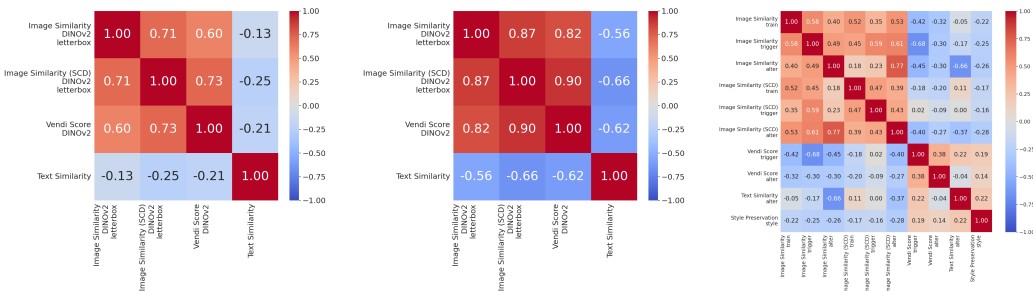

(a) Correlation matrix heatmap for image and text similarity metrics on <train> prompts.

(b) Correlation matrix heatmap for image and text similarity metrics on <alter> prompts.

(c) Correlation matrix heatmap for the main metrics studied in this work.

**Figure 6:** Pearson correlation for the main metrics studied in this work (best viewed when zoomed in). SCD stands for squared centroid distance.

regardless of the encoder used. Conversely, for training prompts and simple captions containing only the concept descriptor, a more nuanced analysis is required to gauge image similarity, making the choice of encoder more critical.

Finally, we observe a lower positive correlation (often below 0.5) between metrics evaluated on images generated from different prompt types. This suggests that model rankings could vary considerably depending on the types of prompts under consideration, highlighting again the importance of evaluating images generated by each type of prompt individually. Interestingly, utilizing the CLIP encoder tends to yield the highest correlations across images generated using different prompt types.

### E.2 RELATION BETWEEN DIFFERENT METRICS

We next examine how the various types of metrics under consideration relate to each other. The results are shown in Figure 6. Since our primary focus here is on the metrics used in Section 5.3, we employ DINOv2 as the encoder for the computation of image similarity and Vendi score, and we use letterbox resizing to resize dataset images to the target resolution when computing image similarity.

As we can see from Figures 6a and 6b, text and image similarity metrics exhibit a strong negative correlation when generalization prompts are used to generate images. However, this correlation weakens considerably when training prompts are utilized. This suggests again that the trade-off between producing images that look similar to those in the training set and producing images that follow the prompts is much easier to be evaluated by the considered metrics. Finally, we make two observations in Figure 6c. First, the correlation between Vendi score, text similarity, and base model style preservation (as measured by the style loss) is relatively weak, even though they are all negatively correlated with image similarity. This underscores the fact that these metrics serve as distinct indicators of model performance and should be independently evaluated. Secondly, we observe that the correlation between Vendi score and image similarity weakens when the latter is assessed using squared centroid distance. This observation is consistent with our prior discussion in Appendix D.4 and validates our decision to contrast diversity against image similarity as measured by squared centroid distance in Figure 2.

## F    Supporting Plots

To substantiate the claims made in Section 5.3, we include in this section a comprehensive set of plots for the key metrics under review. This set comprises both the beeswarm plots showcasing the SHAP values and the scatter plots contrasting various metrics. Among these, we have made the decision to omit the scatter plots for the epoch 30 checkpoints, as we believe that the plots for epochs 10 and 50 are sufficient to demonstrate the sensitivity of the results to the number of training epochs. As in Section 5.3, these plots are organized according to concept categories.

It is important to acknowledge that some discrepancies do arise when comparing these plots to the principles laid out in Section 5.3. Upon closer examination, we categorize these discrepancies into two groups. First, there are genuine deviations, which indicate that the models behave in ways not entirely captured by our initial guidelines. Second, there are metric-induced deviations, which arise from the limitations or biases in the metrics themselves. As we explore the plots by category in the remainder of this section, we will briefly touch upon some of these discrepancies. Further elaboration on the genuine deviations, supported by qualitative examples, are presented in Appendix G.4.

## F.1 PLOTS FOR CATEGORY "MOVIE CHARACTERS"

The plots for the "movie characters" category are shown in Figures 7 to 9. These results largely align with our general guidelines. Specifically, we observe that for generalization prompts with content alteration, LoKr exhibits high image similarity and low text similarity at lower epochs, while LoRA achieves similar result at higher training epochs.

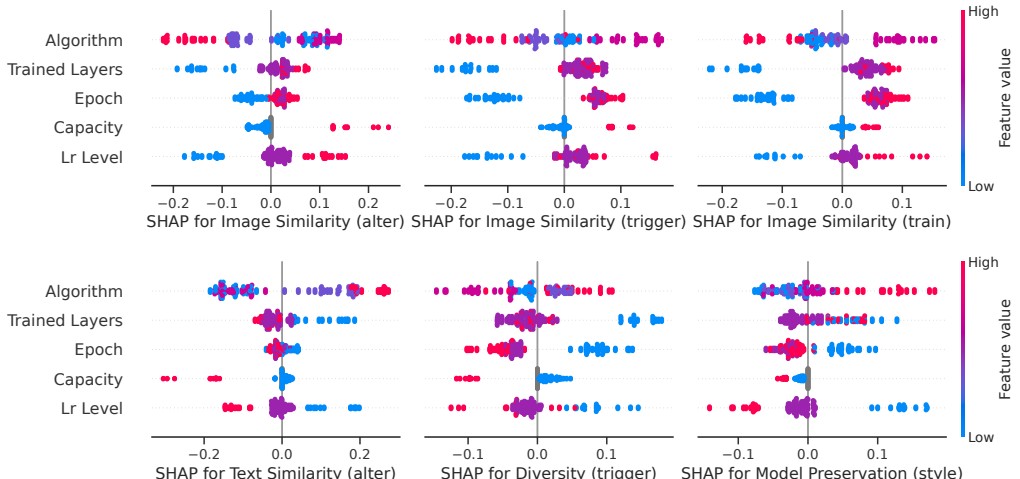

**Figure 7:** SHAP beeswarm charts for the category "Movie Characters" showing the impact of diverse algorithm factors on the evaluation metrics. LoRA is in blue, LoHa is in purple, LoKr is in purple red, and native fine-tuning is in red.

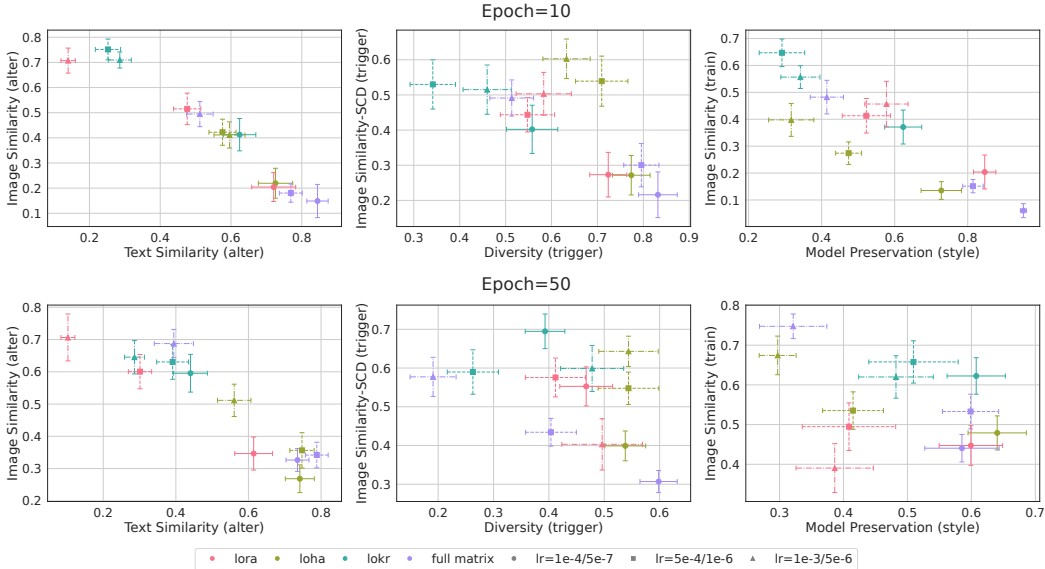

**Figure 8:** Scatter plots comparing different evaluation metrics for the category "Movie Characters", with variations across algorithms and learning rates. As in Section 5.3, in the middle column we use squared centroid distance (SCD) to measure image similarity.

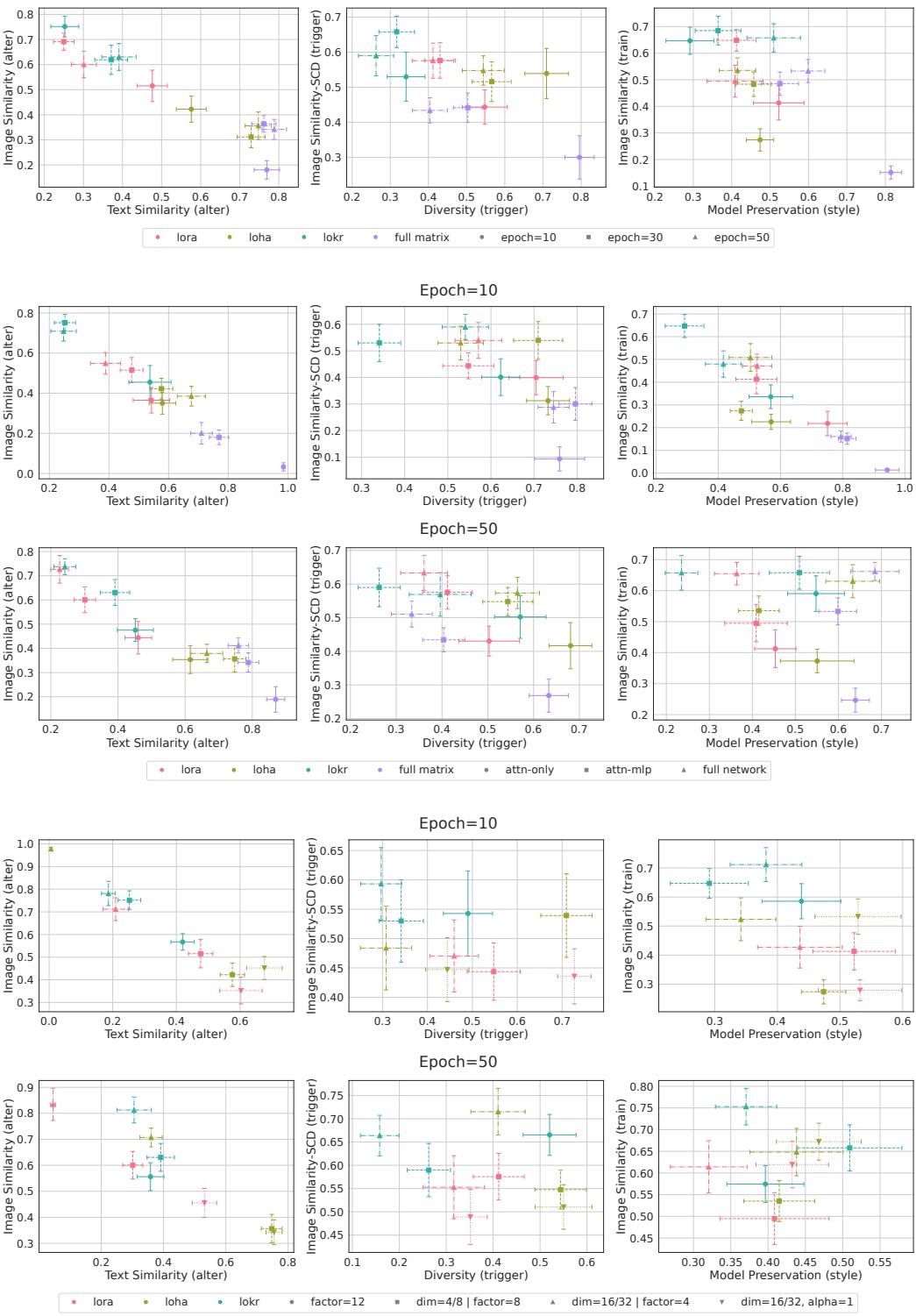

**Figure 9:** Scatter plots comparing different evaluation metrics for the category "Movie Characters", with variations across algorithms and either *i*) top: number of training epochs, *ii*) middle: trained layers, or *iii*) bottom: dimensions, factors, and alphas.

## F.2 PLOTS FOR CATEGORY "SCENES"

The plots for the "scenes" category are shown in Figures 10 to 12. As both the checkpoints after 30 and 50 epochs are severely overtrained for this category, we only consider the epoch 10 checkpoints when performing SHAP analysis. Again, the results mostly agree with the claims made in Section 5.3, though the difference between LoRA and LoHa are less pronounced here. We do however note that LoRA now has the best base model style preservation among all the algorithms. The qualitative comparison in Appendix G.4.1 further validates this observation.

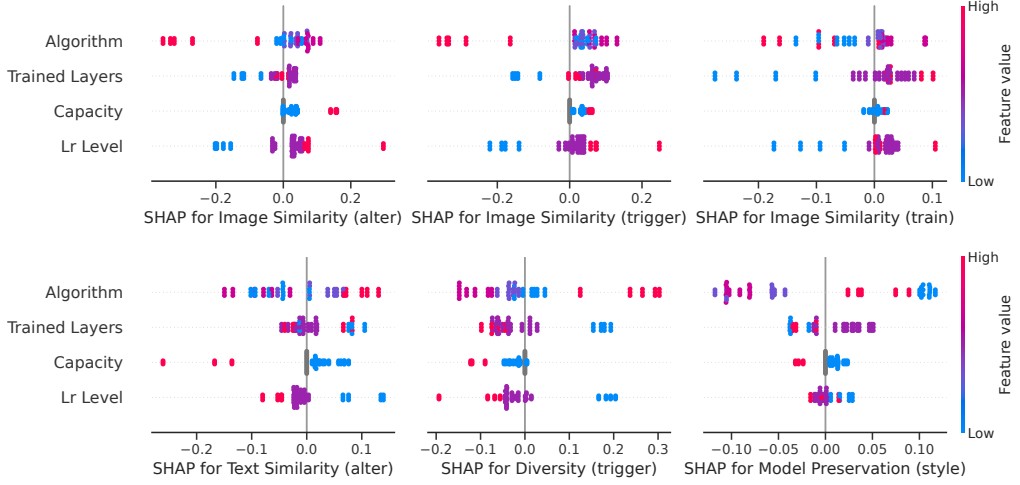

**Figure 10:** SHAP beeswarm charts for the category "Scenes" showing the impact of diverse algorithm factors on the evaluation metrics. LoRA is in blue, LoHa is in purple, LoKr is in purple red, and native fine-tuning is in red.

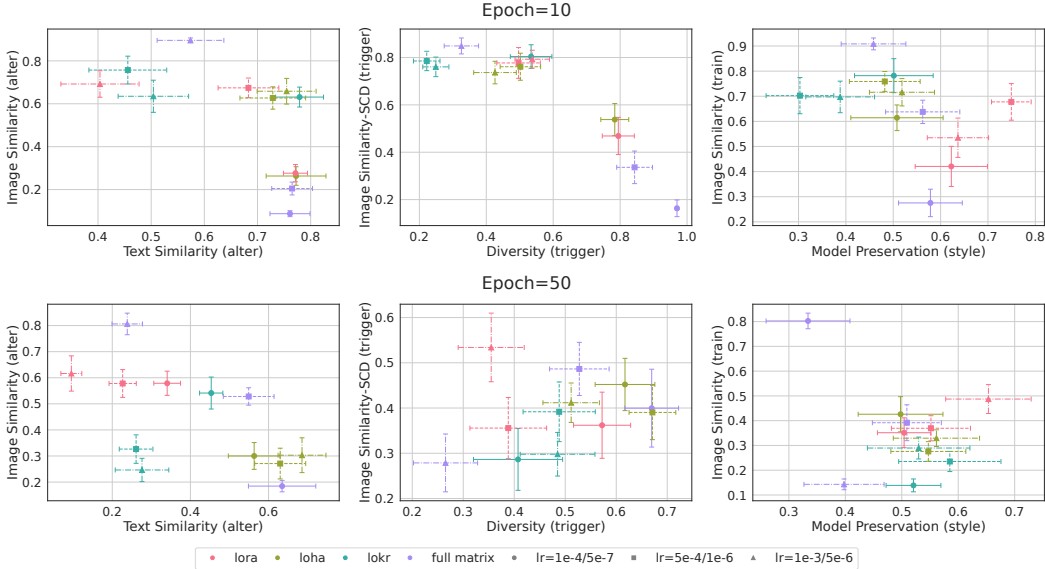

**Figure 11:** Scatter plots comparing different evaluation metrics for the category "Scenes", with variations across algorithms and learning rates. As in Section 5.3, in the middle column we use squared centroid distance (SCD) to measure image similarity.

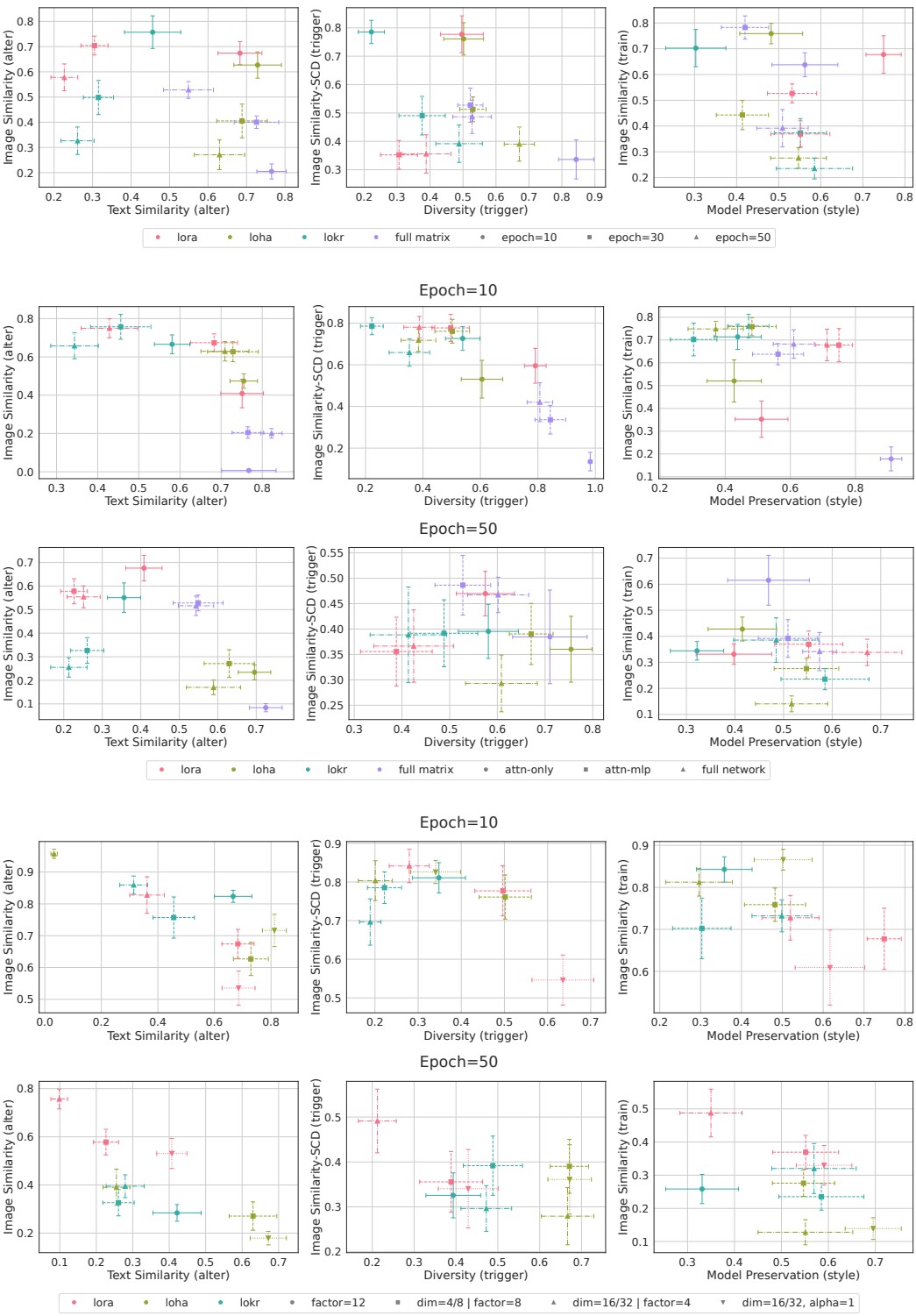

**Figure 12:** Scatter plots comparing different evaluation metrics for the category "Scenes", with variations across algorithms and either *i*) top: number of training epochs, *ii*) middle: trained layers, or *iii*) bottom: dimensions, factors, and alphas.

### F.3 PLOTS FOR CATEGORY "STUFFED TOYS"

The plots for the "stuffed toys" category are shown in Figures 13 to 15. Following the reasoning of Appendix F.2, the SHAP analysis is only conducted with epoch 10 checkpoints. For the most part, these plots corroborate the statements of Section 5.3. Although the evaluation metrics indicate that training LoRA leads to greater diversity at the expense of base model style preservation, this trend is not visually discernible in the generated images. Another interesting observation is the significant improvement in base model style preservation for LoKr when the factor is decreased, which corresponds to an increase in model capacity. We further investigate this phenomenon in Appendix G.4.3.

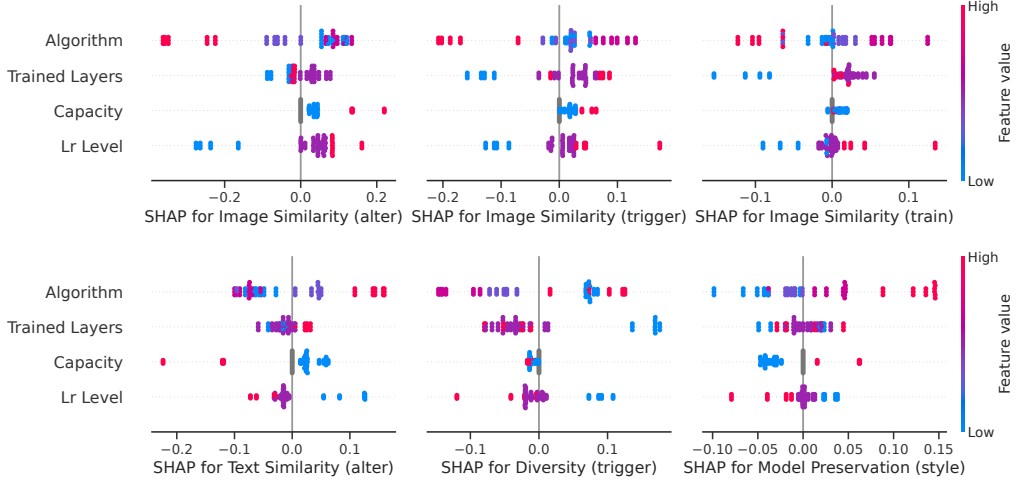

**Figure 13:** SHAP beeswarm charts for the category "Stuffed Toys" showing the impact of diverse algorithm factors on the evaluation metrics. LoRA is in blue, LoHa is in purple, LoKr is in purple red, and native fine-tuning is in red.

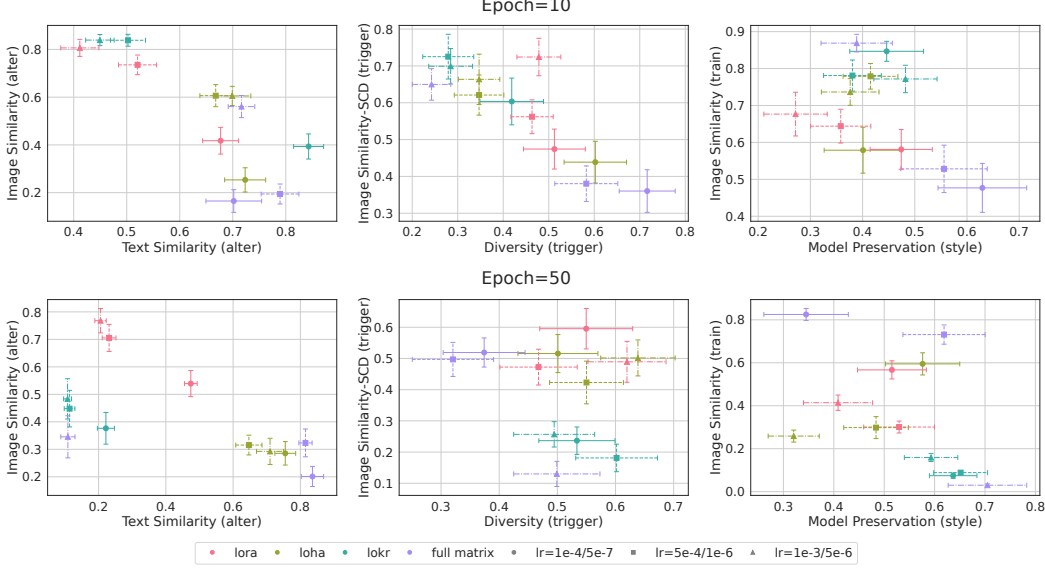

**Figure 14:** Scatter plots comparing different evaluation metrics for the category "Stuffed Toys", with variations across algorithms and learning rates. As in Section 5.3, in the middle column we use squared centroid distance (SCD) to measure image similarity.

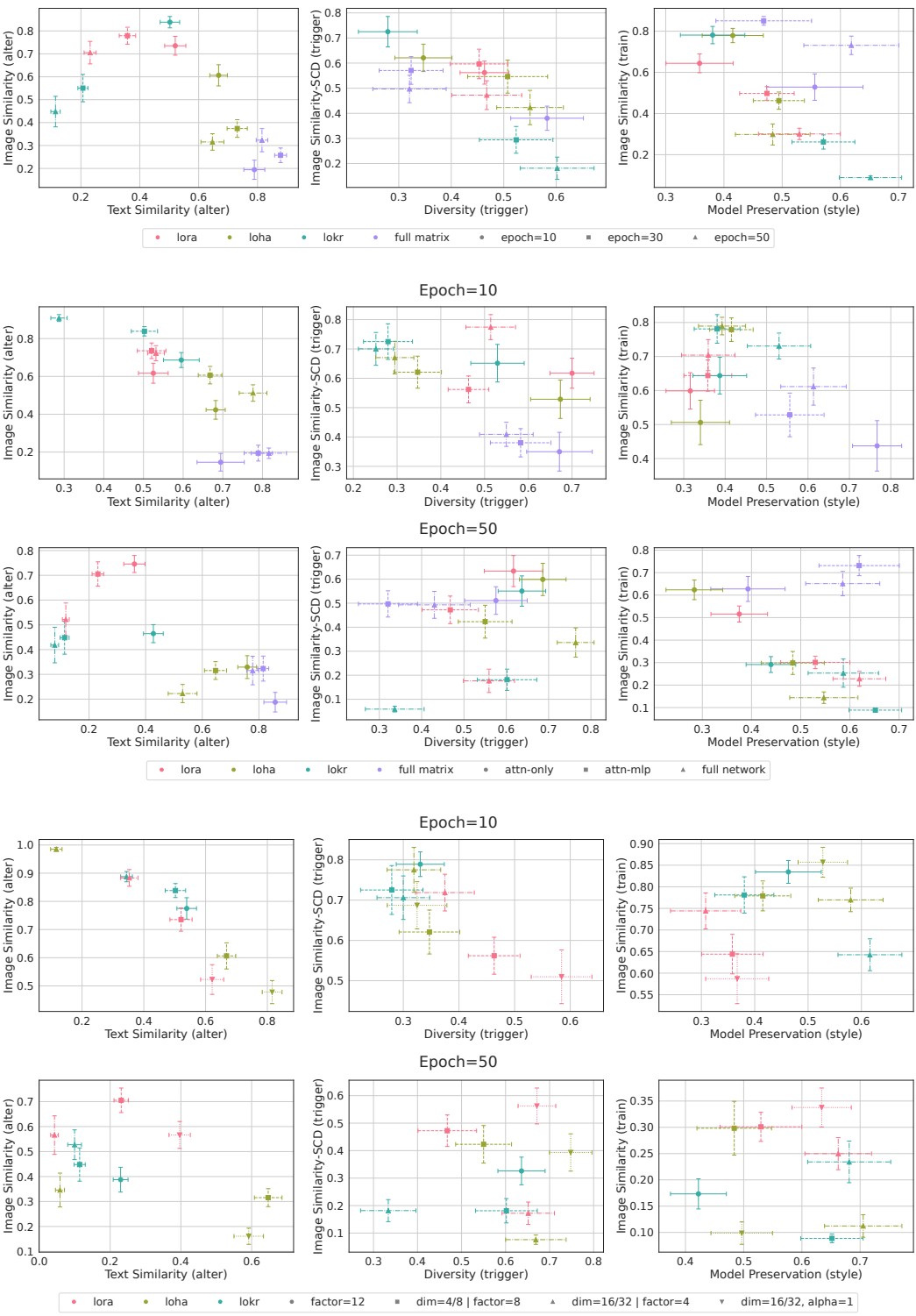

**Figure 15:** Scatter plots comparing different evaluation metrics for the category "Stuffed Toys", with variations across algorithms and either *i*) top: number of training epochs, *ii*) middle: trained layers, or *iii*) bottom: dimensions, factors, and alphas.

### F.4 PLOTS FOR CATEGORY "ANIME CHARACTERS"

The plots for the "anime characters" category are shown in Figures 16 to 18. As we will discuss in Appendix G.2, the metrics that we consider are less suitable for this category. There are thus a number of inconsistencies with our general guidelines that can be attributed to the limitations of these metrics, especially for the image similarity of images generated from training prompts. In spite of this, it could still be surprising to see that text-image alignment for generalization prompts gets improved over training. Our qualitative results in Appendix G.4.2 suggest that this can indeed happen.

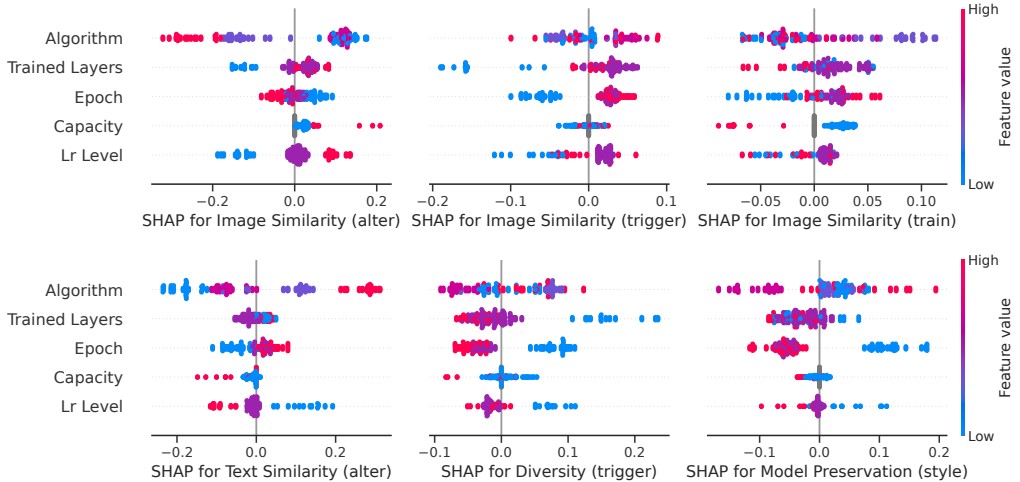

**Figure 16:** SHAP beeswarm charts for the category "Anime Characters" showing the impact of diverse algorithm factors on the evaluation metrics. LoRA is in blue, LoHa is in purple, LoKr is in purple red, and native fine-tuning is in red.

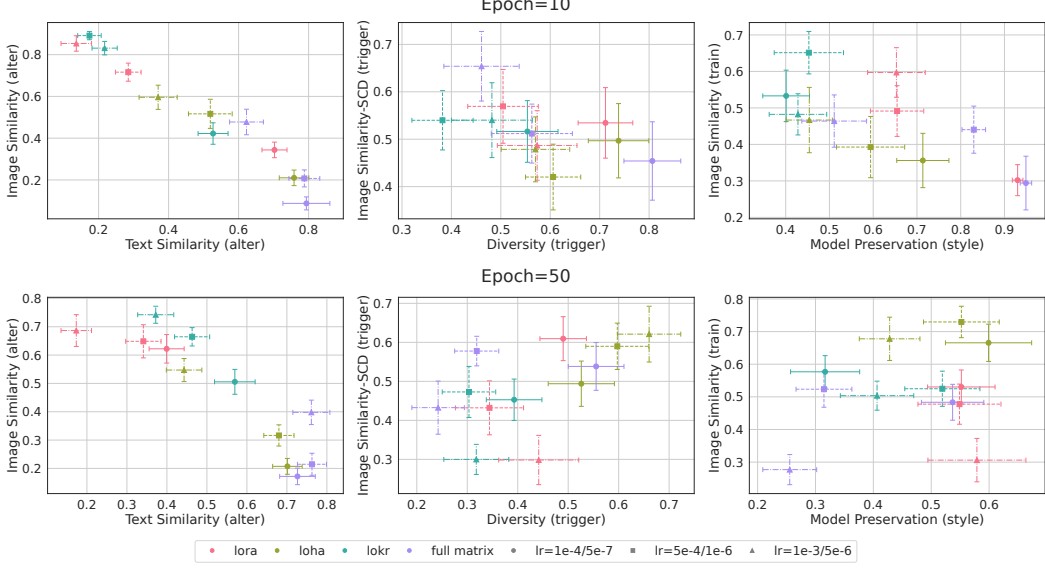

**Figure 17:** Scatter plots comparing different evaluation metrics for the category "Anime Characters", with variations across algorithms and learning rates. As in Section 5.3, in the middle column we use squared centroid distance (SCD) to measure image similarity.

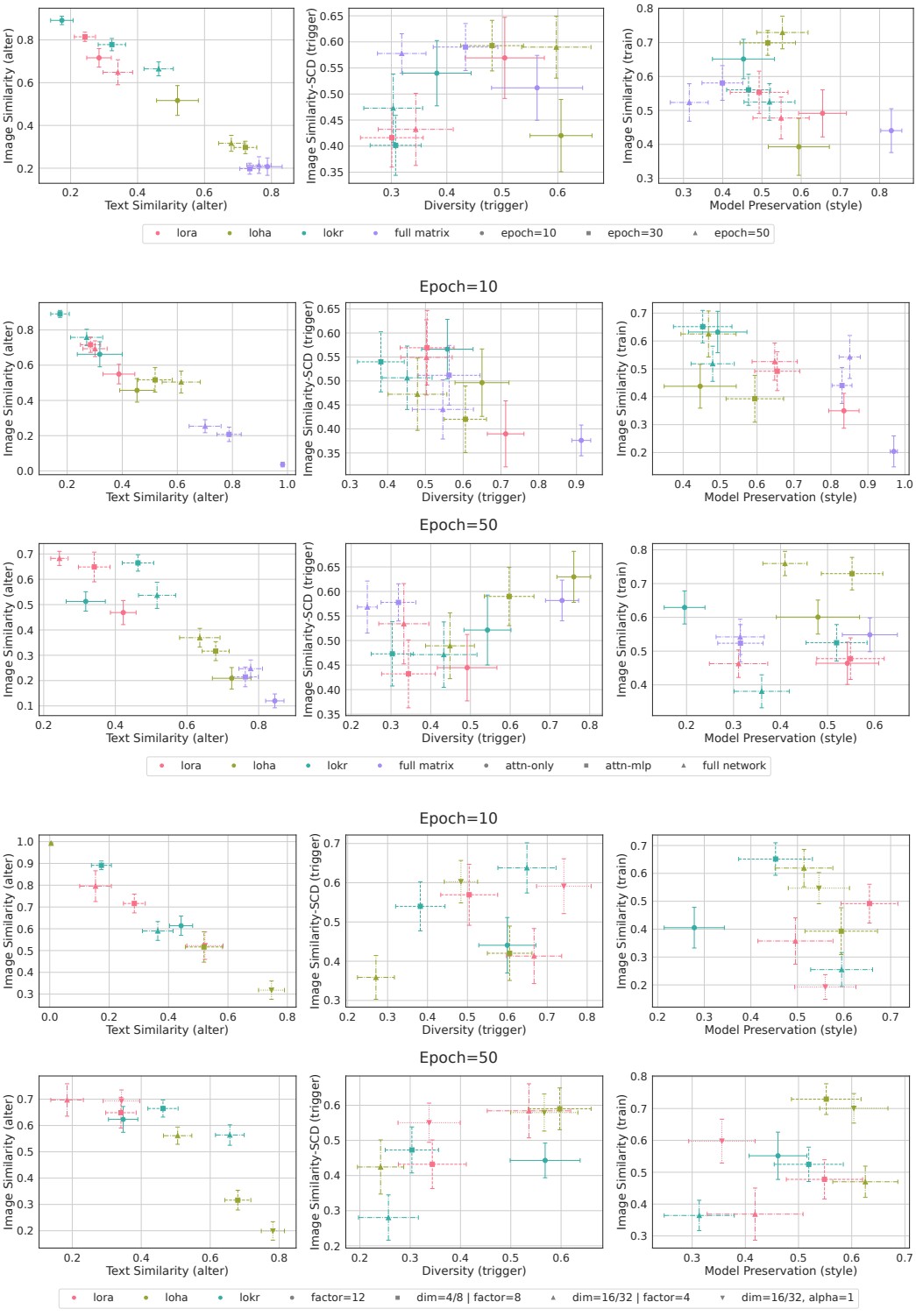

**Figure 18:** Scatter plots comparing different evaluation metrics for the category "Anime Characters", with variations across algorithms and either *i*) top: number of training epochs, *ii*) middle: trained layers, or *iii*) bottom: dimensions, factors, and alphas.

## F.5 PLOTS FOR CATEGORY "STYLES"

The plots for the "styles" category are shown in Figures 19 to 21. These plots differ from the plots for the other categories in the two following ways: First, we also use average style loss to evaluate concept fidelity. For consistency, in the plots, we still use "Image Similarity" to refer to the average cosine similarity measured in the DINOv2 feature space, and we use "Style Similarity" to refer to the similarity measured by the average style loss between dataset and generated images. Second, we do not measure base model style preservation here.

As we can see in the plots, the tendency suggested by the two different ways to measure fidelity do not always agree, and violations of the claims made in Section 5.3 are common. In fact, despite the seemly advantage of using style loss to measure style similarity as we will demonstrate in Appendix H.1, we recognize this metric may still fall short of capturing all the nuanced elements that should be considered when comparing styles reproduced by different models. Furthermore, the very notion of what we refer to as "style" is inherently ambiguous and may require more specific and finely detailed methods of study. Consequently, it becomes challenging for us to render a definitive judgment on this topic. For illustration, some generated images for this category are shown in Appendix G.5.

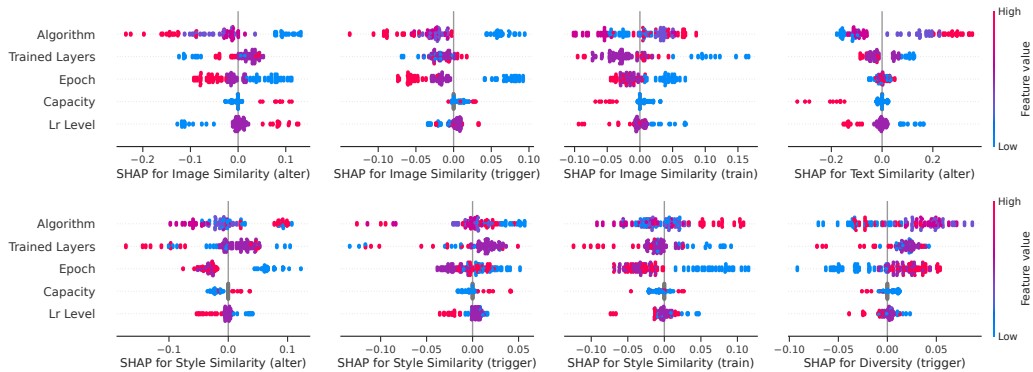

**Figure 19:** SHAP beeswarm charts for the category "Styles" showing the impact of diverse algorithm factors on the evaluation metrics. LoRA is in blue, LoHa is in purple, LoKr is in purple red, and native fine-tuning is in red.

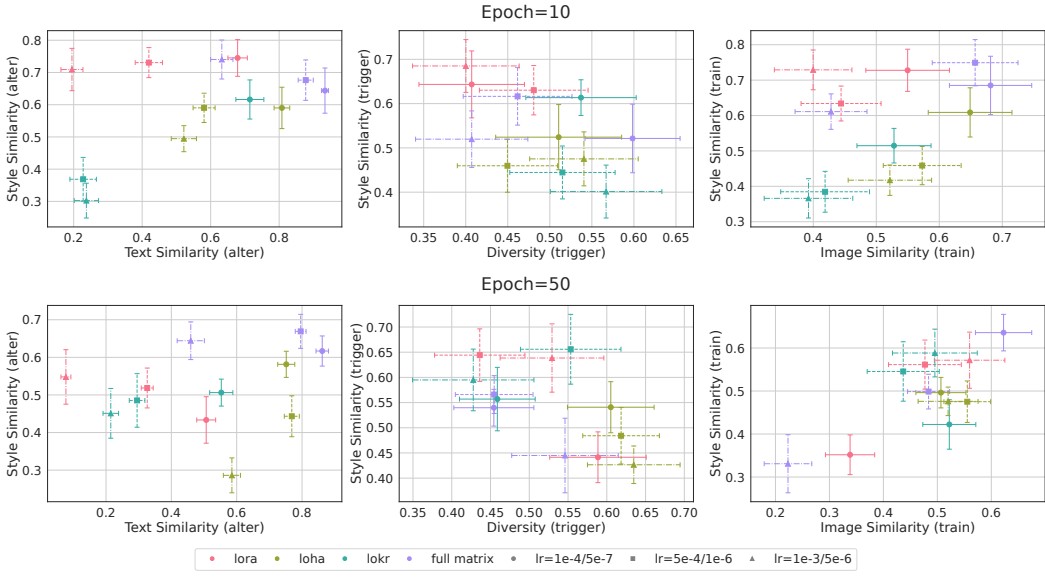

**Figure 20:** Scatter plots comparing different evaluation metrics for the category "Styles", with variations across algorithms and learning rates.

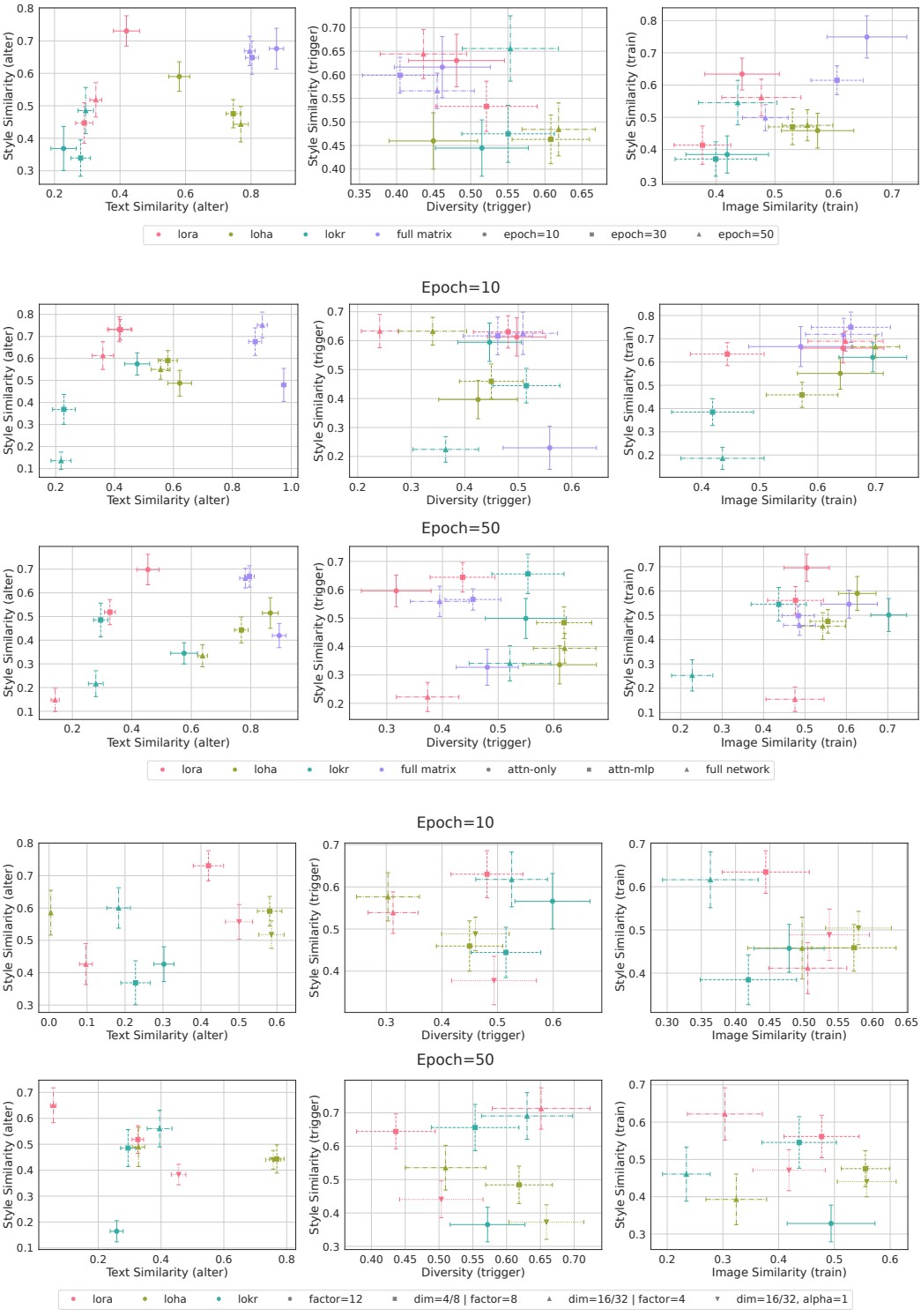

**Figure 21:** Scatter plots comparing different evaluation metrics for the category "Styles", with variations across algorithms and either *i*) top: number of training epochs, *ii*) middle: trained layers, or *iii*) bottom: dimensions/factors, and alphas.

# G  Further Qualitative Results

In this section, we include an extensive set of qualitative results to support the claims that we have made throughout the paper. We highlight especially the challenges in algorithm evaluation, the prevailing trends related to the impact of different algorithmic factors, and some noteworthy deviations from these general principles. As for the prompts that are used to generate these images, please refer to Appendix D.3.

## G.1  Discrepancy of Results Across Classes

We have seen in Section 5.3 and Appendix F that the performance of an algorithm can vary greatly across different concept categories. This variability is due to both differences in the number of training samples as well as intrinsic differences in the complexities of the concepts themselves. In Figures 22 and 23, we further demonstrate that such discrepancies can still occur even among classes with the same type of objects and comparable number of images. In Figure 22, the left model learns the "lobster stuffed toy" concept better, while the right model has more success with the "panda stuffed toy" concept. In Figure 23, the left model allows to generalize the "canal" concept to broader contexts while the right model has more flexibility when dealing with the "garden" concept.

## G.2  Unreliability of Metrics

While it is expected that the considered metrics may not always give results that align with human perception, it remains the hope that these deviations are simply "noises" that can be averaged out when we perform the evaluation on a large number of images. Unfortunately, this is not necessarily the case. For example, CLIP score for measuring text-image alignment often fails to understand compositional relationships between objects or attributes (Thrush et al., 2022; Yuksekgonul et al., 2023). Here, we further demonstrate that the image similarity metrics we consider may have some inconsistencies in fully capturing the nuances of likeness across various image types, and this becomes more noticeable for categories that are less commonly seen during the pretraining of the encoders. Concretely, Figure 17 suggests that a model trained with native fine-tuning at a learning rate of $5 \cdot 10^{-6}$ has the lowest image similarity for images generated with training prompts in the "anime characters" category. On closer inspection, we find this is misleading. A specific example is given in Figure 24, where we demonstrate that none of the encoders we consider could accurately assess similarity in anime character appearance.

Such observations caution against an over-reliance on metrics, and emphasize the importance of using task-relevant metrics and encoders. In particular, the metrics we consider in our work may not be nuanced enough for specialized applications.

## G.3  Illustrating the Impact of Different Algorithm Components

In this part, we illustrate the general principles that we established in Section 5.3 through qualitative examples of three (sub-)classes: Abukuma [dark color uniform], Bohdi Rock [realistic], and castle. Example images for these three (sub-)classes are provided in Figure 25.

### G.3.1  A Case Study on Sub-Class "Abukuma [Dark Color Uniform]"

It is known that Stable Diffusion 1.5 does not perform well in generating anime-style images. Moreover, complex outfits are generally hard to learn. With these in mind, we believe that the uniforms of "Abukuma" would be a good test bed to evaluate the methods' capacity in learning more difficult concepts. We visualize the images generated with the prompt "$[V_{abukuma}]$ anime girl, $[V_{dark\ uniform}]$ outfit" in Figure 26 (we consider the epoch 50 checkpoints here for they being the ones that are the most trained). Although none of the models can perfectly reproduce the outfit, we do notice that a number of them can generate quite similar uniforms. This includes LoRA and LoKr trained with default parameters, and native fine-tuning with learning rate $5 \cdot 10^{-6}$. For LoHa, among all the considered configurations, only increasing both dimension and alpha allows us to learn the outfit to some extent. For LoRA and LoKr, with the hyperparameters that we consider, fine-tuning only the attention layers is however not sufficient for learning the appearance of the outfit.

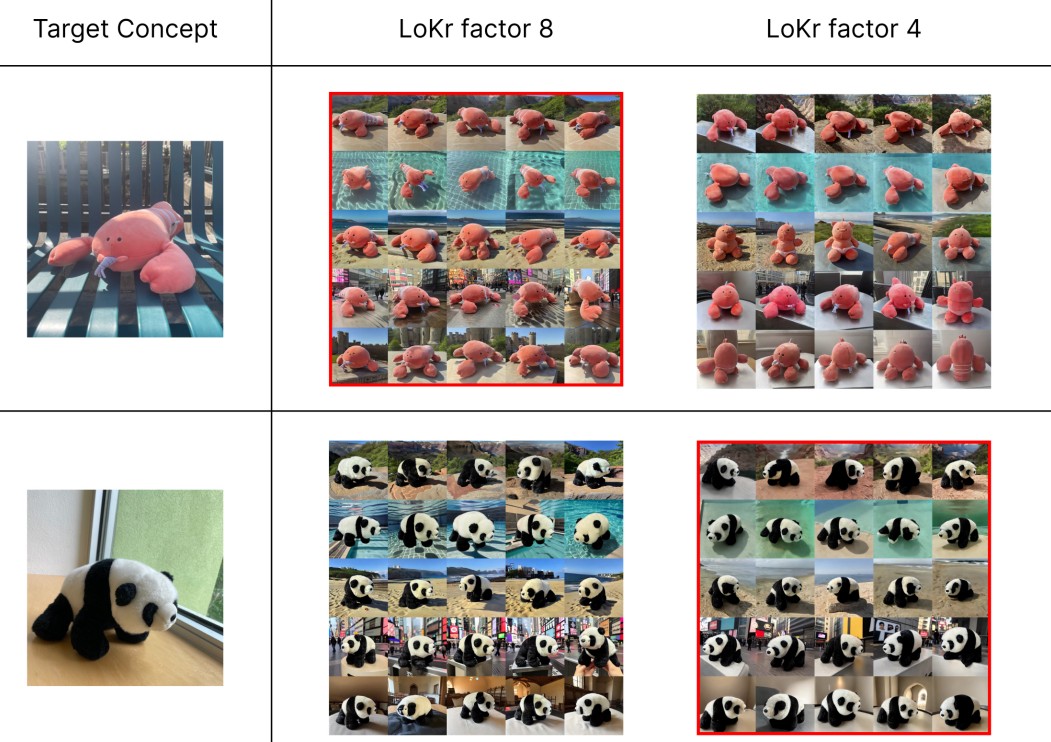

**Figure 22:** Example generations for "lobster" and "panda" classes using two 10 epoch checkpoints trained with different algorithm configurations. The first 5 prompts of type <alter> are used to generate these samples.

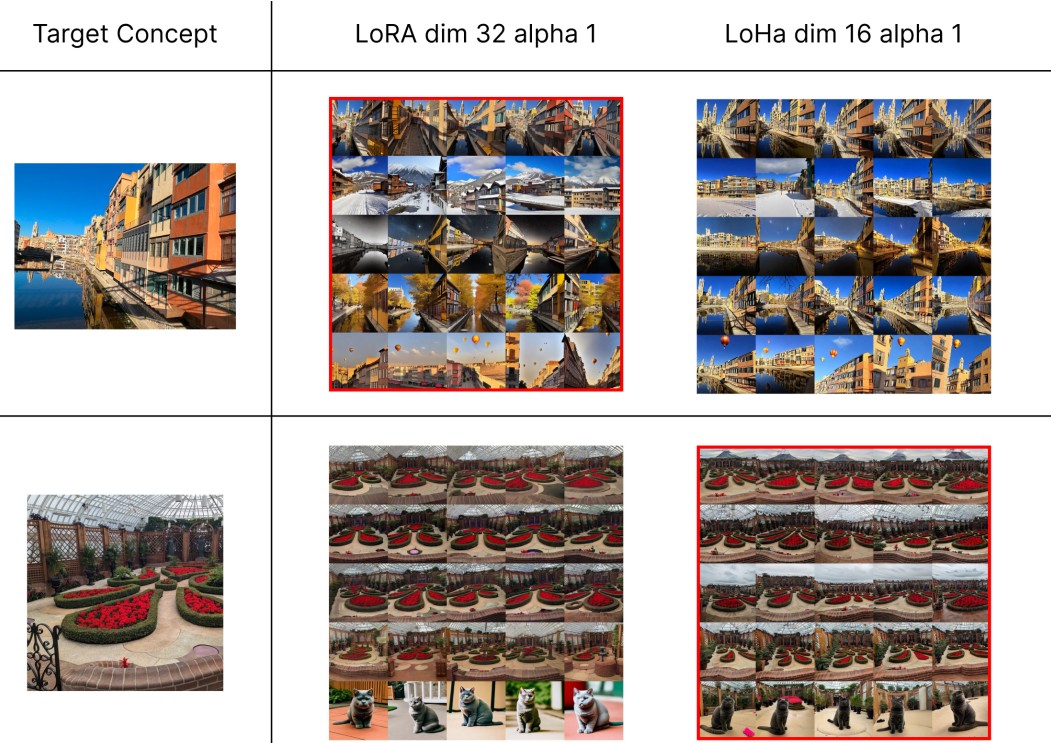

**Figure 23:** Example generations for "canal" and "garden" classes using two 10 epoch checkpoints trained with different algorithm configurations. The first 5 prompts of type <alter> are used to generate these samples.

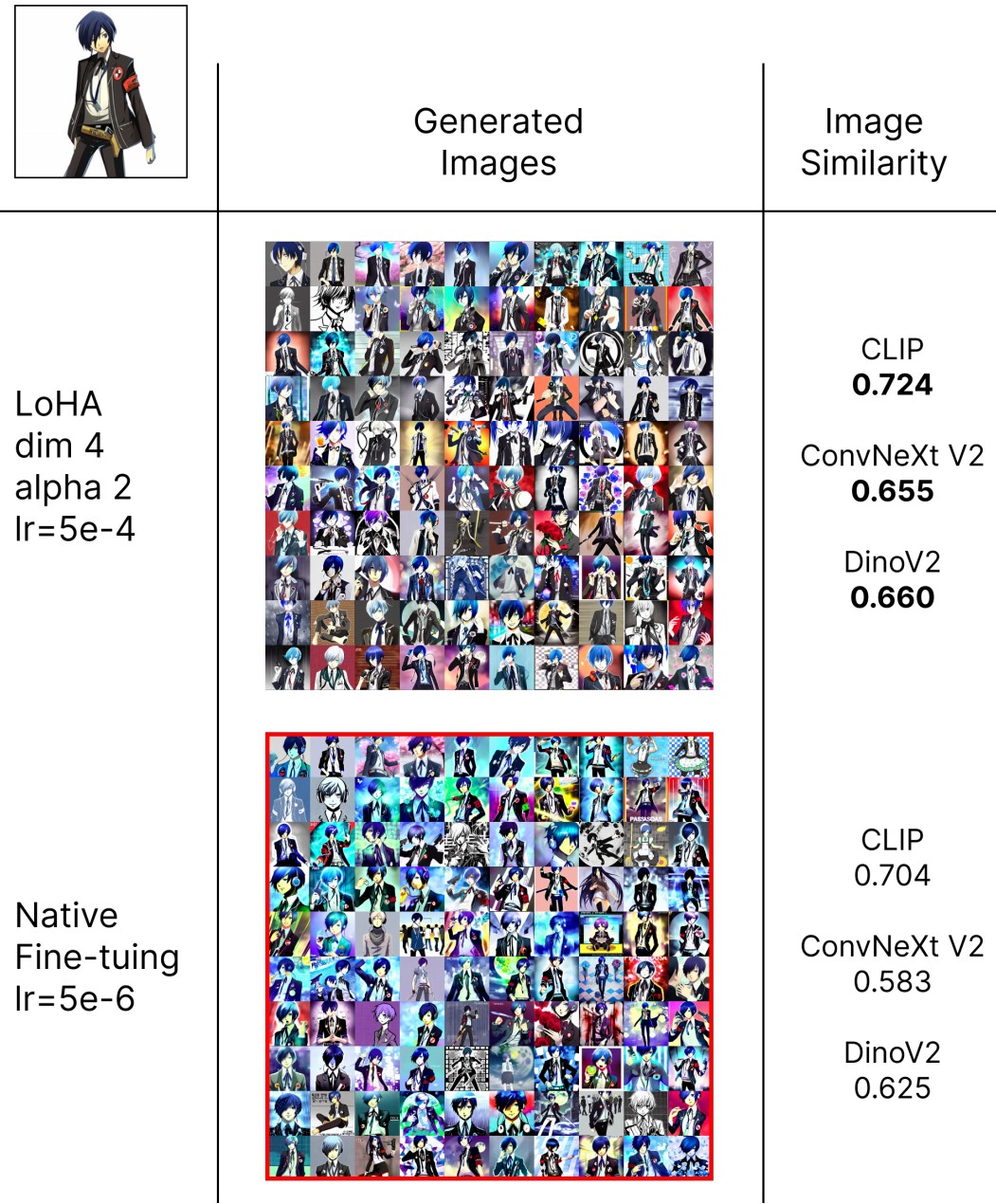

**Figure 24:** An illustrative example showing that image similarity scores could be misleading. Here, we compute the average cosine similarity between the features of the generated images and of the reference image shown on the left top corner. We consider images generated from two epoch 50 checkpoints. No matter what encoder we use, we get a higher score for the top model. Nonetheless, looking closely at the hairstyle, the outfit, and the armband, one would conclude that the bottom model performs better in generating the same character.

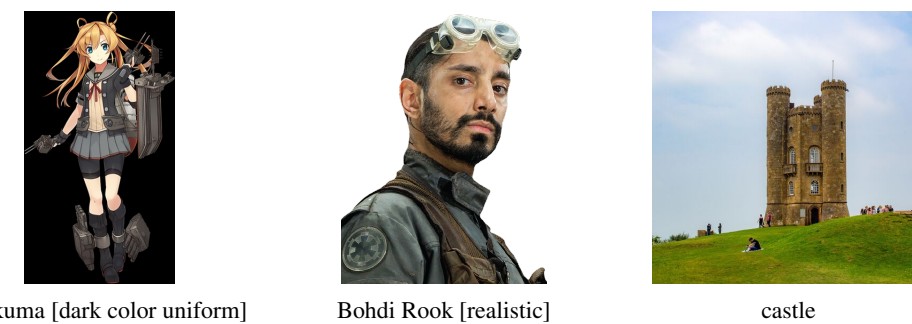

| Abukuma [dark color uniform] | Bohdi Rook [realistic] | castle |

**Figure 25:** Example training images for the three (sub-)classes studied Appendix G.3.

| LoRA, dim 8, alpha 4 | LoRA, dim 32, alpha 16 | LoRA, dim 32, alpha 1 | LoRA, attn-only |
| LoKr, factor 8 | LoKr, factor 4 | LoKr, factor 12 | LoKr, attn-only |
| LoHa, dim 4, alpha 2 | LoHa, dim 16, alpha 8 | LoHa, dim 16, alpha 1 | LoHa, full network |
| LoHa, lr $10^{-3}$ | Native | Native, full network | Native, lr $5 \cdot 10^{-6}$ |

**Figure 26:** Example generations for "Abukuma [dark color uniform]" from checkpoints trained with different configurations. The images are generated with only the concept descriptor (i.e., trigger and class words) as input. These checkpoints are obtained after 50 training epochs and the default hyperparameters are used unless otherwise specified. Models that learn this concept more successfully are marked with red frames.

### G.3.2 A Case Study on Class "Castle"

We next zoom in on the learning of the class "castle". We first present a number of failure cases in Figure 27. For models with higher capacity, we observe an increased tendency for mode collapse and artifacts that arise from overtraining (the latter, of course, can potentially be avoided by reducing the number of training epochs). On the other hand, when the learning rate is too small, even with native fine-tuning, we may fail to learn the concept correctly, and this cannot be solved with a larger number of training epochs. In particular, we see that after 50 epochs of native fine-tuning at learning rate $10^{-6}$, the image quality already gets compromised while the concept is still barely learned.

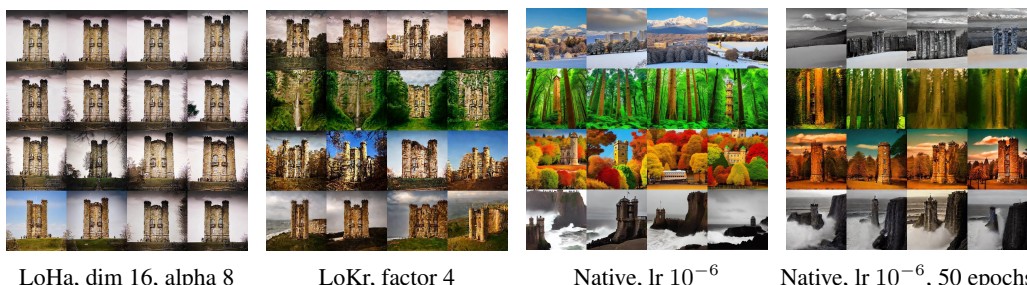

| LoHa, dim 16, alpha 8 | LoKr, factor 4 | Native, lr $10^{-6}$ | Native, lr $10^{-6}$, 50 epochs |
|---|---|---|---|

**Figure 27:** Illustrations of typical failures that we may encounter during concept customization. This includes mode collapse (leftmost), overtraining artifact (second to the left), and underfitting (right two). These images are generated for class "castle" with prompts of type <alter>. The left three models are trained for 10 epochs.

The generated images of several other models are presented in Figures 29 to 32 (we show the results obtained from the epoch 10 checkpoints as 30 epochs systematically cause overtraining for this concept). First, comparing LoRA, LoHa, LoKr trained with default hyperparameters and native fine-tuning with learning rate $5 \cdot 10^{-6}$, we can see that LoKr and LoHa respectively give the most and the least fitted model. This can be adjusted by modifying the hyperparameters. In particular, we show in these figures that a LoRA with higher model capacity (higher dimension and alpha) would be more fitted than a LoKr with lower model capacity (higher factor).

Another observation from these figures is the occurrence of concept leaks from other classes. For instance, elements like waterfalls and sculptures from two other classes (illustrated in Figure 28) appear in several generated samples in Figure 31. This leakage appears to stem from either the word "forest" or "peacock", and is more pronounced in models that are more fitted to the concepts. Notably, this phenomenon is less observed in LoHa, LoKr with a factor of 12, and native fine-tuning.

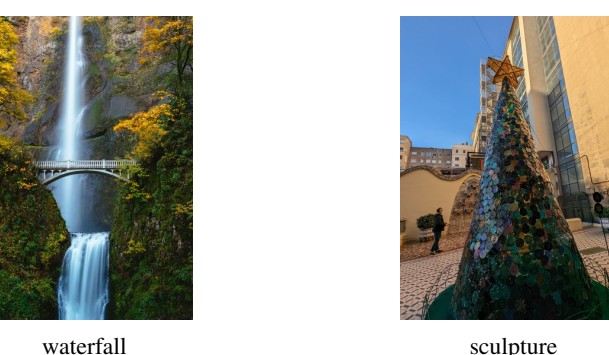

| waterfall | sculpture |
|---|---|

**Figure 28:** Examples training images for the two classes "waterfall" and "sculpture".

### G.3.3 A Case Study on Sub-Class "Bodhi Rook [Realistic]"

Here, we turn our attention to the generation of photorealistic images of Bodhi Rook, with sample generations shown in Figures 33 to 36. We find most models perform equally well in generating the character (while some details may still be to desired, we leave the judgement to the readers).

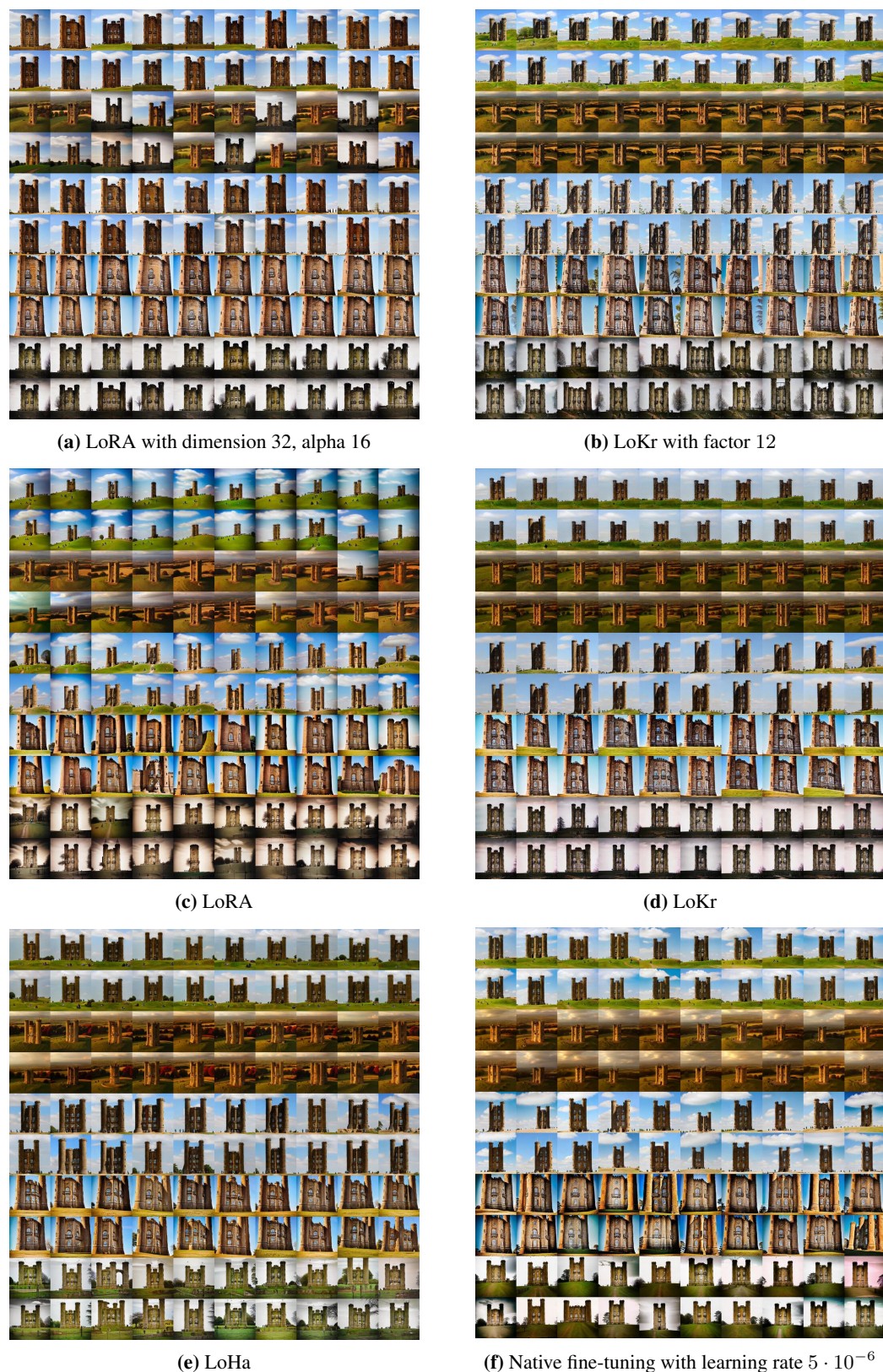

**Figure 29:** Synthetic images of class "castle" that are generated using the prompts of type <train>. The models are trained for 10 epochs and the default hyperparameters are used unless otherwise specified.

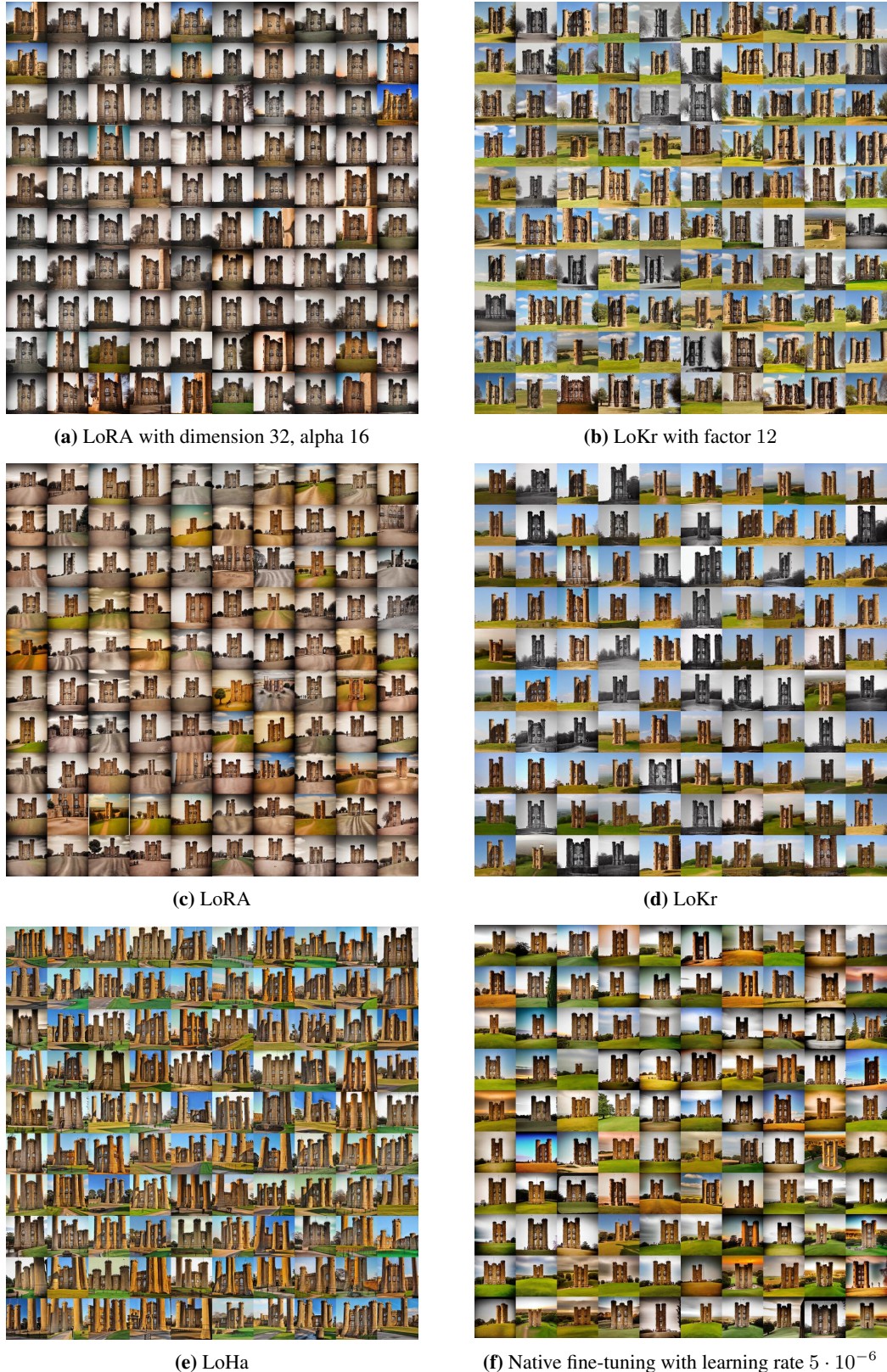

**(a)** LoRA with dimension 32, alpha 16

**(b)** LoKr with factor 12

**(c)** LoRA

**(d)** LoKr

**(e)** LoHa

**(f)** Native fine-tuning with learning rate $5 \cdot 10^{-6}$

**Figure 30:** Synthetic images of class "castle" that are generated using the prompts of type <trigger>. The models are trained for 10 epochs and the default hyperparameters are used unless otherwise specified.

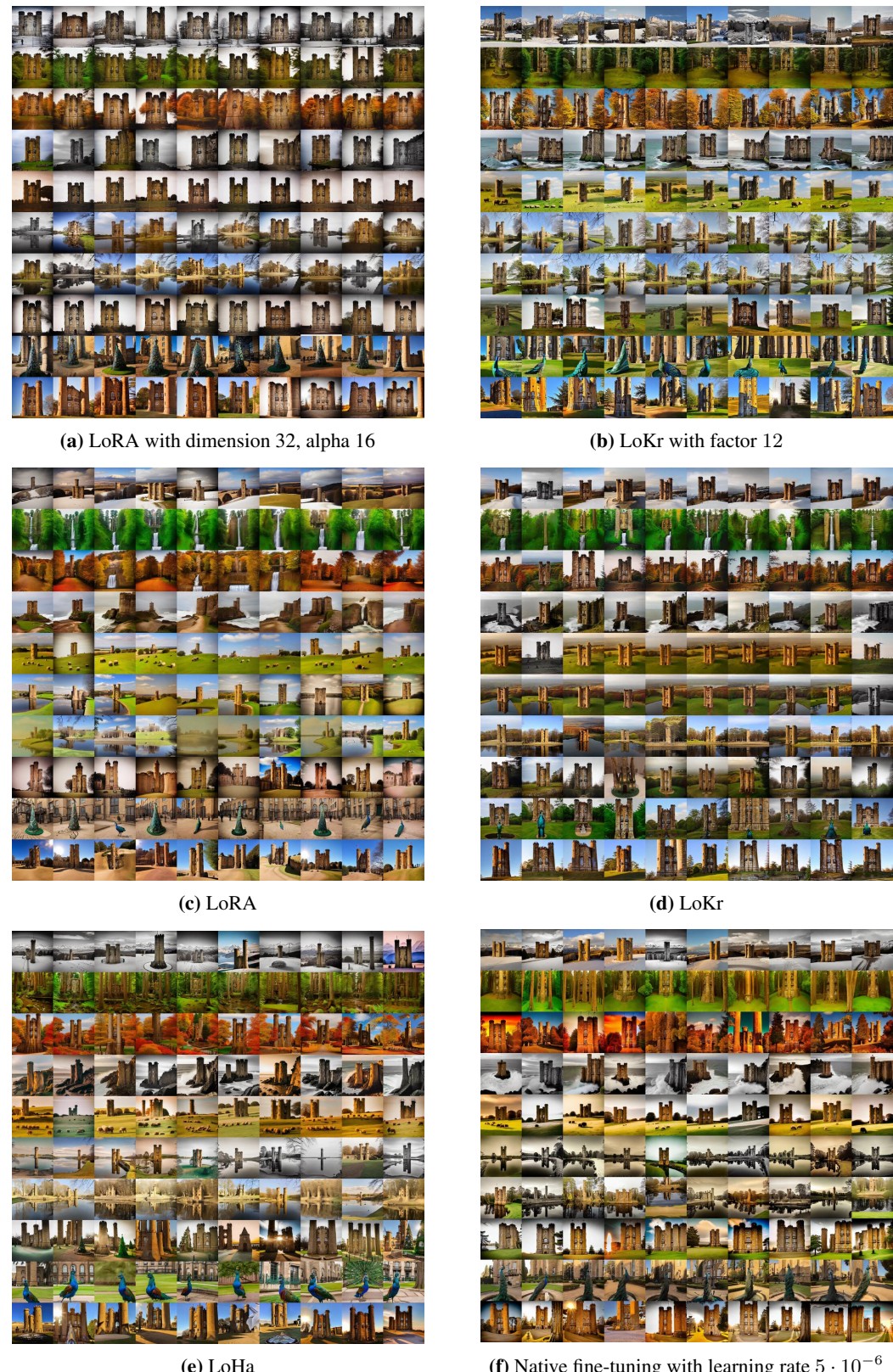

**(a)** LoRA with dimension 32, alpha 16

**(b)** LoKr with factor 12

**(c)** LoRA

**(d)** LoKr

**(e)** LoHa

**(f)** Native fine-tuning with learning rate $5 \cdot 10^{-6}$

**Figure 31:** Synthetic images of class "castle" that are generated using the prompts of type <alter>. The models are trained for 10 epochs and the default hyperparameters are used unless otherwise specified.

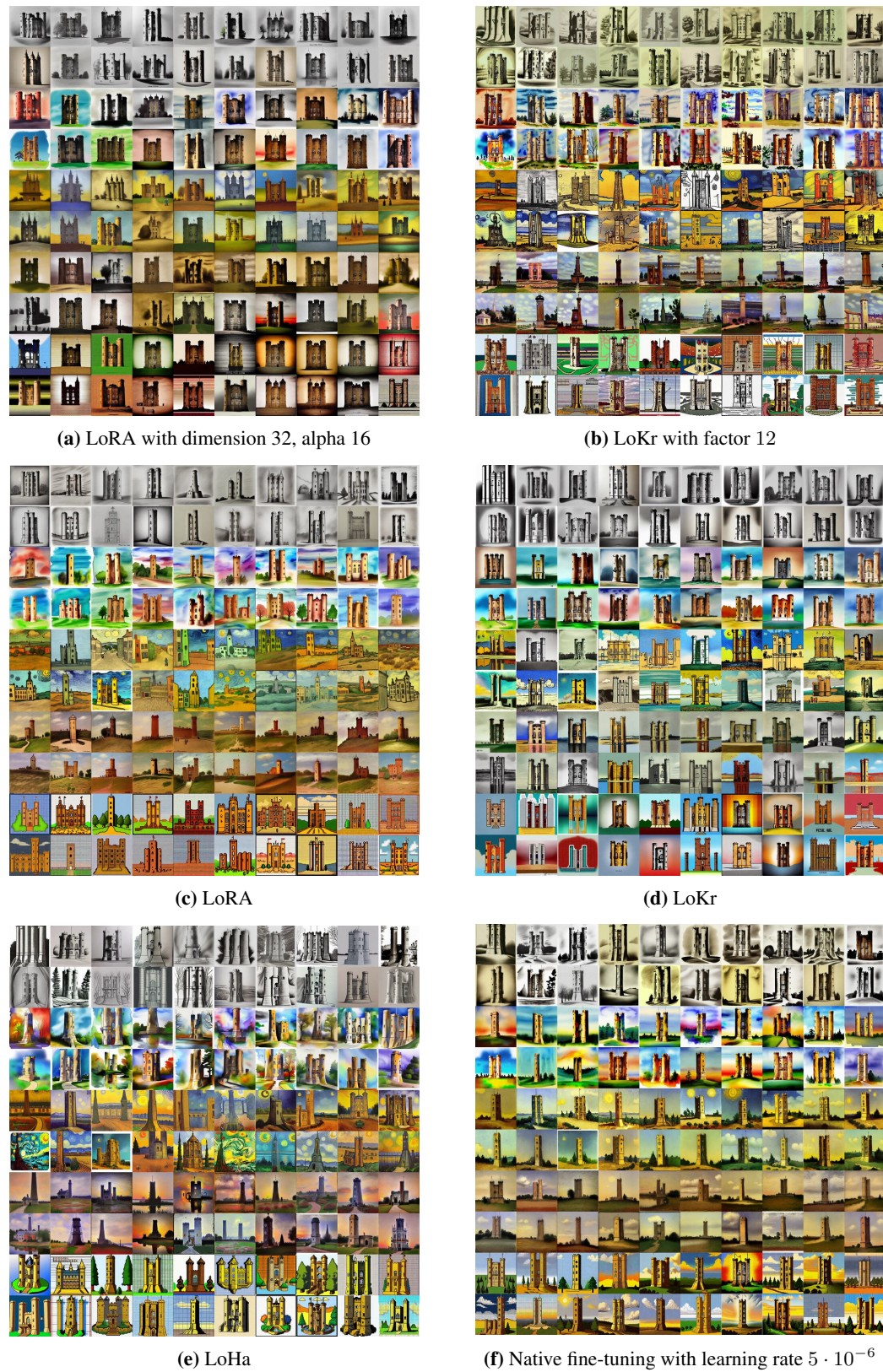

(a) LoRA with dimension 32, alpha 16

(b) LoKr with factor 12

(c) LoRA

(d) LoKr

(e) LoHa

(f) Native fine-tuning with learning rate $5 \cdot 10^{-6}$

**Figure 32:** Synthetic images of class "castle" that are generated using the prompts of type <style>. The models are trained for 10 epochs and the default hyperparameters are used unless otherwise specified.

Nonetheless, the models that are considered to be "more fitted", such as LoRA and LoKr with default hyperparameters, and native fine-tuning with learning rate $5 \cdot 10^{-6}$ would have a higher tendency to generate clothes that look similar to those in the dataset, even when it is asked to generate a different outfit, as seen in the "chef's outfit" examples in the second row of each image grid in Figure 35. In spite of this, these models are still flexible enough to capture other elements of the prompts, such as backgrounds, like cherry blossoms, poses, like kneeling, or interactions, like playing with dogs.

Yet another way to distinguish between these groups of models is to look at image diversity when we only prompt with the concept descriptor "$[V_{bodhi\ rook}]$ man, realistic". Models with lower learning rate (LoKr with learning rate $10^{-4}$, native fine-tuning with default learning rate $10^{-6}$) and LoHa clearly have more diverse generations as can be seen from Figure 34. It is worth noticing that it may not always be possible to really compare the diversity of two sets of images, as one set may be more diverse in background while the other being more diverse in pose, but this does not seem to be the case here. Finally, we do observe that native fine-tuning seems to yield models with the best base model style preservation for this sub-class, no matter whether we train with learning rate $10^{-6}$ or $5 \cdot 10^{-6}$, as shown in Figure 36.

### G.4 VIOLATIONS OF THE GENERAL PRINCIPLES

The goal of this section is to showcase a number of violations to the general principle that we outlined in Section 5.3. Such exceptions indicate that we are still far from having a comprehensive understanding of the myriad factors that influence the fine-tuning process.

### G.4.1 LoRA WITH BETTER STYLE PRESERVATION FOR "CANAL" AND "WATERFALL"

We have seen in Figure 36 that for the sub-class "Bodhi Rook [realistic]", native fine-tuning seems to perform the best in terms of base model style preservation. Nonetheless, the plots shown in Appendix F.2 suggest that this would not be the case for the "scenes" category. Instead, the metrics suggest that LoRA has the best base model style preservation in this case. Upon visual inspection, we confirmed that this is indeed the case, as evidenced by the examples provided in Figure 37. This observation highlights the benefits of reducing the number of fine-tuned parameters especially when working with a small dataset.

### G.4.2 IMPROVED CONTROLLABILITY OVER TRAINING

Although more training would generally increase concept fidelity at the expense of controllability, this rule does not hold universally. For example, Figures 16 to 18 in Appendix F.4 suggest that, when considering the "anime characters" category, the models' text-image alignment for generalization prompts actually improves with more training. While the difference is rather subtle, we do observe this for several configurations as shown in Figures 38 and 39. Specifically, note how "Tsushima Yoshiko" is more likely to be depicted riding a horse, as indicated by the prompt in Figure 38 after longer training (8th row of the grids). Similarly, "Yuuki Makoto" dons a more accurate space suit when LoKr undergoes more training, as shown in the 5th row of the grids in Figure 39. This tendency might be linked to the fact that Stable Diffusion 1.5 struggles with generating anime-style images. For these specific concepts, it appears that the models need to *overfit before generalize*.

In a related vein, this phenomenon bears similarities to what is described as grokking (Power et al., 2022) or double descent (Nakkiran et al., 2021) in the literature. More compelling evidence for this comes from training LoHa with a dimension of 16 and an alpha value of 8. Here, the overfit-then-generalize effect is not confined to the "anime characters" category but becomes more pronounced across all classes. Initially, after 10 epochs of training, the model barely reacts to the prompts. However, its generalization capabilities show a significant improvement after 50 epochs of training. An illustration of this is given in Figure 40. It is also worth noticing that this behavior is consistent across all three runs trained with identical configurations but different random seeds, suggesting that this is not merely coincidental. Indeed, this trend is also manifested in our plots in Appendix F. In the 10-epoch plots, the point representing this configuration, denoted by a yellow triangle, consistently occupies the upper-left corner when comparing "image similarity <alter>" against "text similarity <alter>", while in the 50-epoch plots, this point shifts towards the lower-right corner.

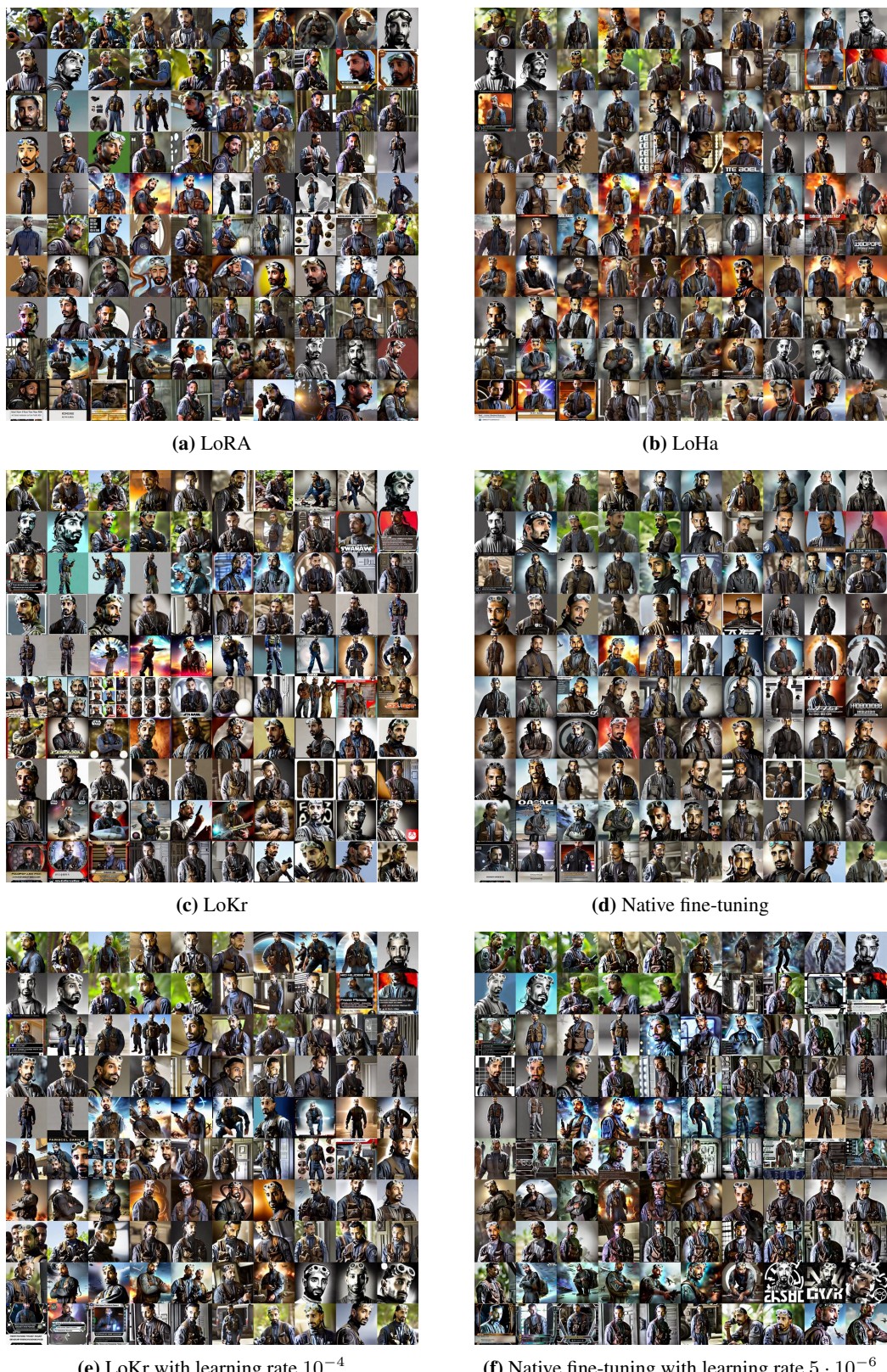

**Figure 33:** Synthetic images of "Bodhi Rook [realistic]" that are generated using the prompts of type <train>. The models are trained for 30 epochs and the default hyperparameters are used unless otherwise specified. See Appendix G.3.3 for the accompanying discussion.

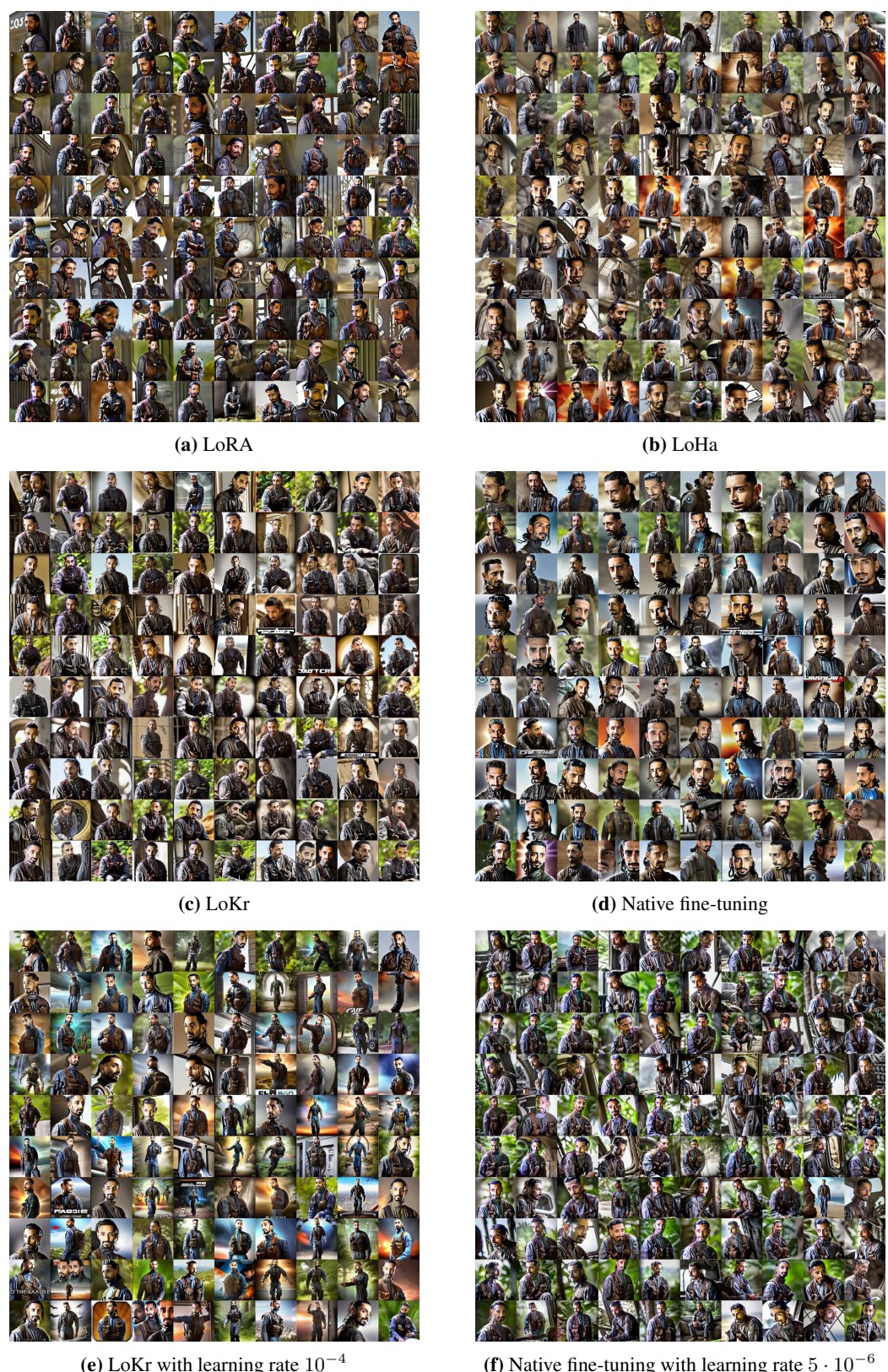

**(a)** LoRA

**(b)** LoHa

**(c)** LoKr

**(d)** Native fine-tuning

**(e)** LoKr with learning rate $10^{-4}$

**(f)** Native fine-tuning with learning rate $5 \cdot 10^{-6}$

**Figure 34:** Synthetic images of "Bodhi Rook [realistic]" that are generated using the prompts of type <trigger>. The models are trained for 30 epochs and the default hyperparameters are used unless otherwise specified. See Appendix G.3.3 for the accompanying discussion.

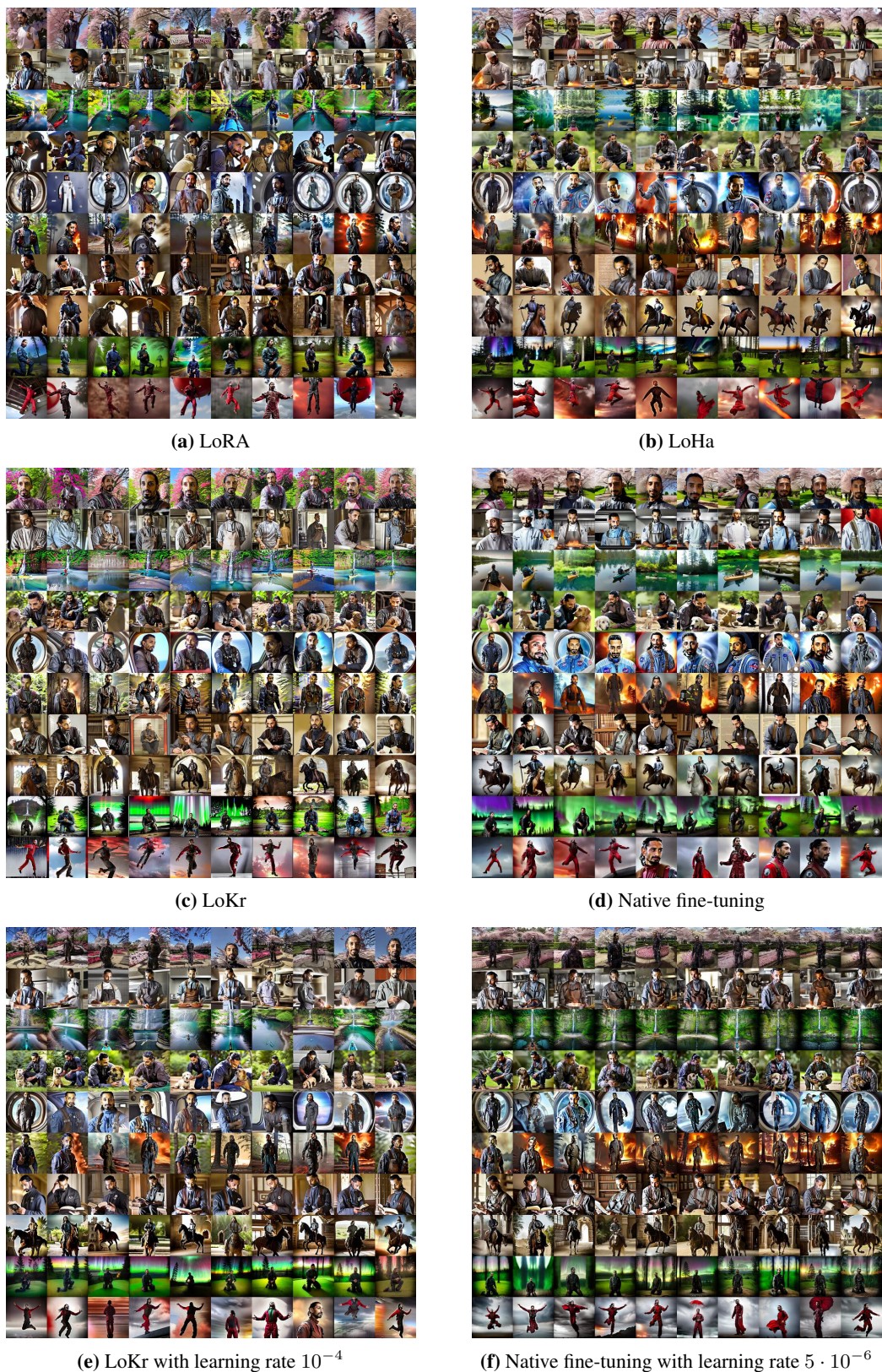

**(a)** LoRA

**(b)** LoHa

**(c)** LoKr

**(d)** Native fine-tuning

**(e)** LoKr with learning rate $10^{-4}$

**(f)** Native fine-tuning with learning rate $5 \cdot 10^{-6}$

**Figure 35:** Synthetic images of "Bodhi Rook [realistic]" that are generated using the prompts of type <alter>. The models are trained for 30 epochs and the default hyperparameters are used unless otherwise specified. See Appendix G.3.3 for the accompanying discussion.

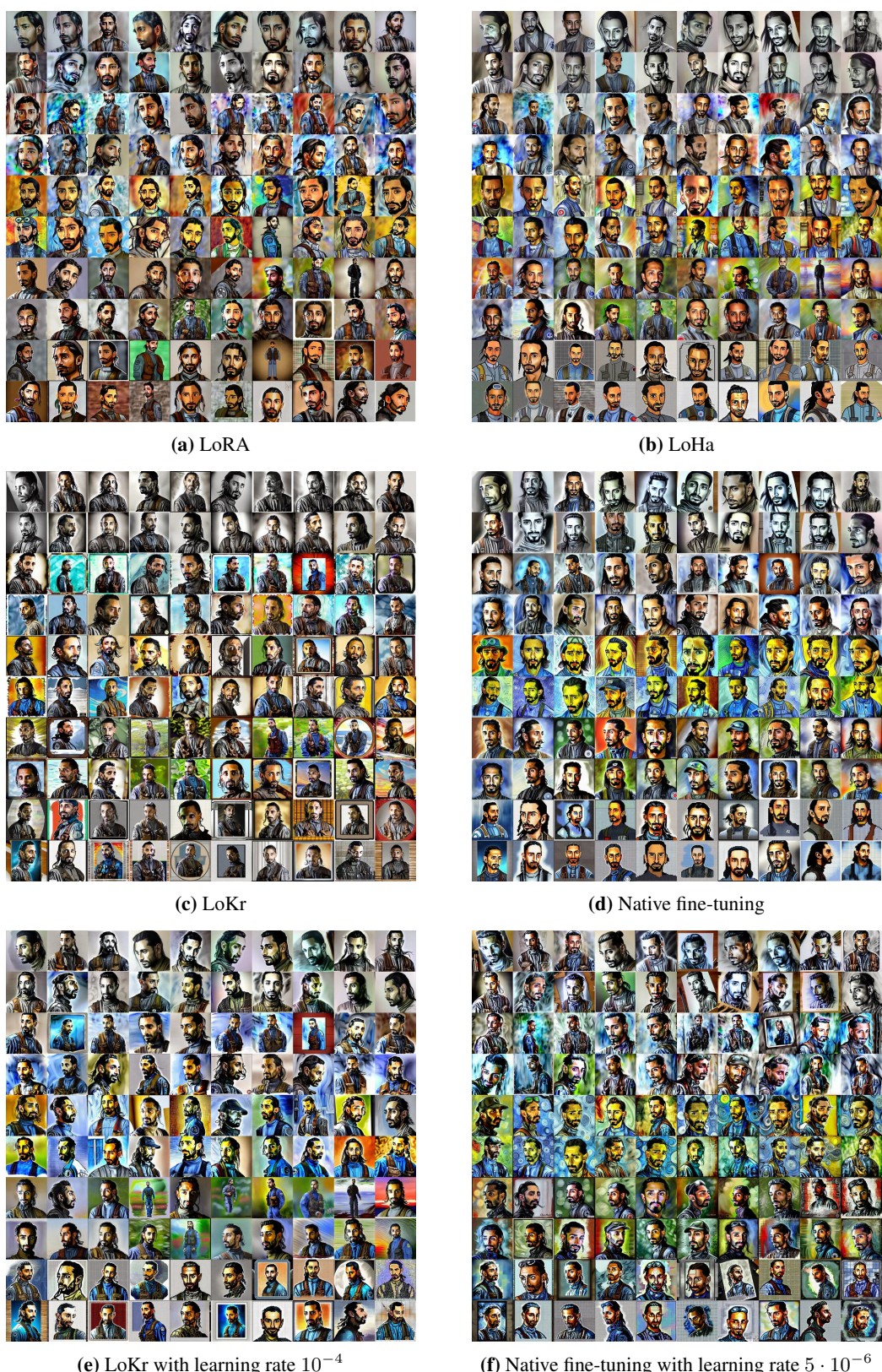

**(a)** LoRA

**(b)** LoHa

**(c)** LoKr

**(d)** Native fine-tuning

**(e)** LoKr with learning rate $10^{-4}$

**(f)** Native fine-tuning with learning rate $5 \cdot 10^{-6}$

**Figure 36:** Synthetic images of "Bodhi Rook [realistic]" that are generated using the prompts of type <style>. The models are trained for 30 epochs and the default hyperparameters are used unless otherwise specified. See Appendix G.3.3 for the accompanying discussion.

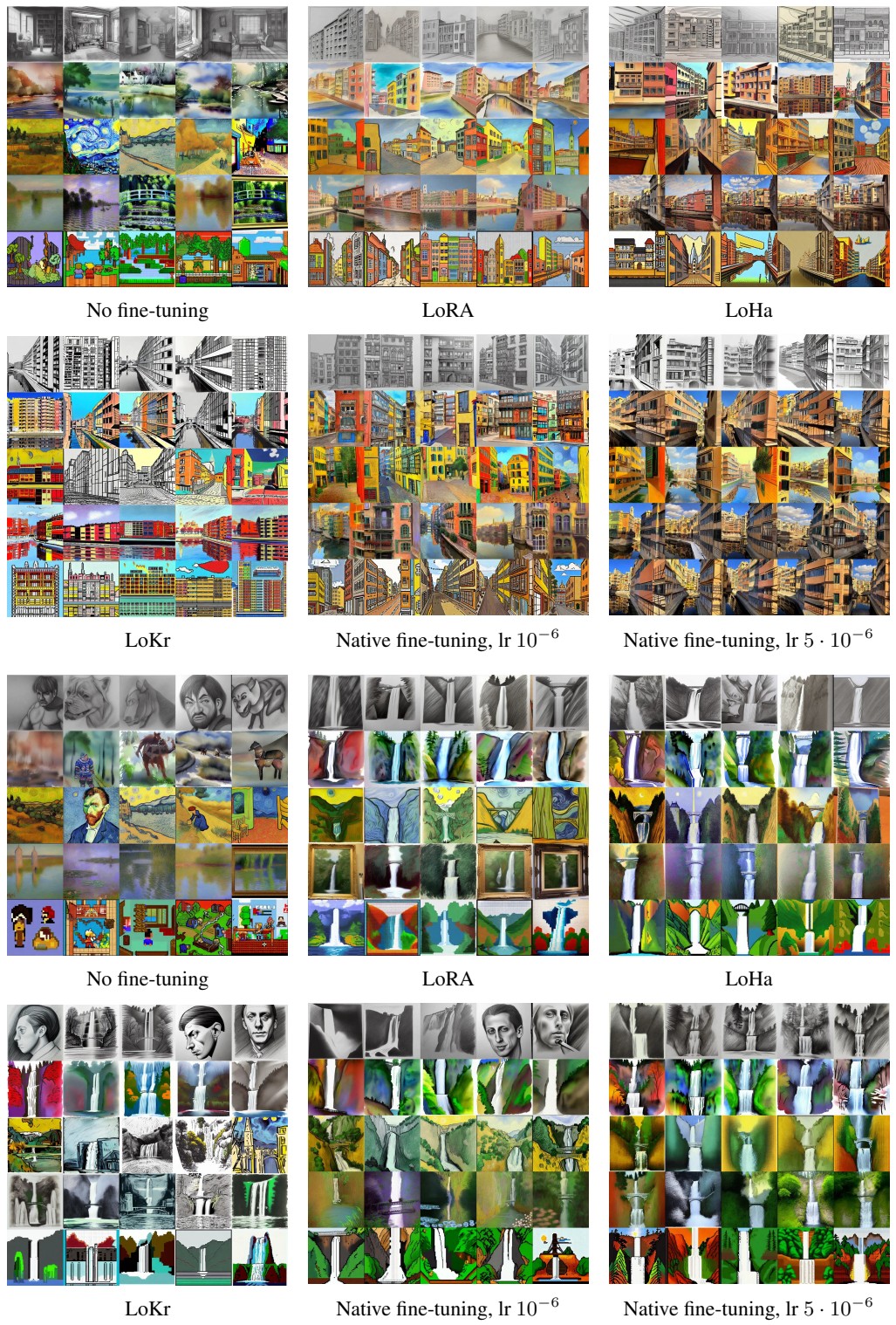

**Figure 37:** Example generations for classes "canal" and "waterfall" using the prompts of type <style> (cf. Appendix D.3). The models are trained for 10 epochs and the default hyperparameters are used unless otherwise specified. We note that LoRA seems to have the best style preservation here.

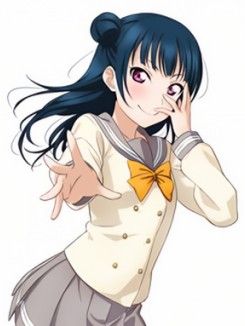

(a) Example image

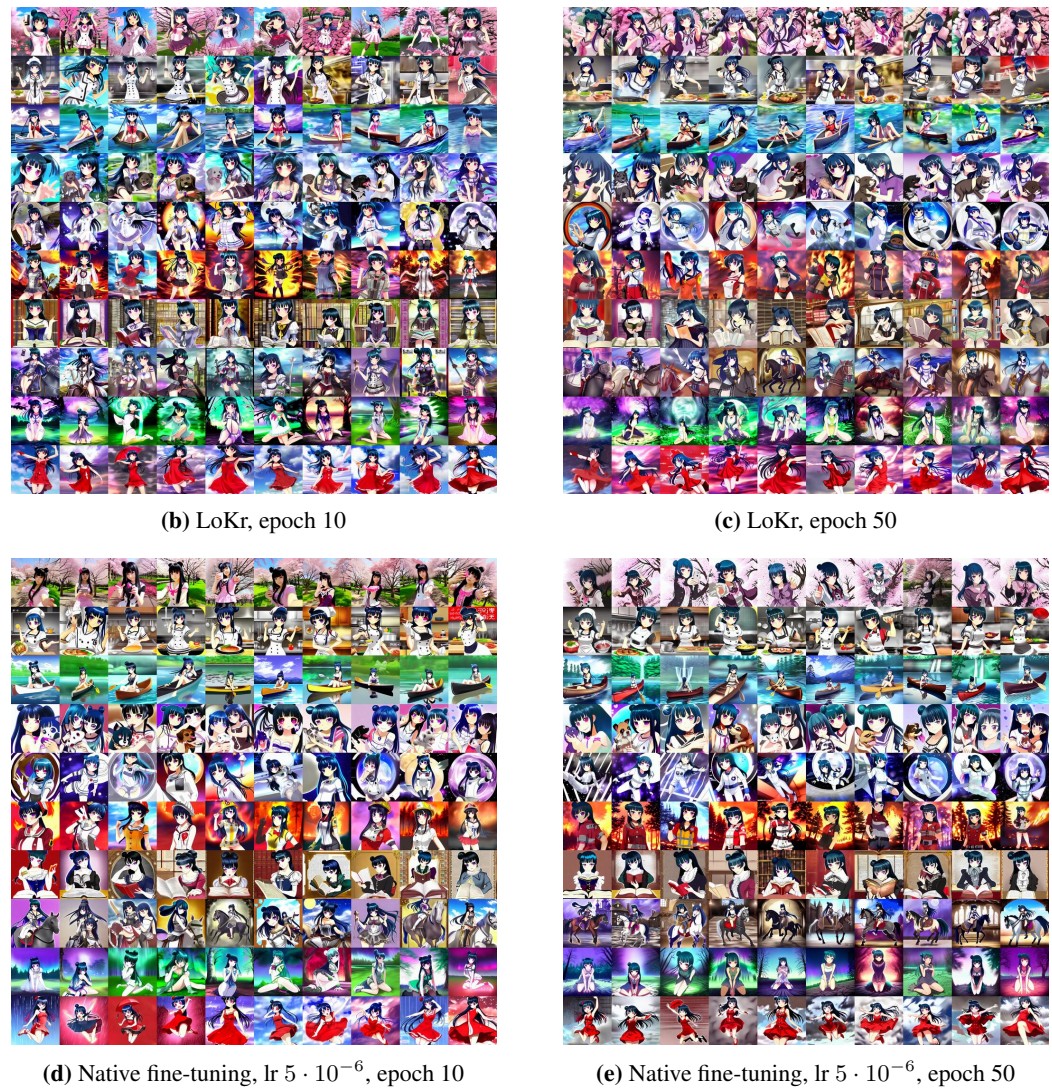

(b) LoKr, epoch 10

(c) LoKr, epoch 50

(d) Native fine-tuning, lr $5 \cdot 10^{-6}$, epoch 10

(e) Native fine-tuning, lr $5 \cdot 10^{-6}$, epoch 50

**Figure 38:** Example image and generated samples for "Tsushima Yoshiko". Prompts of type <alter> are used for the generations (cf. Table 6). We observe improved text-image alignment over training.

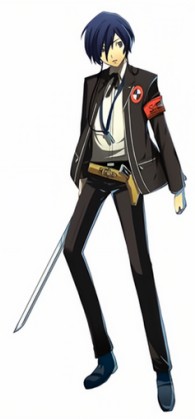

(a) Example image

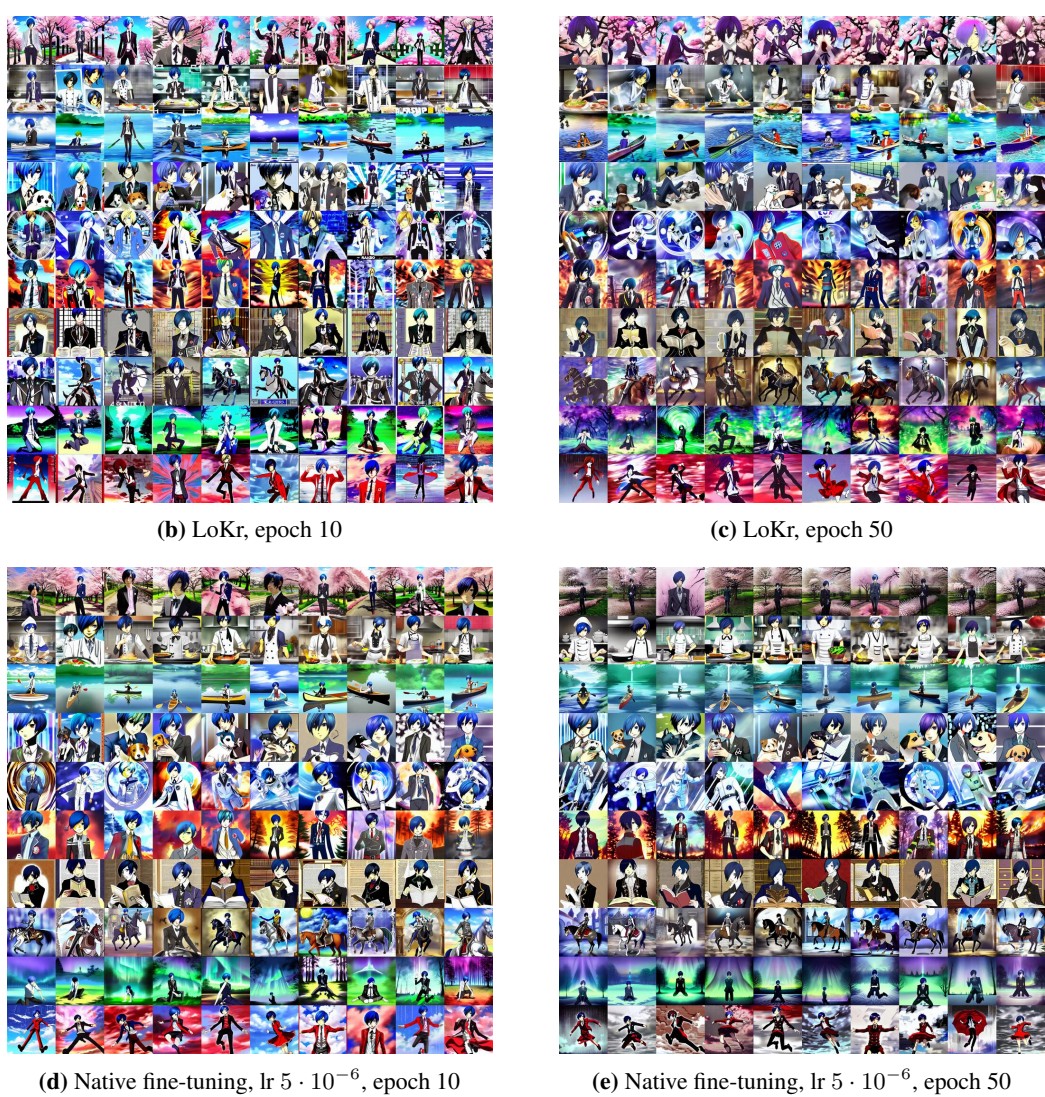

(b) LoKr, epoch 10

(c) LoKr, epoch 50

(d) Native fine-tuning, lr $5 \cdot 10^{-6}$, epoch 10

(e) Native fine-tuning, lr $5 \cdot 10^{-6}$, epoch 50

**Figure 39:** Example image and generated samples for "Yuuki Makoto". Prompts of type <alter> are used for the generations (cf. Table 6). We observe improved text-image alignment over training.

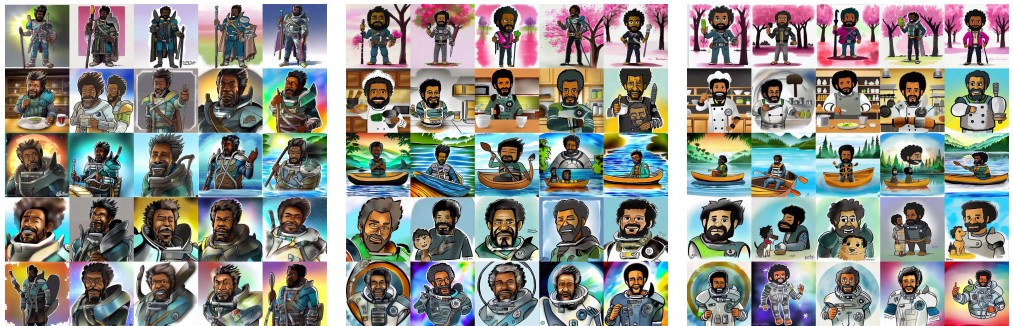

**(a)** Generations with prompts of type <alter>

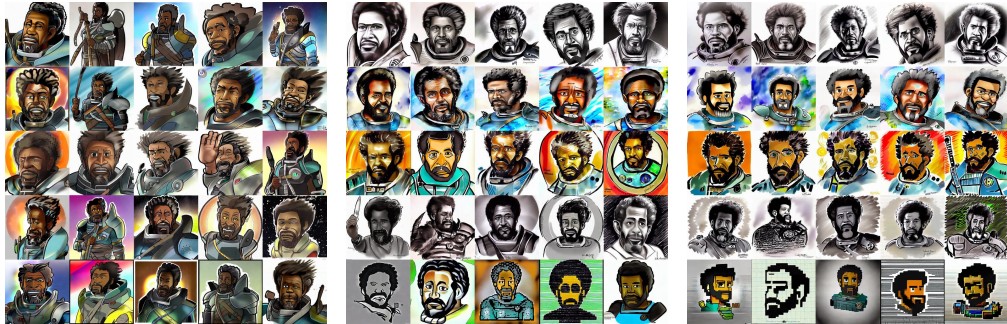

**(b)** Generations with prompts of type <style>

**Figure 40:** Example generations for sub-class "Saw Gerrera [afro, illustration]" using a LoHa with dimension 16 and alpha 8. From left to right we use respectively the epoch 10, 30, and 50 checkpoints. We observe that both text-image alignment and style preservation improve over training.

### G.4.3 RICHER STYLISTIC VARIATIONS WITH INCREASED MODEL CAPACITY

While our general guidelines propose that increasing a model's capacity—while keeping other hyperparameters constant (for alpha, we maintain the alpha/dimension ratio)—tends to diminish the model's ability to preserve the base model's style, Figure 13 for the "stuffed toy" category suggests otherwise. To better understand this apparent discrepancy, we delve deeper into this issue in Figure 41. For LoRA, there is no evidence for such improvement in base style preservation when increasing dimension and alpha. However, for LoKr, the improvement may be real. At least, LoKr with smaller factors (and hence larger capacity) seems to generate more stylistically rich images with <style> prompts, though whether these images authentically align with the styles specified in the prompts is subject to discussion. One plausible reason for the emergence of more stylistic images with larger models could be the presence of various styles in our dataset, which the model then incorporates during the fine-tuning process.

### G.5 EXAMPLE GENERATIONS FOR CATEGORY "STYLES"

For the sake of illustration, we present a number of generated samples for the "styles" category in Figures 43, 45 and 47. We note that different styles need different number of training epochs and model capacity to be learned properly. Moreover, when the dataset only depicts a specific type of object, as in the class "Felix Valloton", increasing dimension and alpha of LoRA or LoHa can be harmful for these models' performance on generalization prompts (cf. Figure 47). Conversely, these changes can be beneficial for the learning of other styles, as shown in Figure 43.

Generally speaking, the trend observed in these figures align with the guiding principles we outlined in Section 5.3. Although Figure 19 suggest that both image and style similarities decrease over training for these classes, we find no evidence of this. One potential explanation for this decrease could then be that the images become "oversaturated" in certain style classes as training progresses, as illustrated in Figure 45.

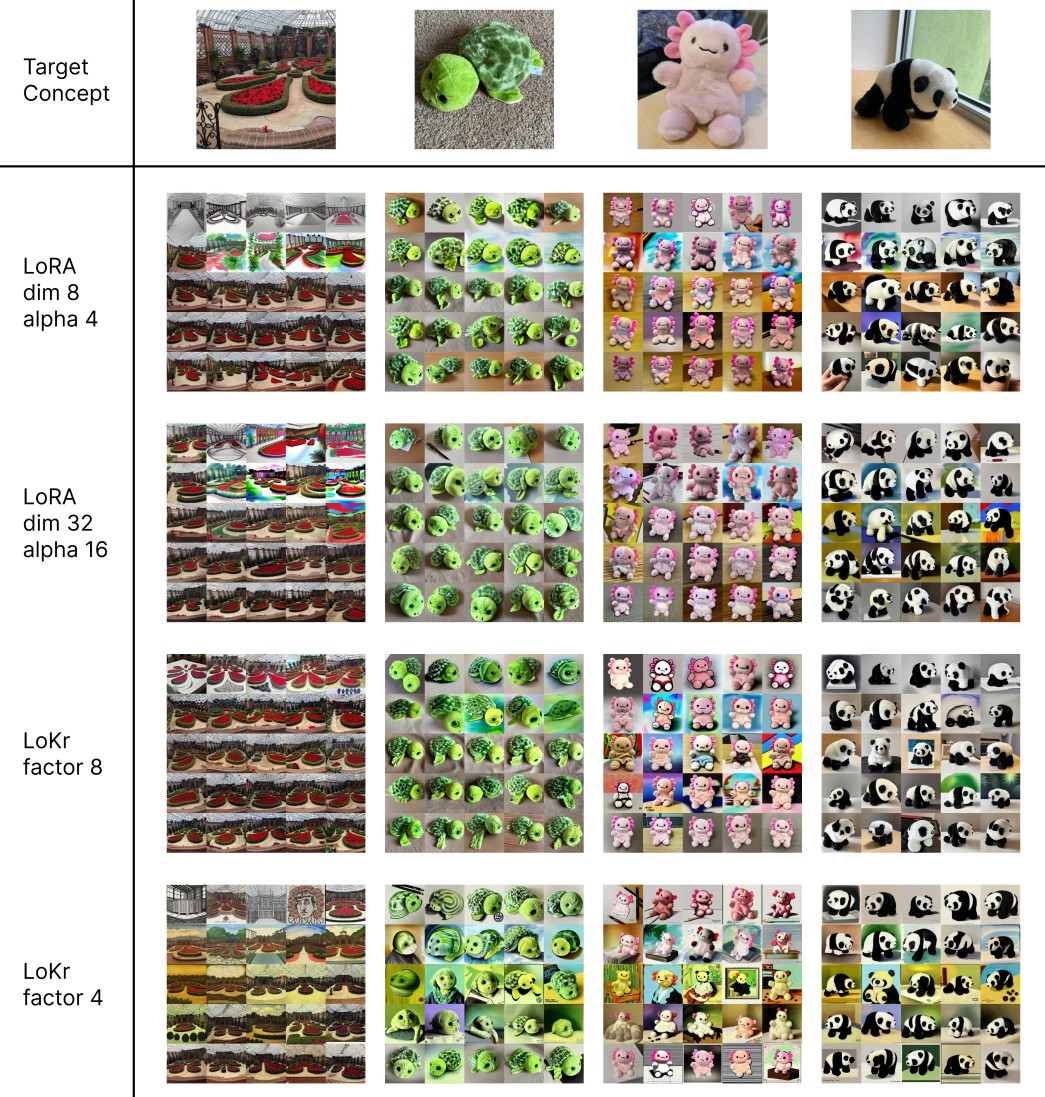

**Figure 41:** Generated images from four models trained with different configurations. The prompts of type <style> are used here (cf. Appendix D.3). Note that increasing model capacity enhances LoKr's ability to generate more stylistically rich images.

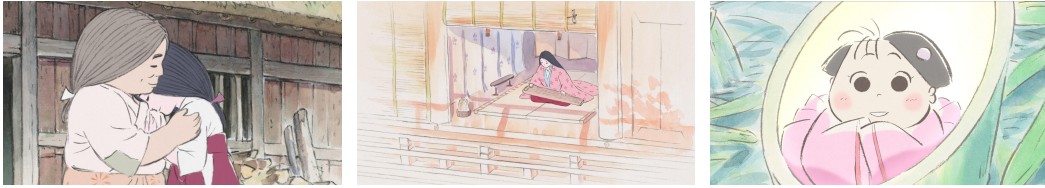

**Figure 42:** Example training images for class "Ghibli 2".

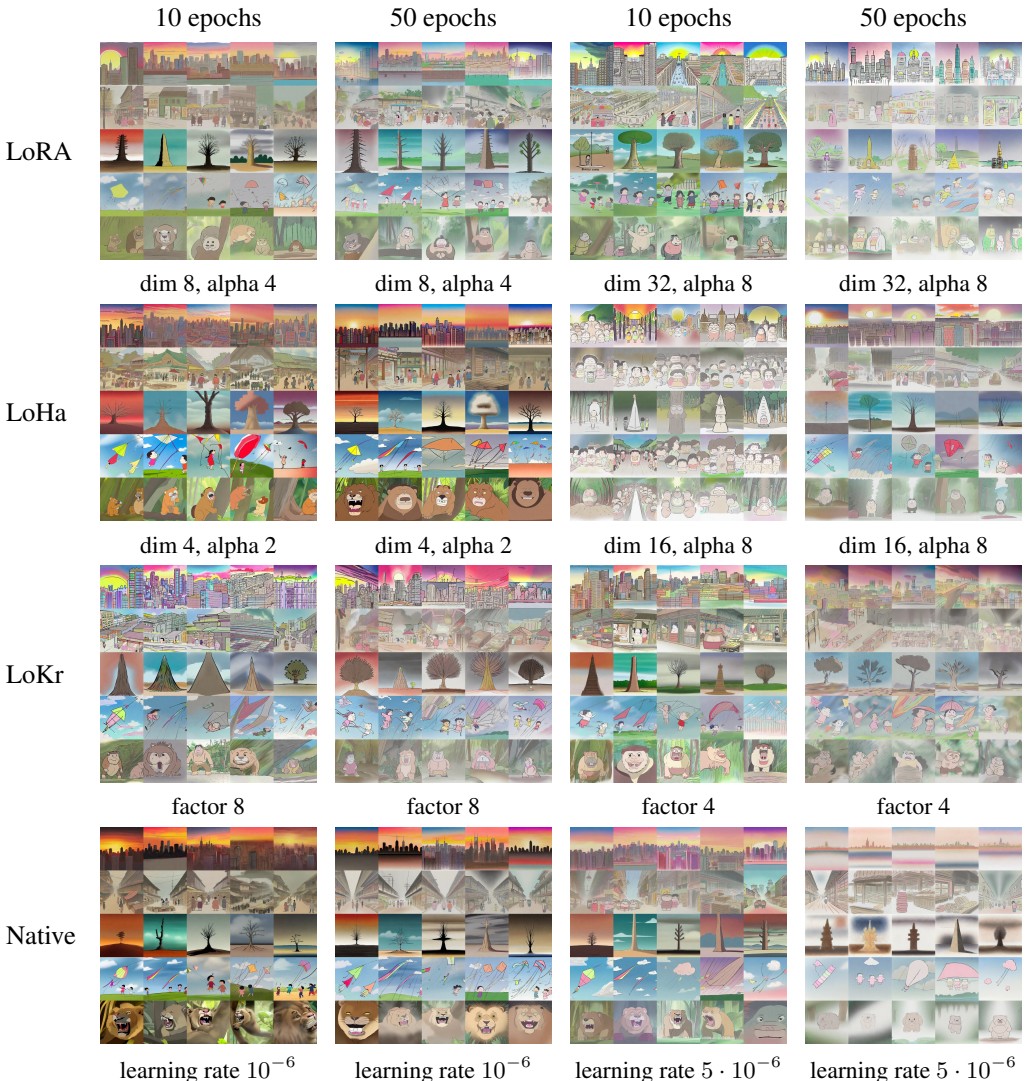

**Figure 43:** Example generations for "Ghibli 2" from different checkpoints. The first 5 prompts of type <alter> (cf. Table 6) are used to generate these images.

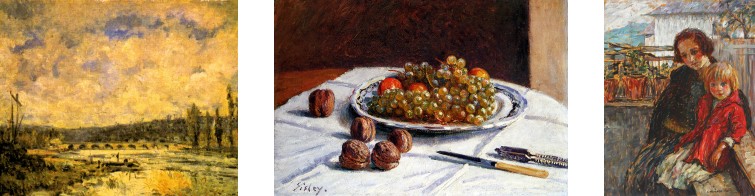

**Figure 44:** Example training images for class "impressionism".

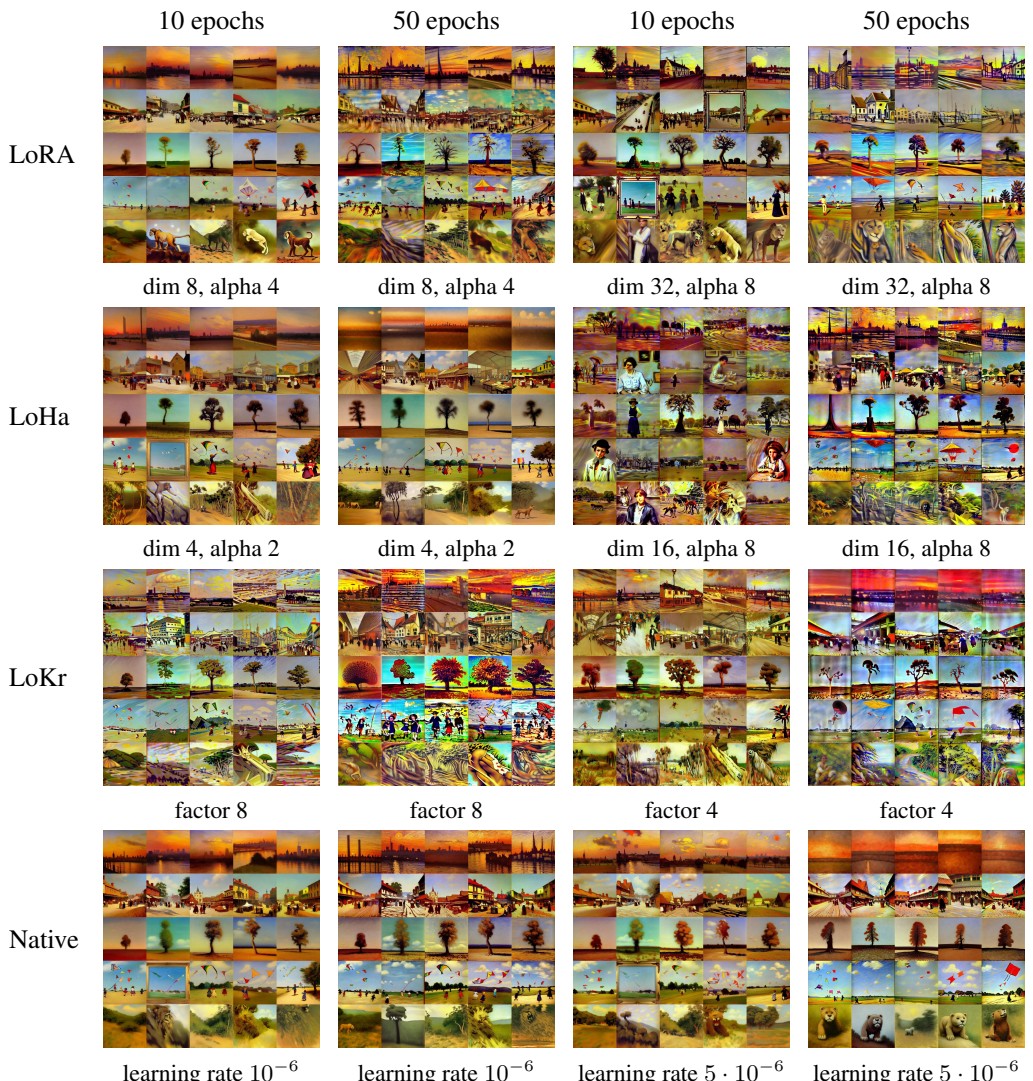

**Figure 45:** Example generations for "impressionism" from different checkpoints. The first 5 prompts of type <alter> (cf. Table 6) are used to generate these images.

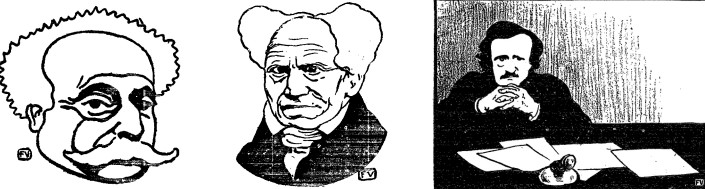

Figure 46: Example training images for class "Felix Vallotton".

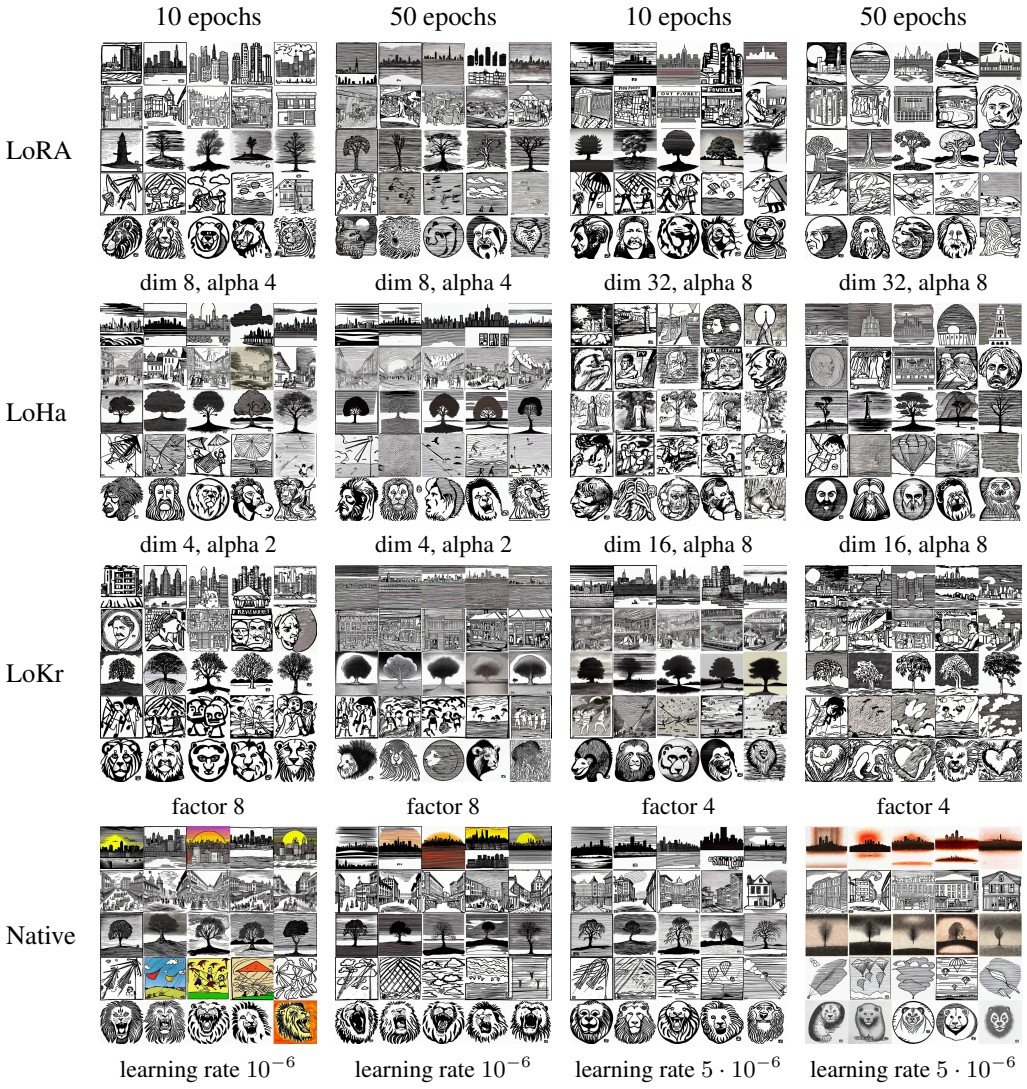

Figure 47: Example generations for "Felix Vallotton" from different checkpoints. The first 5 prompts of type <alter> (cf. Table 6) are used to generate these images.

# H   Additional Experiments

This appendix presents a number of additional experiments that have been omitted from the main text. These experiments validate some design choices that we have made for our main study, and provide further insights into fine-tuning of text-to-image models.

## H.1   INVESTIGATING THE RELEVANCE OF IMAGE FEATURES

The primary objective of this section is to investigate how different image features focus on distinct aspects when measuring similarity between images. This serves as a critical sanity check to shed light on why different encoders are preferable for calculating style loss versus semantic similarity.

**Encoders and Resizing Methods.**   Precisely, we compare a number of ways to extract image features which mainly differ in the choice of encoders and resizing methods.

- **Encoders:** We consider four distinct encoders: DINOv2, CLIP, ConvNeXt V2, and VGG-19, as explained in Appendices D.4 and D.5. For VGG-19, the features are generated by concatenating the flattened normalized Gram matrices that are involved in the computation of the style loss.
- **Resizing Methods:** Four different resizing techniques are considered: scale, letterbox, center crop, and stretch. These are applied to DINOv2, CLIP, and ConvNeXt V2 as discussed in Appendix E. For VGG-19, we only consider the scale method as it can take non-square inputs.

In sum, we explore a total of 10 distinct encoding methods, each representing a unique combination of an encoder and a resizing technique.

**Datasets.**   Our experiments make use of the following three classification datasets.

- **ImageNet100:** A subset of ImageNet (Russakovsky et al., 2015) containing 100 distinct classes, comprising a total of 130,000 images.[14]
- **DAF:re-250:** A subset of DAF:re (Rios et al., 2021) with 250 classes, featuring 99,361 images.
- **Style30:** An expanded version of the new WikiArt dataset (Tan et al., 2019), enriched with three additional style classes, summing up to 30 classes and 112,349 images. Notably, the "anime" class in this dataset contains artworks from 279 different artists and includes a total of 29,830 images. Each of these artists' work can be further considered as a unique style.

By considering datasets with varied image types and classification criteria, we would like to dissect how changes in either the image's style or content affect the distribution of different image features.

**Diversity Ratio.**   Building upon the above, for each encoding method we compute the *diversity ratio* defined as $\text{diversity}_{\text{class}}/\text{diversity}_{\text{dataset}}$. It compares the intra-class feature diversity against the overall dataset diversity. Then, for example, low diversity ratios in our style dataset would imply that the features in question are particularly sensitive to stylistic changes, while high diversity ratios in other classification tasks might suggest their insensitivity to changes in the subject of the images.

Nonetheless, it still remains the question of how we evaluate feature diversity. For this, we consider three different metrics as listed below.

- **Vendi Score:** This follows the definition given in Appendix D.4.
- **Intra-Dissimilarity:** For a set of features $\mathcal{Z}$, its intra-dissimilarity is computed as $1 - S_C(\mathcal{Z}, \mathcal{Z})$ where $S_C$ is the average cosine similarity metric defined in (22). This measure is directly related to the use of cosine similarity to assess the similarity of two images.
- **Variance:** We also consider the variance of the feature set. This is directly related to the use of Euclidean distance to assess the similarity of two images, as what we do in computing style loss.

It is worth noticing that the intra-dissimilarity is nothing but the variance of the normalized vectors, as can be seen from (23). Therefore, in reality, the only difference between the second and the third metrics boils down to whether the feature vectors are normalized or not.

**Result.**   We compute the diversity ratios for the three datasets, as well as for the anime class within the style dataset, putting each artist's work in a separate sub-class. Moreover, we only compute

---

[14]https://www.kaggle.com/datasets/ambityga/imagenet100 (Accessed: 2023-08-20)

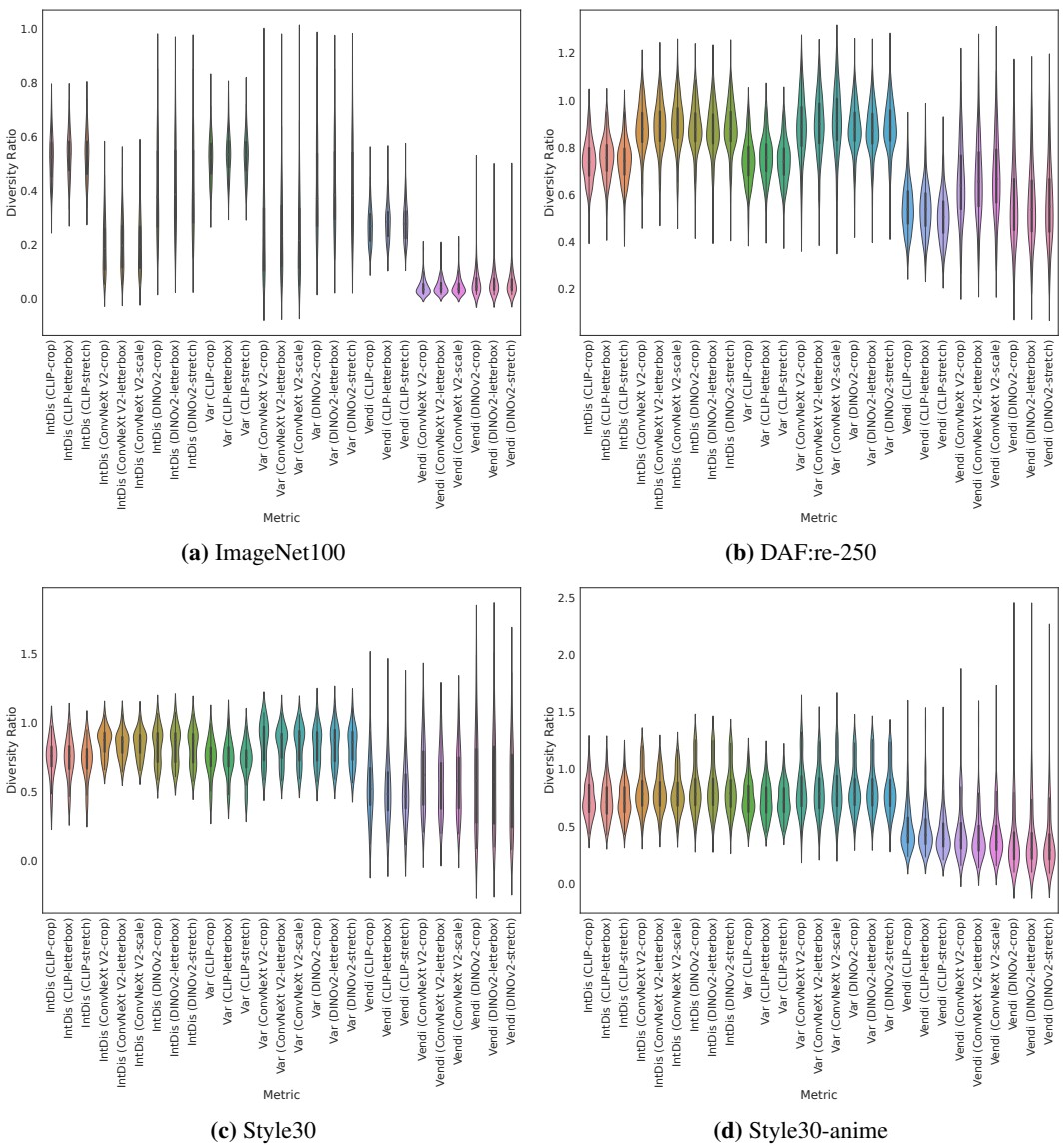

**Figure 48:** Distribution of diversity ratios across classes for different metrics/encoding methods. IntDis and Var respectively stand for intra-dissimilarity and variance. We observe that the choice of resizing methods has little influence on the results.

diversity scores for classes with more than 50 images, and whenever we need to compute the diversity score for a set of more than 1,000 images, we sub-sample it to a fixed size of 1,000 before performing the computation. The distributions of the diversity ratios obtained in our experiments are shown in Figures 48 and 49.

In Figure 48, we see that the choice of resizing method has little influence on the results. This is consistent with what we have observed for our main experiments in the correlation analysis of Appendix E. We next zoom in on the influence of encoders and metrics in Figure 49. We first observe that the distributions of diversity ratios for intra-dissimilarity and variance are relatively close. For these two metrics, using the VGG-19 features gives lower diversity ratios for style datasets and higher diversity ratios for ImageNet100 and DAF:re-250, suggesting that VGG-19 features are indeed the most suitable for evaluating style similarity among all the four features that we consider here. Intriguingly, this observation does not hold true when we compute the diversity ratio using the Vendi score. The reason behind this discrepancy warrants further investigation. Finally, we generally

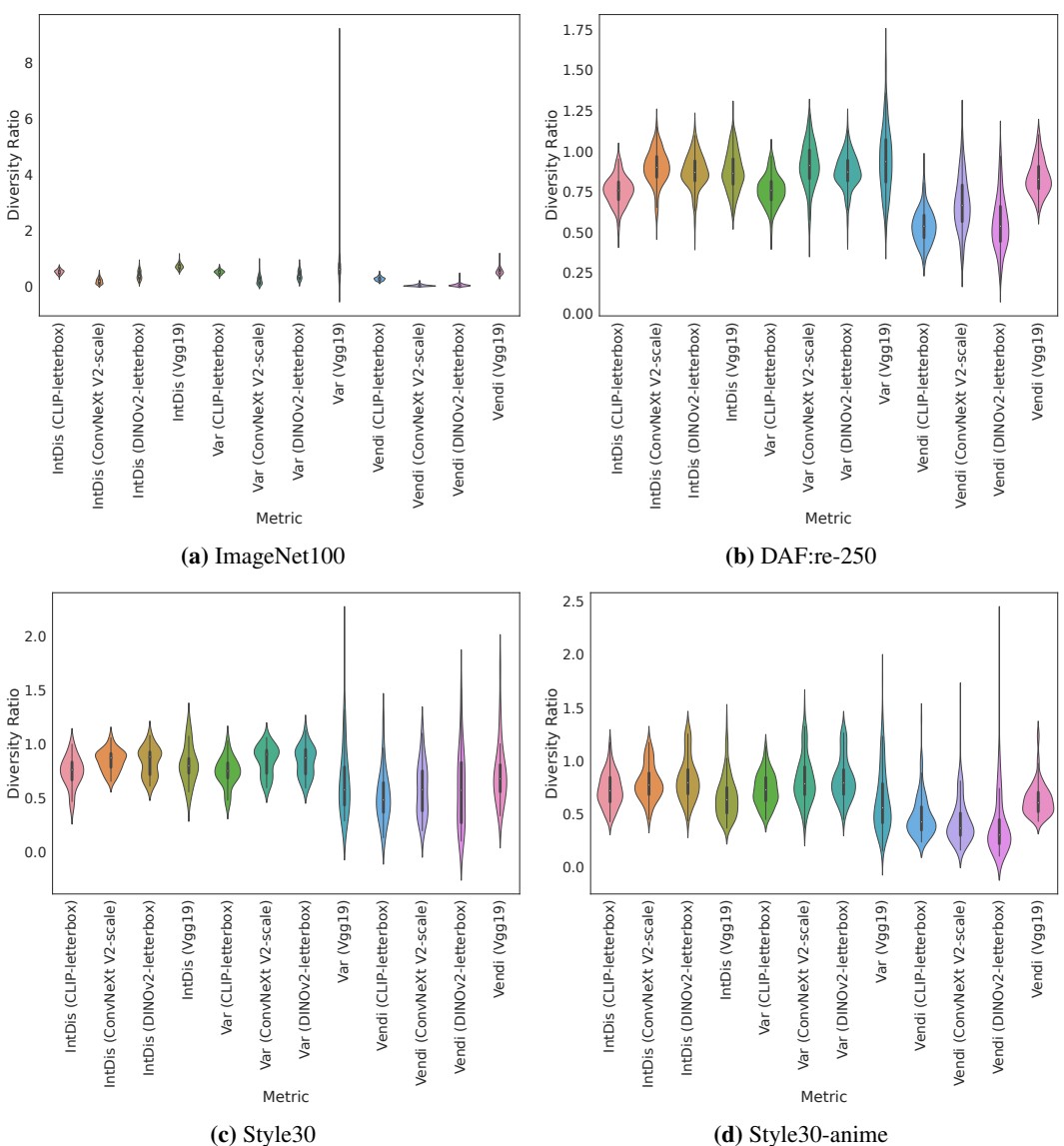

**(a)** ImageNet100

**(b)** DAF:re-250

**(c)** Style30

**(d)** Style30-anime

**Figure 49:** Distribution of diversity ratios across classes for different metrics/encoding methods. IntDis and Var respectively stand for intra-dissimilarity and variance. Only one resizing method is chosen for each encoder here.

observe a lower diversity ratio when Vendi score is used to compute diversity. This does suggest that the Vendi score can better discern datasets of various degrees of diversity.

## H.2 IMAGE QUALITY ASSESSMENT WITH PRETRAINED MODELS

Evaluating the quality of images generated by deep generative models is itself a challenge. In this appendix, we show that existing state-of-the-art image quality assessment (IQA) models may not be suitable for this task due to distribution shift. Specifically, We examine three leading IQA models: LIQE (Zhang et al., 2023), MANIQA (Yang et al., 2022), and a publicly available artifact scorer trained on the AI Horde ratings dataset (Haidra-Org, 2023; Wang et al., 2022).[15]

In more detail, we use the MANIQA model pretrained on the KONIQ-10K dataset (Hosu et al., 2020), and the artifact scorer that uses the `openclip_vit_l_14` features. For LIQE and MANIQA, multiple

---

[15]The artifact scorer is available at `https://github.com/kenjiqq/aesthetics-scorer` (Accessed: 2023-08-20).

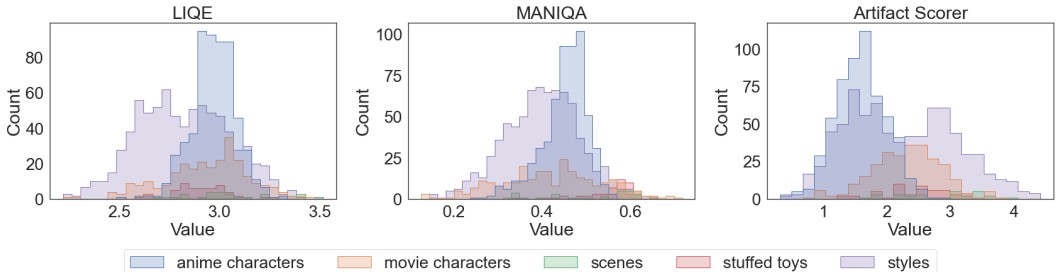

**Figure 50:** Distributions of the image quality scores on the training dataset. We note an important bias in these scores that favor or disfavor images of certain categories.

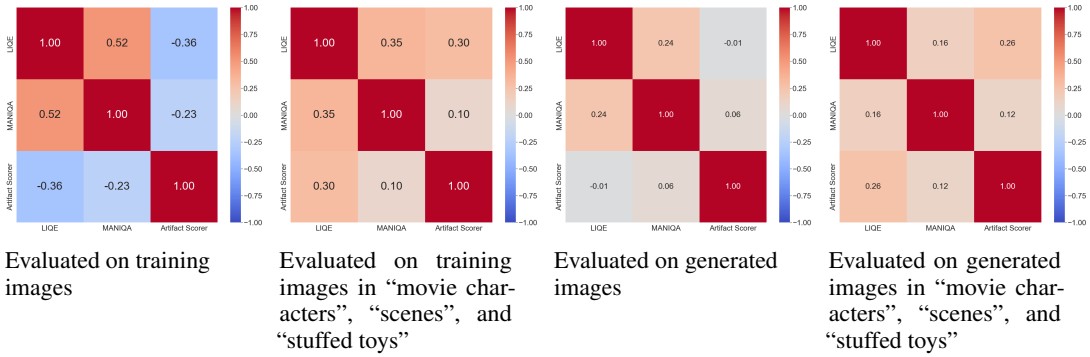

| Evaluated on training images | Evaluated on training images in "movie characters", "scenes", and "stuffed toys" | Evaluated on generated images | Evaluated on generated images in "movie characters", "scenes", and "stuffed toys" |
|---|---|---|---|

**Figure 51:** Correlation matrices of the considered image quality scores. We compute the correlation coefficients for scores obtained on different sets of images.

patches of size $224 \times 224$ need to be extracted from the images to estimate the scores. For this, we first resize the images so their shortest edges have 512 pixels, and then we extract 10 patches from each image. As for the artifact scorer, we resize the images to resolution $224 \times 224$, with black padding added to the images if necessary (i.e., lettebox resizing).

Importantly, among the three models, only the artifact scorer is specifically made to evaluate AI-generated images, which is why we have included it in our study. Also, it is worth mentioning that while LIQE and MANIQA assign higher scores to better-quality images, the artifact scorer does the opposite. It gives a score between 0 and 5, where a higher score signifies more artifacts in the image. To make our results easier to compare, we thus convert this to "5 minus the original score" below.

**Results.** With these three IQA models, we evaluate the quality of both the images from our training set and those generated using training prompts and a 10th-epoch LoRA checkpoint trained with default hyperparameters. Our observations are as follows:

- **Style-Related Bias:** As shown in Figure 50, we find the image quality scores predicted by the models vary based on the style of the images. In particular, LIQE and MANIQA tend to give higher scores to anime-style images whereas the artifact scorer tends to give lower scores to them. For LIQE and MANIQA, we believe this is because they are trained exclusively on natural images, thereby limiting their predictive accuracy for artworks. In the case of the artifact scorer, the observed bias likely stems from inherent biases in the AI Horde ratings dataset. Such biases undermine the credibility of these IQA models, as image quality should ideally be style-agnostic.

- **Disagreement Among Models:** The correlation coefficients of the scores computed by different models are illustrated in Figure 51. Given the biases previously mentioned, we analyze the correlations in two distinct contexts: first across all training or considered generated images, and then excluding images from the "anime characters" and "styles" categories. In all the scenarios, we observe that the models exhibit only a weak correlation in their predicted scores, indicating a lack of agreement among the predictions made by these models.

- **Inability to Distinguish Real from Generated Images:** In Figure 52, we contrast the quality scores assigned to dataset images with those given to generated images. The plots reveal that none

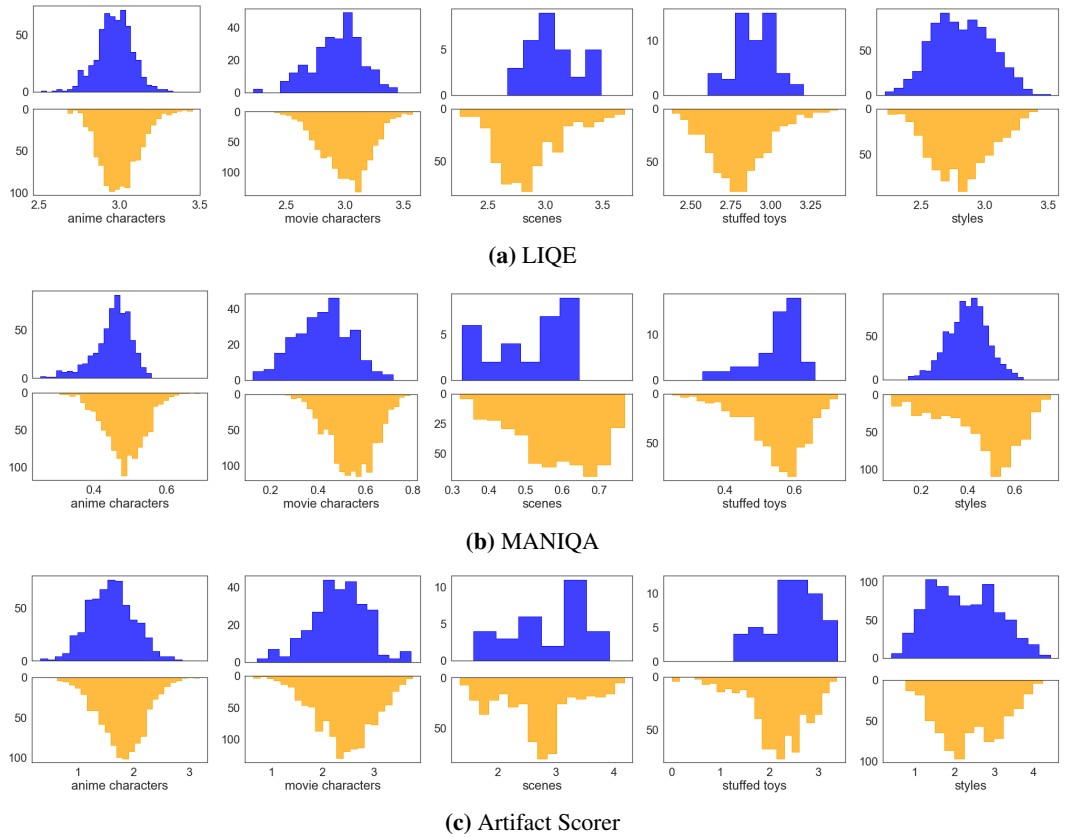

**(a)** LIQE

**(b)** MANIQA

**(c)** Artifact Scorer

**Figure 52:** Distributions of the image quality scores on training (top, blue) and generated (bottom, yellow) images. We observe that these scores often fail to attribute higher scores to real images.

of the three models consistently award higher scores to real images. More surprisingly, MANIQA often assigns higher scores to generated images, even when these contain noticeable artifacts.

In summary, the above observations cast doubt on the reliability of the three IQA models under consideration in evaluating the quality of AI-generated images. As a consequence, we have opted not to include them in our primary experiments.

### H.3 IMPACT OF CAPTIONING STRATEGIES ON MODEL PERFORMANCE

In this section, we explore the influence of captioning strategies on the performance of fine-tuned models. Specifically, we focus on three captioning strategies:

- **No Tags:** This approach follows the common practice in existing literature, using brief captions like "`a photo of [V]`" or "`an illustration of [V]`".
- **All Tags:** In this setup, we append all tags predicted by the employed tagger to the concept descriptor, thereby creating richer captions.
- **Adjusted Tags:** This strategy is the one employed in our main experiment, wherein the tags are manually adjusted after the initial tagging phase. See Appendix D.1 for more details.

For the sake of simplicity, in the following, we just consider LoRA, LoHa, and LoKr trained with default hyperparameters, and native fine-tuning with learning rate $5 \times 10^{-6}$ as our main experiment suggests that this consistently leads to better results than using the default $10^{-6}$ learning rate.

The image generation, metric computation, and metric processing procedures follow those employed in our main experiment. It is however important to note that when it comes to rank-based normalization of the metrics, we only compare with the checkpoints studied in this section. We report quantitative and qualitative results across these captioning strategies respectively in Figures 53 and 54.

**Impact on Concept Fidelity, Controllability, and Base Style Preservation.** A closer look at Figure 53 reveals the following trade-offs between different captioning strategies.

- **With no tags:** Although the use of simple caption seems to enhance concept fidelity, it comes at the expense of both controllability and base style preservation.[16]
- **With all tags:** Conversely, including all tags improves controllability and base style preservation but sacrifices concept fidelity. This compromise occurs because each component of the target concept is automatically mapped to the tag that most closely describes it. Without manual adjustments to these tags, the concept becomes fragmented across its various components. As a result, the concept descriptor captures only a partial and limited aspect of the target concept.

These contrasting effects are demonstrated in Figure 54. The leftmost figures of the first and third rows show that models trained on captions devoid of tags struggle to appropriately respond to prompts. On the other hand, in the rightmost figure of the first row, it is shown that a model trained with all the tags produced by the tagger fails to accurately capture the hairstyle and the uniform of the character. Even worse, in the rightmost figure of the second row, the sculpture, which should be the focus of the concept, is completely absent. Instead, the concept descriptor is associated with the background. This misdirection is likely due to the consistent presence of the tag "Christmas tree" in the training captions for this specific class, causing the model to associate the sculpture with "Christmas tree" rather than with the intended concept descriptor.

The above findings also validate our choice of using adjusted tags in our main experiment. This approach strikes a delicate balance between the extremes observed in the "no tags" and "all tags" strategies, ensuring degrees of controllability without sacrificing concept integrity.

**Impact on Diversity.** Interestingly, Figure 53 also reveals that both alternative captioning strategies improve image diversity compared to the default "adjusted tags" approach. However, we believe this occurs for two distinct reasons. For the "no tags" strategy, this is mostly because the concept descriptor unintentionally captures additional elements from the training set, leading to a broader, albeit less accurate, interpretation of the target concept. In contrast, for the "all tags" strategy, the diversity seems to stem from the fact that the concept descriptor only captures a limited part of the target concept, allowing the model greater freedom in generating more diverse images.

In the meantime, we observe that in the "stuffed toys" category, the "all tags" strategy not only improves diversity but also enhances image similarity for images generated using solely the concept descriptor. This likely happens because our manually adjusted tags better encapsulate the image backgrounds compared to the predicted tags, resulting in models generating images with more "neutral" backgrounds when only the concept descriptor is involved in the prompts. On the other hand, when using the tags produced by the tagger, details from the training set's background leak into the learned concepts, contributing to both higher similarity and diversity. This phenomenon is further illustrated in the last row of Figure 54.

---

[16] An exception to this general trend is observed in the case of LoHa, where base style preservation actually improves with simple captions, potentially due to the specific phrasing "`A ... of [V]`" used in this case.

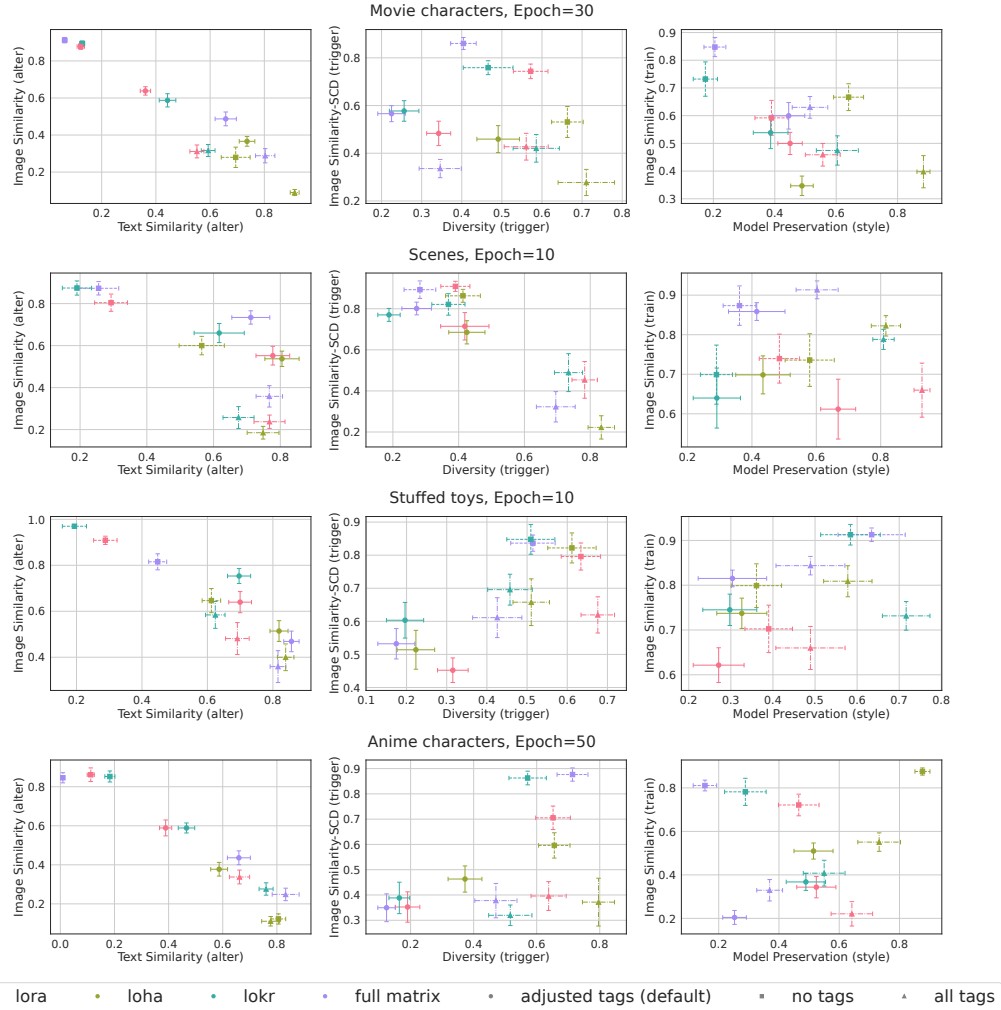

**Figure 53:** Scatter plots comparing different evaluation metrics with variations across algorithms and captioning strategies.

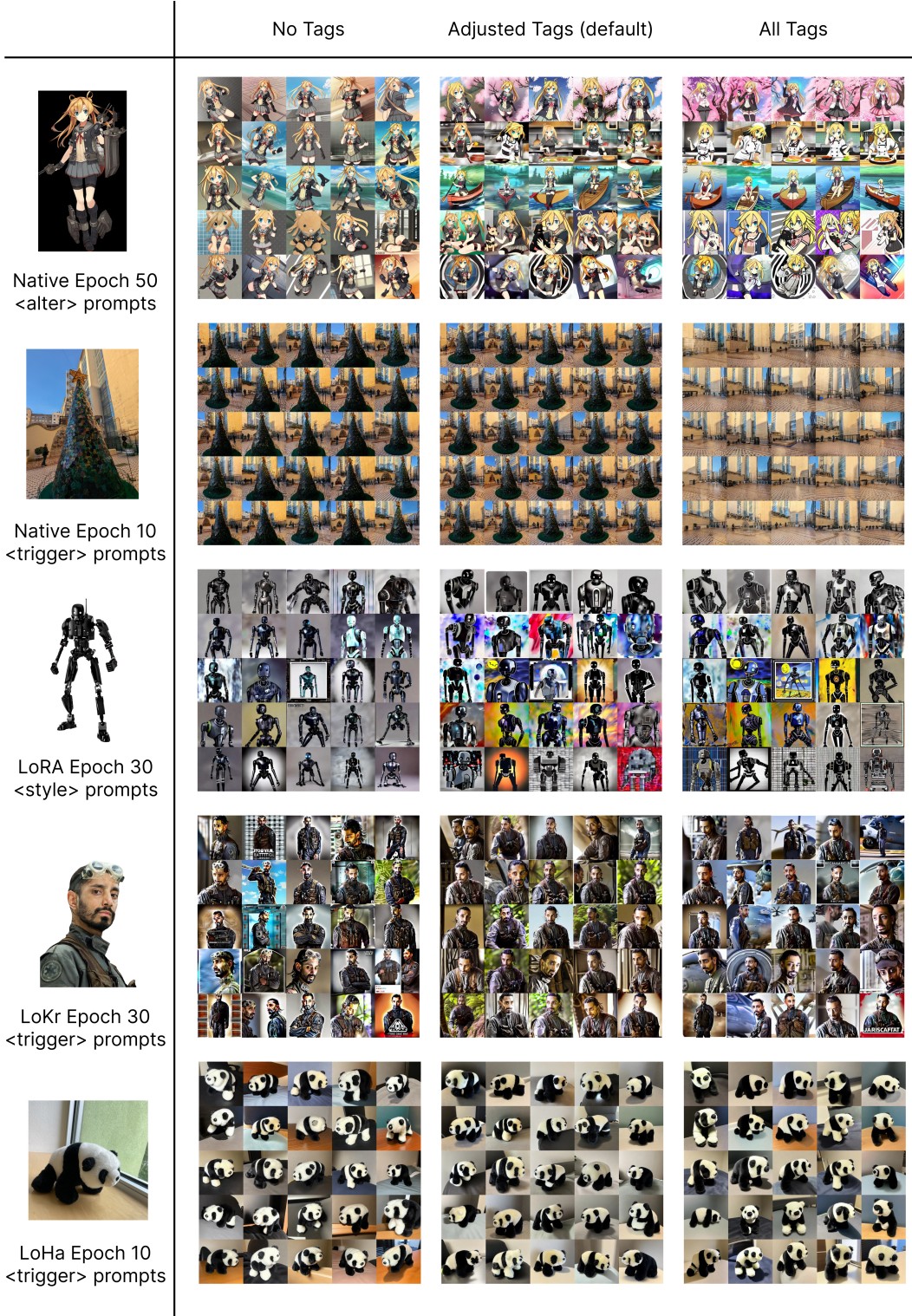

**Figure 54:** Generated images from models trained with different sets of captions. We observe that models trained with short captions without any further description of the images lack flexibility while training with unpruned tags could cause the target concept to be associated with the tags instead of the concept descriptor. Readers are referred to Appendix D.3 for details on the evaluation prompts.

# I   Author Contributions

The contributions of each author are summarized in Table 7.

|  | S.-Y. Yeh | Y.-G. Hsieh | Z. Gao | B. B W Yang | G. Oh | Y. Gong |
|---|---|---|---|---|---|---|
| Algorithm Conception | X |  | X |  | X |  |
| Library Development | X |  |  |  |  |  |
| Evaluation and Experiments |  | X | X |  |  |  |
| Writing | X | XX | X | X |  | X |

**Table 7:** Author contributions.