# OpenReview forum: "Navigating Text-To-Image Customization: From LyCORIS Fine-Tuning to Model Evaluation"
_ICLR.cc/2024/Conference — ICLR 2024 poster_

### Official Review · Reviewer_EGJf · 2023-10-30

**Soundness:** 3 good
**Presentation:** 3 good
**Contribution:** 2 fair
**Rating:** 6
**Confidence:** 3

**Summary:**

**Summary:**
This paper presents an open-source toolkit based on LoRa. I believe this work might be more appropriate for the "benchmarking and datasets" track. Positioned here, it's challenging for me to evaluate the innovation this paper offers.

**Strengths:**

**Remarks:**
While the improvements and variants on LoRa are relatively straightforward, the theoretical part of the paper seems sound.

**Weaknesses:**

**Recommendation:**
I would advise the authors to provide clear insights through experiments and offer some specific suggestions.

**Questions:**

I cannot evaluate this paper because I believe it is proper for a benchmarking and dataset track, not the main track.

---

> ### Author Response · Authors · 2023-11-15
>
> Dear Reviewer,
>
> Thank you for taking the time to read and review our paper! We address your comments in detail below.
>
> > **I would advise the authors to provide clear insights through experiments and offer some specific suggestions.**
>
> In response to your advice for clearer insights and specific suggestions through experiments, we have introduced a new table (Table 1) in the revised manuscript. This table is aimed at providing a straightforward guide for readers in choosing suitable algorithms for various fine-tuning scenarios. For instance, based on our comprehensive experiments, we suggest that LoHa may be more effective for fine-tuning multiple "easier" concepts, whereas LoKr with full dimension appears better suited for targeting single "harder" concepts. This insight is reflected in the revised conclusion of our paper to provide more specific guidance.
>
> In the meantime, we want to emphasize that while our summarized findings and suggestions provide a useful overview, they should be considered with caution. The complexity of fine-tuning text-to-image models, as discussed in detail in appendix C, means that a simplified summary could hardly capture the full picture. In this regard, our detailed text discussion offers a more nuanced explanation of the various ways in which a hyperparameter can influence the results, and is indispensable for a more comprehensive understanding of the fine-tuning process.
>
>
> > **I cannot evaluate this paper because I believe it is proper for a benchmarking and dataset track, not the main track.**
>
> We appreciate your perspective regarding the suitability of our paper for a benchmarking and dataset track. While our work does include benchmarking, it is important to highlight that these are not its only contributions. A major part of our research introduces and elaborates on new fine-tuning method such as LoHa and LoKr. Moreover, the introduction of the library should also be regarded as an independent contribution. Generally speaking, our work aims to identify the specific fine-tuning methods that are more appropriate for a task of interest. Many works in this line of research (discussed in detail in Appendix A) have been previously accepted by the main track of the conferences, such as
>
> - Zhiheng Liu et al. Cones: Concept neurons in diffusion models for customized generation. In ICML, 2023.
> -  Yuchao Gu et al. Mix-of-show: Decentralized low-rank adaptation for multi-concept customization of diffusion models. In NeurIPS, 2023.
> -  Zeju Qiu et al. Controlling text-to-image diffusion by orthogonal finetuning. In NeurIPS, 2023
> - Nupur Kumari et al. Multi-concept customization of text-to-image diffusion. In CVPR, 2023.
> - Nataniel Ruiz et al. Dreambooth: Fine tuning text-to-image diffusion models for subject-driven generation. In CVPR, 2023.
>
> Finally, we do not believe ICLR possesses a separate "dataset and benchmark" track. In fact, on the [call for paper page of ICLR 2024](https://iclr.cc/Conferences/2024/CallForPapers), datasets and benchmarks is clearly indicated as a valid subject area, even though in our case, as explained, benchmarking serves more to support and validate the novel contributions, rather than being the sole focus of our work.

---

> > ### Comment · Reviewer_EGJf · 2023-12-04
> > **Official comment  by Reviewer EGJf**
> >
> > Thanks for the authors' reply. I read all the reviewers' comments and am happy to raise my score to 6.

---

### Official Review · Reviewer_DWom · 2023-10-31

**Soundness:** 3 good
**Presentation:** 3 good
**Contribution:** 3 good
**Rating:** 6
**Confidence:** 3

**Summary:**

This paper proposes a comprehensive library for evaluating text-to-image finetuning methods, typically based on LoRA. In addition to different algorithms, it also provides comprehensive evaluation criteria. Finally, some experimental results provide some insight about different finetuning methods.

**Strengths:**

1. This is a good engineering paper that provides a library for text-to-image finetuning methods evaluation.
2. It support different matrix factorization techniques such as LoRA, LoHa, LoKr, DyLoRA, GLoRA, GLoKr and so on.
3. This paper also consider comprehensive evaluation metrics, including fieldity, controllability, diversity, base model preservation and image quality.

**Weaknesses:**

1. This paper mainly focus on LoRA-based finetuing strategies, can it be expanded to other parameter-efficient finetuning methods such as [1] and [2]? It doesn't provide a clear explanation.
2. The conclusion about the performance of different finetuning methods is not clearly presented in the experimental section. Maybe some tables can more straightforwardly represent your final conclusions.

[1] Qiu, Zeju, et al. "Controlling Text-to-Image Diffusion by Orthogonal Finetuning." arXiv preprint arXiv:2306.07280 (2023).
[2] Xie, Enze, et al. "DiffFit: Unlocking Transferability of Large Diffusion Models via Simple Parameter-Efficient Fine-Tuning." arXiv preprint arXiv:2304.06648 (2023).

**Questions:**

Please refer to the weakness section.

---

> ### Author Response · Authors · 2023-11-15
>
> Dear Reviewer,
>
> Thank you for your feedback and for raising important points regarding our paper. Please find below our responses to your questions.
>
> > **This paper mainly focus on LoRA-based finetuing strategies, can it be expanded to other parameter-efficient finetuning methods such as [1] and [2]?**
>
> While our paper is primarily focused on LoRA-type methods, it is completely possible to include other methods in our library and experiments.
>
> In fact, IA3 that we mentioned in the paper provides a different strategy via fine-tuning a multiplicative term, which is later generalized by orthogonal fine-tuning (OFT) [1] that you mention. Although we only got aware of this work after submission of the paper, we have already included it in the latest version of LyCORIS. This is also indicated in the revised version of the paper.
>
> On the other hand, as far as we understand, DiffFit [2] is tailored to class-conditional models and can otherwise be regarded as a more restrained case of IA3 with several additional components, notably the bias and normalization layers, being fine-tuned. We can thus readily configure the training to get something that is really close with earlier version of our library.
>
> Finally, the choice of the focus on LoRA, LoHa, LoKr, and native fine-tuning in the paper are due to their wider use in the community, but we are also considering conducting the same experiments for other methods, and notably OFT.
>
>
>
> > **The conclusion about the performance of different finetuning methods is not clearly presented in the experimental section. Maybe some tables can more straightforwardly represent your final conclusions.**
>
> Following your suggestion, we have added a new table (Table 1) in the revised version of our paper to demonstrate the influence of algorithm choice in the fine-tuning process. This addition should enhance clarity and accessibility for readers, providing a more direct understanding of our findings. Thank you for this valuable suggestion!
>
> In addition to the table, the SHAP beeswarm charts readily included in our original paper serve as another effective tool for illustrating the influence of hyperparameters. However, it is crucial to note that many subtleties and context-specific insights discussed in our text cannot be fully captured by these visualizations alone. The readers should thus consider both the table and the charts in conjunction with the detailed textual explanations to gain a comprehensive understanding of the influence various factors on the fine-tuning process.

---

### Official Review · Reviewer_PnHf · 2023-11-01

**Soundness:** 4 excellent
**Presentation:** 4 excellent
**Contribution:** 3 good
**Rating:** 6
**Confidence:** 4

**Summary:**

This author introduces LyCORIS, an open source library dedicated to fine-tuning of Stable Diffusion, which integrates a comprehensive range of finetuning methods. For rigorous comparisons between the implemented methods, the author proposes a comprehensive evaluation framework that incorporates a wide range of metrics. Based on the evaluation framework, the author performs extensive experiments to compare different fine-tuning algorithms and to assess the impact of the hyperparameters (i.e, training epochs, learning rate, trained layers, et al). Overall, the experiments, comparisons, analyses, and results of the entire paper are very well-rounded and thorough.

**Strengths:**

1. Developing an open-source library is of great significance in fostering the advancement of a particular field. After comparing the existing open-source libraries available online, the LyCORIS library offers a relatively more comprehensive set of algorithms.

2. The author has developed a comprehensive benchmark to evaluate various algorithms from multiple perspectives, addressing a significant gap in the text-to-image field. This thorough evaluation and comparison of existing finetuning methods have been lacking in the domain until now.

3. The author conducted comprehensive experiments for different algorithms and parameters; in addition, the author also provided a detailed analysis of the current mainstream fine-tuning algorithms.

**Weaknesses:**

1. HuggingFace has also released the PEFT library, which supports a wider range of pre-trained models and includes the methods mentioned in the paper. Therefore, what are the advantages of the LyCORIS library compared to PEFT?

2. The paper conducted a multitude of experiments and comparisons on existing methods and various hyperparameters, leading to certain conclusions. Based on these findings, could there be a more optimal algorithm or design compared to previous ones?

**Questions:**

For this kind of paper that builds benchmarks based on a certain field, I would recommend the author to submit to a journal.

---

> ### Author Response · Authors · 2023-11-15
>
> Dear Reviewer,
>
> Thank you for your insightful comments and questions. Please find our responses to your comments below.
>
> > **HuggingFace has also released the PEFT library, which supports a wider range of pre-trained models and includes the methods mentioned in the paper. Therefore, what are the advantages of the LyCORIS library compared to PEFT?**
>
> We appreciate your question about the advantages of the LyCORIS library compared to HuggingFace's PEFT. PEFT indeed provides a wide collection of parameter-efficient fine-tuning methods. However, there are two key distinctions that set LyCORIS apart.
> - LyCORIS is specifically designed to cater to users with less expertise in AI and Stable Diffusion, making it more accessible to a wider audience. In contrast, PEFT is tailored more towards expert users.
> - In terms of fine-tuning of text-to-image models, LyCORIS covers more methods than PEFT. In fact, while PEFT initially supported LoRA, our library has incorporated other methods, with some of them, notably LoHa and LoKr, later got implemented by PEFT.
>
> > **The paper conducted a multitude of experiments and comparisons on existing methods and various hyperparameters, leading to certain conclusions. Based on these findings, could there be a more optimal algorithm or design compared to previous ones?**
>
> We conducted extensive experiments and comparisons on existing methods and various hyperparameters. While these studies have yielded important insights, we agree that asserting the existence of a universally "optimal" algorithm is challenging. The effectiveness of any given approach largely depends on specific datasets and tasks. For example, as we indicate in the conclusion of the revised paper, our experiments seem to suggest that LoHa would fit better the case of fine-tuning for multiple easy concepts while LoKr could be more suitable for the case of fine-tuning for a single difficult concept.
> Besides this, our findings offer guidance on hyperparameter choices, while acknowledging the dynamic and context-dependent nature of performance in this field.
>
>
> > **For this kind of paper that builds benchmarks based on a certain field, I would recommend the author to submit to a journal.**
>
> Your suggestion to submit our work to a journal is appreciated. While benchmarking is a significant contribution of our paper, it is not the sole one. Our work also introduces innovative approaches like LoHa and LoKr, which represent important advancements in the field. Therefore, while a journal submission is certainly a viable path, we believe our paper’s diverse contributions warrant consideration in its current format and venue.

---

### Official Review · Reviewer_ekPo · 2023-11-06

**Soundness:** 3 good
**Presentation:** 3 good
**Contribution:** 4 excellent
**Rating:** 8
**Confidence:** 4

**Summary:**

The authors propose LyCORIS, an open-source library that contains multiple fine-tuning techniques for Stable Diffusion. The authors also explore many improved fine-tuning techniques such as LoCon, LoHa and LoKr. This paper also presents evaluations for different fine-tuning techniques using multiple metrics and prompt types.

**Strengths:**

(1) The theory and experiments are both solid. The paper has over 57 pages devoted to analyzing the fine-tuning techniques.
(2) The details for experiments are very clear.
(3) In addition to the framework, the authors also explore other fine-tuning techniques.

**Weaknesses:**

(1) The results of this framework combined with ControlNet can be presented in this paper.
(2) Efficiency (time and GPU memory cost) of different approaches are not provided and analyzed.

**Questions:**

(1) Please refer to the main questions in the weakness section.
(2) A minor question: It will be better if the authors provide the results on other versions of stable diffusion, such as SD2.0 and SDXL.

---

> ### Author Response · Authors · 2023-11-15
>
> Dear Reviewer,
>
> Thank you for your encouraging comments and valuable feedback. We have addressed each of the points raised in your review as follows.
>
> > **The results of this framework combined with ControlNet can be presented in this paper.**
>
> We appreciate your suggestion in combining our framework with ControlNet. This is an interesting idea. StabilityAi indeed implemented LoRA for ControlNet during the writing of this paper [1]. However, their dimension is relative large (256 and 128) due to the heavy training workload and task drift. The introduction of other methods implemented in LyCORIS for ControlNet may then be helpful in this regard. While these have not yet been included in our library, we acknowledge their potential and are committed to exploring this integration in future work.
>
>
> [1] https://huggingface.co/stabilityai/control-lora
>
>
> > **Efficiency (time and GPU memory cost) of different approaches are not provided and analyzed.**
>
> To address this point, we have included a summary of the time and GPU memory costs of different configurations in Table 5 of the revised version of our paper. It is worth noticing that both the actual VRAM usage and training efficiency can vary greatly depending on the implementation and the used hardware. For example, we have modified the implementation of native training in LyCORIS after the experiments, and this should lead to faster training compared to what we report in the table.
>
> > **It will be better if the authors provide the results on other versions of stable diffusion, such as SD2.0 and SDXL.**
>
> Our library has added the support for other versions of Stable Diffusion, such as SD2.0 and SDXL. In the paper, we choose to focus on SD 1.5 due to its wider use and larger user base in the community. Moreover, due to page limitations, we are constrained in the number of experiments we can include. Nevertheless, we believe the experiment based on SD 1.5 already provides a substantial assessment of our framework's capabilities.

---

> > ### Comment · Reviewer_ekPo · 2023-12-05
> > **Thank authors for the rebuttal**
> >
> > Thank authors for the rebuttal. I will keep my original score unchanged.

---

### Author Response · Authors · 2023-11-15
**Modification in the revised paper and supplementary**

Dear Reviewers,

We would like to thank all of you for your appreciation of our work and for your valuable feedback. We have revised our paper accordingly. In particular, we have made the following changes which are highlighted in blue in the revised paper:

1. Presentation of the results: We have introduced a new table (Table 1) in the revised version of our paper to clearly demonstrate the influence of algorithm choice on the fine-tuning process. This table provides a direct overview of our findings, facilitating a more straightforward understanding for readers. Additionally, we have refined the conclusion section to offer some example guidance based on our experimental results.
2. Orthogonal fine-tuning (OFT): We have now incorporated OFT in the LyCORIS library and included it in our related work section in Appendix A.
3. Time and GPU costs: Table 5 now features the time and GPU costs of each configuration that we consider in the experiments. This addition should offer valuable insights into the practical aspects of the fine-tuning methods discussed in our paper.

In addition to these changes, we have addressed the comments of the reviewers individually in our responses. Please let us know if you have any further questions or require additional clarifications.

Sincerely,
The authors



[1] Zeju Qiu, Weiyang Liu, Haiwen Feng, Yuxuan Xue, Yao Feng, Zhen Liu, Dan Zhang, Adrian Weller, and Bernhard Schölkopf. Controlling text-to-image diffusion by orthogonal finetuning. In Advances in neural information processing system, 2023.

---

### Meta-Review · Area_Chair_TqtY · 2023-12-06

**Metareview:**

This paper developed LyCORIS, an open source library dedicated to fine-tuning of Stable diffusion with methods like LoRA, LoHa, LoKr, GLoRA, GLoKR, etc. Authors also proposed a comprehensive evaluation framework that incorporates a wide range of metrics, compared the performances of different tine-tuning algorithms. As pointed by reviewers the strengths of this paper are: 1) developing an open-source library is of great significance in fostering the advancement of a particular field; 2) a comprehensive benchmark to compare existing finetuning methods that have been lacking in the domain until now. Weaknesses are: 1) HuggingFace has also released the PEFT library, which supports a wider range of pre-trained models and includes the methods mentioned in the paper; 2) This is more like benchmark paper the technical novelty is limited.

3 reviewers gave "6: marginally above the acceptance threshold" ratings and 1 gave "8: accept, good paper" rating.

**Justification For Why Not Higher Score:**

1) HuggingFace has also released the PEFT library, which supports a wider range of pre-trained models and includes the methods mentioned in the paper; 2) This is more like benchmark paper the technical novelty is limited.

**Justification For Why Not Lower Score:**

All the reviewers support acceptance of this paper.

---

### Decision · Program_Chairs · 2024-01-16

Accept (poster)